# Large-scale climate signals of a European oxygen isotope network from tree-rings

Daniel F. Balting[1], Monica Ionita[1], Martin Wegmann[1], Gerhard Helle[2], Gerhard H. Schleser[3], Norel Rimbu[1], Mandy B. Freund[4,5], Ingo Heinrich[2,6], Diana Caldarescu[1] & Gerrit Lohmann[1,7]

[1]Alfred Wegener Institute, Bremerhaven, 27570, Germany
[2]German Research Centre for Geosciences, Potsdam, 14473, Germany
[3]IBG-3, Forschungszentrum Jülich, 52428, Germany
[4]Climate and Energy College, University of Melbourne, Melbourne, 3010, Australia
[5]CSIRO Agriculture and Food, Melbourne, Australia
[6]Geography Department, Humboldt University, Berlin, 10099, Germany
[7]Physics Department, University of Bremen, Bremen, 28359, Germany

*Correspondence to*: Daniel F. Balting (daniel.balting@awi.de)

**Abstract.** We investigate the climate signature of $\delta^{18}O$ tree ring records from sites distributed all over Europe covering the last 400 years. An Empirical Orthogonal Function (EOF) analysis reveals two distinct modes of variability on the basis of the existing $\delta^{18}O$ tree ring records. The first mode is associated with anomaly patterns projecting onto the El Niño-Southern Oscillation (ENSO) and reflects a multi-seasonal climatic signal. The ENSO link is pronounced for the last 130 years, but is found to be weak over the period 1600 to 1850, suggesting that the relationship between ENSO and the European climate may not be stable over time. The second mode of $\delta^{18}O$ variability, which captures a north-south dipole in the European $\delta^{18}O$ tree ring records, is related to a regional summer atmospheric circulation pattern revealing a pronounced centre over the North Sea. Locally, the $\delta^{18}O$ anomalies associated with this mode show the same (opposite) sign with temperature (precipitation). Based on the oxygen isotopic signature derived from tree rings, we argue that the prevailing large-scale atmospheric circulation patterns and the related teleconnections can be analysed beyond instrumental records.

## 1 Introduction

Tree growth is irrevocably affected by interactions with hydrosphere, atmosphere, and pedosphere and the influence of environmental factors is stored in the physical and chemical properties of each tree ring (Schweingruber, 1996). A major component of a tree ring is cellulose, which consists of the elements carbon, oxygen and hydrogen. Their stable isotope signatures are determined by varying environmental conditions influencing a series of fractionation processes during the uptake of $CO_2$ and $H_2O$ from atmosphere and soil and biosynthesis of tree-ring cellulose. For instance, the climate signature of $\delta^{13}C$ values of tree-ring cellulose basically originates from fractionations during photosynthesis at the leaf or needle level that generally lower the $\delta^{13}C$ of the atmospheric $CO_2$ source which contains no direct climatic signal (e.g., Schleser et al., 1995). $\delta^{18}O$ of tree-ring cellulose ($\delta^{18}O_{cel}$) is of particular interest for paleoclimate studies because it is related to source water, i.e., $\delta^{18}O$ of precipitation ($\delta^{18}O_P$), which is directly affected by climate processes, such as temperature during droplet

condensation within air masses, transport distance from ocean source, type of precipitation (e.g., rain or snow) and precipitation amount (e.g., Dansgaard, 1964; Epstein et al., 1977; Rozanski et al., 1993). Within the arboreal system, $\delta^{18}O$ of soil water ($\delta^{18}O_{SW}$) constitutes the $\delta^{18}O$ input and usually represents an average $\delta^{18}O_P$ over several precipitation events modified by partial evaporation from the soil (depending on soil texture and porosity) and by a possible time lag, depending on rooting depth (Saurer et al., 2012). Representing the baseline variability, the oxygen isotope signature of tree-ring cellulose ($\delta^{18}O_{cel}$) is invariably tied to $\delta^{18}O_{SW}$.

However, $\delta^{18}O_{cel}$ is dependent on two more clusters of fractionations that reflect tree-internal processes, namely evaporative $^{18}O$-enrichment of leaf or needle water via transpiration, and biochemical fractionations including partial isotopic exchange of cellulose precursors with trunk water during cellulose biosynthesis (e.g., Saurer et al., 1997; Roden et al., 2000; Barbour, 2007; Kahmen et al., 2011; Treydte et al., 2014 and citations therein). The biochemical fractionation during cellulose biosynthesis can be largely considered as constant at 27 ±4‰ (Sternberg and DeNiro, 1983). Nonetheless, varying leaf-to-air vapour pressure deficit together with a varying air humidity causes corresponding changes in the $\delta^{18}O$ signature of leaf or needle water (e.g., Helliker and Griffiths, 2007). Although modified and dampened by physiological processes (e.g., Pèclet effect (Farquhar and Lloyd, 1993)) and oxygen isotope exchange with stem water during cellulose synthesis (Hill et al., 1995) variability of $^{18}O$ enrichment of leaf-water clearly affects $\delta^{18}O_{cel}$, besides the strong signature of $\delta^{18}O_P$. For example, the $\delta^{18}O_{cel}$ are used to reconstruct precipitation (e.g., Rinne et al., 2013), air temperature (e.g., Porter et al., 2014) and drought (e.g., Nagavciuc et al., 2019). Since these quantities are largely based on transport processes within the atmosphere, the $\delta^{18}O_{cel}$ values can be used to get detailed information about large-scale atmospheric circulation patterns (Brienen et al., 2012; Lavergne et al., 2016; Andreu-Hayles et al., 2017; Trouet et al., 2018, Nagavciuc et al., 2019, 2020). The resulting long-term perspective on the climate from the usage of $\delta^{18}O_{cel}$ as a climate proxy may be the key to identify the influence of different external forcing on, and internal variability of the behaviour of large-scale atmospheric circulation.

One of the most important components for the internal climate variability is the El Niño – Southern Oscillation (ENSO) which influences the atmospheric circulation globally (Allan et al., 1996). Since ENSO variability is strongest in winter, multiple studies have identified a significant ENSO impact on the European climate during this season. Observational studies (Fraedrich and Müller, 1992; Fraedrich, 1994; Brönnimann et al., 2004; Pozo-Vazquez et al., 2005; Brönnimann et al., 2007) and model studies (Merkel and Latif, 2002; Mathieu et al., 2004) suggest that an El Niño event leads to a negative phase of the North Atlantic Oscillation (NAO) with cold and dry conditions over Northern Europe and wet and warm conditions over southeaster Europe in winter. Furthermore, it is also possible to identify a significant ENSO influence with regard to European precipitation in spring (van Oldenborgh, 1998; Lloyd-Hughes and Saunders, 2002; Brönnimann et al., 2007; Helama et al., 2009). However, significant impacts of ENSO regarding European droughts could only be detected for the most extreme El Niño events (King et al., 2020). To what extent these preconditions influence the climate conditions during summer is not yet known for Europe, but it can serve as a key for a better understanding of European climate. However, the relatively short period of existing instrumental data (van Oldenburgh and Burgers, 2005; Brönnimann, 2007), makes it difficult to describe the full range of ENSO variability and its possible consequences for the climate of the

European continent (Domeisen et al., 2019). Furthermore, the response of different climate variables to ENSO variability is either non-stationary (e.g., Fraedrich and Müller, 1992) and/or non-linear. For example, Wu & Hsieh (2004) showed that the large-scale atmospheric circulation response to El Niño and La Niña is asymmetrical.

The aim of this study is to present a comprehensive spatio-temporal analysis of the large-scale European atmospheric circulation based on the climatological signals of a European $\delta^{18}O_{cel}$ network, extending back for ~400 years. This study is based on the climatological signals of a European $\delta^{18}O_{cel}$ network. The climate signal of the isotope network is extracted by using an Empirical Orthogonal Functions (EOF) analysis. The results of the first two components are compared to climate data and different ENSO reconstructions. The comparison is done by a composite analysis and by correlation analysis. To test if $\delta^{18}O_{cel}$ can capture multi seasonal signals, the first component of the isotope network is correlated with gridded fields of modelled $\delta^{18}O_P$ and $\delta^{18}O_{SW}$. Finally, the major results are critically discussed and compared to other studies.

## 2 Data and Methods

### 2.1 The isotope network

In this study, we investigate the first two dominant modes of variability of 26 $\delta^{18}O_{cel}$ records, distributed over Europe, and their relationships with regional and large-scale climate anomalies. 22 of the 26 $\delta^{18}O_{cel}$ records were generated within the EU project ISONET (Annual Reconstructions of European Climate Variability using a High-Resolution Isotopic network) (Treydte et al., 2007a, b). Furthermore, four additional sites from Bulgaria, Turkey, Southwest Germany and Slovenia were added to the ISONET network for the current study. (Heinrich et al., 2013; Hafner et al., 2014). In total, the isotope network contains eight broadleaf tree sites (*Quercus*) and 18 coniferous tree sites (*Pinus, Juniper, Larix, Cedrus*) from altitudes varying for each location from 10 m up to 2200 m above sea level (Fig. 1 and Supp. 1). 24 of the 26 sites are distributed over the European continent whereas two additional sites are located in the Atlas Mountains of Morocco and in the Taurus Mountains of Turkey.

The stable isotopes of oxygen in tree-ring cellulose, reported as $\delta^{18}O_{cel}$ vs. Vienna Standard Mean Ocean Water (Craig, 1957) of each site were determined as described by Treydte et al. (2007a, b). At least four dominant trees were chosen per site and two increment cores were taken per tree. After the standard dendrochronological dating following Fritts (1976), the individual tree rings were dissected from the cores. According to Treydte et. al (2007a, b), all tree rings from the same year were pooled prior to cellulose extraction for the majority of sites. However, for oak only the latewood was used for the analyses. This procedure assumed that climate signals of the current year were predominantly applied since early wood of oaks frequently contains climate information of the preceding year. This is based on the fact that the proportion of the reserves of deciduous trees of the isotope network is higher at the beginning of the tree ring formation as in comparison with conifers of the isotope network because they are evergreen. The temporal resolution of the isotope records is annually.

The first 100 years of data from the network as well as a general description have already been published (Treydte et al., 2007a, b). Data from individual sites or regional groups of sites were published elsewhere (Saurer et al., 2008; Vitas, 2008; Etien et al., 2009; Hilasvuori et al., 2009; Haupt et al., 2011; Saurer et al., 2012; Rinne et al., 2013; Helama et al., 2014; Labuhn et al., 2014; Saurer et al., 2014; Labuhn, et al., 2016; Andreu-Hayles et al., 2017). Here, we use the extended ISONET+ product where the longest chronologies cover a period from 1600 to 2005. The highest data density is available for the period 1850-1998 with 26 time series available for further analysis. 12 time series cover the entire period of 400 years.

## 2.2 Climate data

For gridded the climate information, we used the gridded fields of monthly temperature averages and monthly precipitation sums from the Climatic Research Unit version 4.04 (Harris et al., 2020). Both quantities are derived by the interpolation of monthly climate anomalies from extensive networks of weather station observations. The CRU TS dataset has a spatial resolution of 0.5° x 0.5° and they cover the period 1901 – 2019.

For the large-scale atmospheric circulation, we make use of the gridded fields of geopotential height 500 mb (Z500) from the 20th Century Reanalysis Project (20CR) version V2c (Compo et al., 2011). The 20CR reanalysis dataset has a temporal resolution of six hours and a meridional and zonal resolution of 2°. In our study, the ensemble mean of 20CR is used, which was computed from 56 ensemble members. The climate variables are available for the period from 1851 to 2014 and they are provided by NOAA/OAR/ESRL PSL, Boulder, Colorado, USA (downloadable from their website https://www.psl.noaa.gov/data/gridded/data.20thC_ReanV2c.html).

Since precipitation and temperature are both important for the $\delta^{18}O_{cel}$ ratio, we test the relation between the isotope ratio and drought conditions at European scale. For this purpose, the Standardized Precipitation Evapotranspiration Index dataset from Vicente-Serrano et al. (2010) is used in this study with an aggregation time of three months (SPEI3). The SPEI3 index is suitable for this analysis because we take into account the climate conditions of the pre-season.

Furthermore, the Extended Reconstructed Sea Surface Temperature version 5 (ERSST5; Huang et al., 2017) is included in the study to investigate the correlation between the global sea surface temperature (SST) and the climate signals of $\delta^{18}O_{cel.}$ The ERSST5 dataset was created by Huang et al. (2017) and it is derived from the International Comprehensive Ocean–Atmosphere Dataset. The monthly SST fields has a spatial resolution of 2° x 2° and they are available for the time range from 1854 to present. The gridded fields of the ERSST5 dataset are provided by NOAA/OAR/ESRL PSL, Boulder, Colorado, USA and they can be downloaded from their website (www1.ncdc.noaa.gov/pub/data/cmb/ersst/v5/netcdf/). All of the mentioned climate data were seasonally averaged: DJF (December to February), MAM (March to May), JJA (June to August) and SON (September to November). Furthermore, the linear trend from each grid cell is removed.

Since our study is focused on the relation between ENSO variability and the $\delta^{18}O_{cel}$ network, we used also the anomaly (1981-2010 mean removed) of the December Nino 3.4 index (HadISST1; Rayner et al., 2003). The index represents the

averaged SST from 5°S-5°N and 170°-120°W (downloadable from https://psl.noaa.gov/gcos_wgsp/Timeseries/Nino34/). To compare the relation between ENSO variability and our Isotope network variability, also in the past, three different reconstruction of ENSO activity have been used in this study: i) the reconstruction of Dätwyler et al. (2019) of the annually Niño3.4 index for the last millennium (based on a multiproxy network); ii) the reconstruction of NDJ Niño3.4 index for the last 700 years by Li et al. (2013) (based on a tree-ring network in the tropics and mid-latitudes) and iii) an annually reconstruction of ENSO variability based on the North American Drought Atlas (Cook et al., 2004, Li et al., 2011).

## 2.3 Modelled $\delta^{18}O$ in precipitation and soil water

Beside the observational/reanalysis-based climate data, we also investigate the relation between $\delta^{18}O_{cel}$ and $\delta^{18}O_P/\delta^{18}O_{SW}$, as simulated by the water isotope enabled ECHAM5-wiso model (Werner et al., 2011) in order to get further insights regarding the correlation with other seasons and to identify a multi-seasonal climate signal. The $\delta^{18}O_P/\delta^{18}O_{SW}$ dataset was created in the study of Butzin et al. (2014) where they used the isotope-enabled version of the atmospheric general circulation model ECHAM5 (Roeckner et al., 2003; Hagemann et al., 2006; Roeckner et al., 2006) which is called ECHAM5-wiso (Werner et al., 2011). In the study of Butzin et al. (2014), values of present-day insolation and greenhouse gas concentrations (Intergovernmental Panel on Climate Change, 2000) and monthly varying fields of sea-surface temperatures and sea-ice concentrations according to ERA-40 and ERA-Interim reanalysis data (Uppala et al., 2005; Berrisford et al., 2011; Dee et al., 2011) are used to force the model. To represent the climate conditions of the period 1960 to 2010, Butzin et al. (2014) used an implicit nudging technique (Krishamurti et al., 1991; the implementation in ECHAM is described by Rast et al., 2013). The nudging technique is a part of climate modelling sciences, where modelled fields of climate variables are relaxed to observations or data from reanalysis. In the study of Butzin et al. (2014), the modelled fields of surface pressure, temperature, divergence and vorticity are coupled to ERA-40 and ERA-Interim reanalysis fields (Uppala et al., 2005; Berrisford et al., 2011; Dee et al., 2011) for the period 1960 to 2010. The monthly grids of $\delta^{18}O_P/\delta^{18}O_{SW}$ have a horizontal grid size of approximately 1.9° x 1.9°. Since our study is focused on seasonal variability, we have computed the seasonal averages for the $\delta^{18}O_P/\delta^{18}O_{SW}$ (DJF, MAM, JJA, SON).

## 2.4 Data analysis

As a first step, the characteristics of each time series of the $\delta^{18}O_{cel}$ network and their relation to altitude and latitude are investigated. For a better comparison, the linear trend of each $\delta^{18}O_{cel}$ time series is removed and the time series are standardized (z-values).

To combine the signals of the isotope network, we use the Principal Component Analysis (PCA) and the Empirical Orthogonal Functions (EOF). These techniques were described by Pearson (1902) and Hotteling (1935) and were used for the first time by Lorenz (1956) for climatological studies (Storch and Zwiers, 1999). By applying the PCA, it is possible to extract a common climate signal from $\delta^{18}O_{cel}$ network which explains the highest part of the variability of the input dataset.

This is done by rotating the initial data onto axes which are orthogonal to each other (Schönwiese, 2013) by the corresponding eigenvectors. Therefore, the eigenvectors are used as a transformation matrix. The corresponding analysis of the eigenvector values is known as Empirical Orthogonal Functions (EOF). The goal of it is to identify the most dominant patterns of the $\delta^{18}O_{cel}$ tree network variability which explain a significant part of the variance for a specific region. The largest part of the variance can be explained by the pattern of the leading EOF. The temporal perspective on these patterns is given by the principal components (PC) which describe the phase and the amplitude.

The resulting components are further checked if they fulfil the requirements of the rule of North et al. (1982). This rule states that the pattern of the eigenvectors of one component is strongly contaminated by other EOFs that correspond to the closest eigenvalues (Storch and Zwiers, 1999). To determine whether two consecutive patterns can be interpreted as distinct patterns, it is necessary to calculate the standard error in the estimation of the eigenvalues which is according to North et al. (1982) approximately

$$\Delta\lambda \sim \lambda\sqrt{2/n}$$

where λ is the eigenvalue and n the number of degrees of freedom of the data set. In case that the eigenvalues of two EOF's are not more separated from each other than this standard error, it is unlikely that the two consecutive patterns can be interpreted as distinct from each other, since any linear combination of the two eigenvectors is equally significant. In our study, we consider only that eigenvectors where the successive eigenvalues are distinguishable.

The ISONET network consists of a multi-site and multi-species tree-ring network covering more or less the period between 1600 to 2003. However, some tree-ring series cover the whole period, others cover only a shorter period. In order to be able to have a long-term perspective, one needs to find a statistically meaningful way to extend the shorter records to make use of the whole 400 years of data. Since most Multilinear Principal Component Analysis algorithms do not work with gaps in the initial matrix, we make use of an algorithm developed by Josse and Husson (2016) which is able to fill the temporal gaps without a change of the PC. In the first step we place the mean in the gap and execute a PCA with this dataset. Afterwards, the dataset is projected onto the new component axis. So, that the values are rotated and the value of a gap change. The new value for the gap is placed into the initial dataset. With this new dataset, a PCA is again carried out. This process is repeated until convergence is reached. The result is a gap-free dataset which can be used for PCA. To quantify if our results are influenced by the gap filling method, we tested the correlation between PC1 based on the ISONET network and the first 4 PCs of the Old-World Drought Atlas (OWDA; Cook et al., 2015) for the period 1850 to 2005 and 1600 to 2005. If the filling algorithm altered the representation of climate signals over a longer time period, we would expect that the strength of correlation is changing.

Composite maps of average precipitation, air temperature, geopotential height 500 mb (Z500) and SST are computed using years where the network's principal components are above or below a certain threshold. In our study, we choose maxima events above ($>\mu+\sigma$) and minima events below ($<\mu-\sigma$) one standard deviation ($\sigma$) with respect to the mean ($\mu$). The composite maps allow us to analyse the general climate state occurring at times of minima (low) or maxima (high)

separately. In addition, both composites of climate conditions can be combined under the assumption that the minima and maxima events show the opposite climate state. For this purpose, the minimum composite is subtracted from the maximum composite at each grid point (high-low). Beside the composite maps, we extract the values of PC1 for those years for which ENSO values are higher than the average plus one standard deviation and lower than the average minus one standard deviation. The difference from the former distribution for the values of minima and maxima years is tested with the t-test. To better understand if El Niño or La Niña events coevolve with extremes in the $\delta^{18}O_{cel}$ time series, the Event Coincidence Analysis (ECA) (Donges et al., 2016; Siegmund et al., 2017) using PC1 and a December Nino 3.4 index is applied (HadISST1; Rayner et al., 2003). The ECA quantifies the simultaneity of events contained in two series of observations which can be computed with the R package CoinCalc (Siegmund et al., 2017). Furthermore, the CoinCalc package provides functions to test if the coincidences are significant. In our study, we analyse whether the years in which the Nino 3.4 index is above (below) the 75th (25th) percentile match the 75th (25th) percentile in PC1. In general, a significance level of $\alpha= 0.05$ was used in all analyses.

Finally, we analyse the relation between seasonally averaged $\delta^{18}O_P$ and $\delta^{18}O_{SW}$ from winter, spring and summer, based on nudged ECHAM5-wiso simulations (Butzin et al., 2014), with PC1 based on the $\delta^{18}O_{cel}$ values from the ISONET network. Our goal is to test the correlation between $\delta^{18}O_{cel}$ and modelled $\delta^{18}O_P/\delta^{18}O_{SW}$ to identify if the water, which is used within the photosynthesis processes, has a multi-seasonal isotopic signature.

## 3 Results

### 3.1 Characteristics of the $\delta^{18}O_{cel}$ network

The highest mean $\delta^{18}O_{cel}$ values occur at the southern locations, i.e., in Turkey and Carzola in Spain. For oaks and pines, the lowest mean values are found for the northern sites (Fig. 2A). Moreover, generally lower $\delta^{18}O_{cel}$ values are identified for *Quercus* compared to *Pinus*. Angiosperm wood tissue contains vessels, i.e., specialized water-conducting cells that are generally larger in diameter, and therefore more conductive to water, than conifer wood cells (Sperry et al., 2006; Carnicer et al., 2013). The overall variance of the datasets is not dependent on the type of tree species. The $\delta^{18}O_{cel}$ time series from Poellau in Austria is characterized by the highest standard deviation whereas the time series from Lochwood in Great Britain shows the lowest standard deviation.

Since the $\delta^{18}O$ source values and the fractionation processes are temperature dependent, it is necessary to evaluate the influence of altitude and latitude for the oxygen isotope ratio. The relation of the $\delta^{18}O_{cel}$ values with regard to altitude and latitude of each site is shown in Figs. 2B and 2C. In Fig. 2C, the linear relationship between the average $\delta^{18}O_{cel}$ values of the locations and the corresponding latitudes are plotted. As shown by the boxplots of Fig. 2A, southern sample sites are characterized by the highest average $\delta^{18}O_{cel}$ values whereas northern sites show the lowest average $\delta^{18}O_{cel}$ values. The relation between the $\delta^{18}O_{cel}$ values and latitude is yielding a significant linear regression.

Beside the latitudinal effect, altitude also influences the oxygen isotope ratios as shown in Fig. 2B which can likewise be described by a significant linear regression. It should be noted, that the southern sites are found at higher altitudes than the northern sites, thus the latitudinal and altitudinal gradients may have confounding effects on $\delta^{18}O_{cel}$. Therefore, we show that the $\delta^{18}O_{cel}$ network is influenced by a latitudinal and an altitudinal effect.

## 3.2 Characteristics of the principal components

Based on the EOF analysis, the first component of the isotope network explains 16.2%, the second 9.1%, the third 6.4%, the fourth 5.5%, and the fifth 5.2% of the variance. Therefore, the first five components explain a cumulative variability of around 43%. Since the first two components also fulfilled the requirements of the rule of North et al. (1982), they are investigated by their temporal and spatial characteristics.

The dominant pattern (EOF1), which describes 16.2% of the total variance, shows a spatially homogeneous structure (Fig. 3A). The majority of time series in Europe are characterized by negative eigenvector values. The pole of this EOF pattern is centred over France and Germany. In contrast, tree sites close to the Mediterranean Sea and the northernmost site in Finland are characterized by eigenvector values close to zero. Therefore, these locations contribute to a lesser extent to the first component's time series (PC1) shown in Fig. 3C. The time series of PC1 is characterized by an underlying negative trend from the 17th to the 18th century. From the beginning of the 18th century, a positive trend is observed until the beginning of the 19th century, where the highest values are reached. For the last 150 years, no clear trend is visible.

The second EOF (EOF2) is characterized by a completely different spatial pattern (Fig. 3B). Negative eigenvector values are found around the North and Baltic Sea, with the smallest eigenvector located in Norway, while positive eigenvector values are identified over the Southern/Southeast Europe and the Alpine region. The highest eigenvector values are recorded for the Italian site. In summary, this component highlights a dipole-like structure between Northern and Southern/Southeast Europe. This dipole-like structure is a well-known feature of the European hydroclimate (Ionita et al., 2015). The time series of the second component (PC2) is characterized by an underlying positive trend from the mid of the 19th century onwards. Furthermore, the highest interannual variability is found for the beginning of the 18th and 19th century. The highest values of the second component (PC2) are reached at the beginning of the 18th century, whereas the smallest values are identified for the beginning of the 19th century (Fig. 3 D).

Furthermore, we tested the correlation between PC1 and the first 4 PCs for the summer drought reconstruction based on the Old-World Drought Atlas (OWDA; Cook et al., 2015) for the period 1850 to 2005, where we have the highest sample density. The highest correlation (R=0.43, p-value=3.1e$^{-08}$) is computed with PC2 of the OWDA (EOF plot and time series are shown in Supp. 2), which explains 16.1% of the total variance of the OWDA. If the filling algorithm altered the representation of climate signals over a longer time period, we would expect that the strength of correlation is also changing. Yet, the correlation is only slightly changing to Pearson's R=0.39 (p-value=4.2e$^{-16}$) for the entire period 1600 to 2005, which indicates that the filling algorithm does not impact the results. As a result, the climate signals obtained are robust and

presented in a similar manner as for the period with a high sample coverage (i.e., 1850-2005). This argumentation is also supported by the comparison of the two time series in Supp. 2. We note that by using the gap filling method, uncertainties might arise for the first century where the sample density is low. It is, therefore, important to mention that the interpretation for the first century needs to be handled with care, and statements should be regarded as less robust.

### 3.3 Climate signals of PC1

For the winter season, high values of PC1 are associated with significant warm SSTs in the Equatorial Pacific and on the west coast of North and South America (Fig. 4A), which indicates that the high values of PC1 are co-evolving with the occurrence of the El Niño conditions. The strong signal in the Pacific persists throughout spring (Fig 4C) and reduces in summer (Fig 4E). A significant warming of the Tropical and North Atlantic is visible in spring and summer (Fig 4C, E). According to Latif and Grötzner (1999), the lagged warming of the Equatorial Atlantic can be observed up to 6 months after an El Niño event. In contrast, the Northeast Atlantic and the Mediterranean Sea are characterized by significantly colder SSTs in all three seasons. The low composite maps for PC1 and SST (Fig. 4B, D, F) show features of La Niña conditions associated with colder than average SSTs. This pattern is particularly prominent in the Tropical Pacific during winter and spring.

The related large-scale atmospheric circulation is shown in the composite map of Z500 (Fig. 5). During high values of PC1 which are co-evolving with the occurrence of El Niño conditions, the atmospheric circulation over Europe is characterized by a low-pressure regime in winter, whereas high-pressure regimes can be identified over the Northwest Atlantic as well as in the east of the low-pressure system (Fig. 5A). This fits well to the composite map of air temperature in Fig. 6B which shows significant cold conditions over the North Europe and significant warm conditions for Southeast Europe. The described temperature pattern shows similarities to the effects of a negative phase of the winter NAO. The opposite climate state can be observed for low values of PC1 in winter (Fig. 5B, 6B).

The atmospheric circulation remains in a similar configuration over Europe and the North Atlantic for low and high composite maps for spring (Fig. 5C, 5D) compared to the winter season (Fig. 5C, D). One important difference to the preseason is that a pressure belt is visible between Europe and the Gulf of Maine in both maps.

The composite maps of Z500 for summer are characterized by a pressure regime centred over France and Germany (Fig. 5E, 5F). In both maps, it is visible that the low-/high-pressure regime in Central and Western Europe is surrounded by opposite pressure regimes. In case of a high-pressure system in the centre, it leads to a blocked zonal flow which is shown in Europe in Fig. 5F. The composite maps of precipitation and temperature are also supporting the analysis with the Z500 data. The climate of Central and Western Europe is characterized by significantly higher (lower) precipitation in Central Europe as well as significantly lower (higher) surface air temperatures in summer corresponding to low (high) $\delta^{18}O_{cel}$ values. The significant relation between PC1 and the summer hydroclimate is also resampled by the correlation between a SPEI3 index

for JJA (Vicente-Serrano et al., 2010; Longitude -5° to 10°/Latitude 46° to 52°) and PC1. The correlation is significant (R=0.49; p < 0.01) for the period from 1901 to 2005 which suggests that PC1 can capture the hydroclimate variability in summer.

To further investigate the relation between ENSO variability and the PC1, we apply two different statistical approaches. The first approach is to analyse if El Niño and La Niña events are separated in the probability density plots of PC1. During El Niño years, the distribution of the PC1 is shifted towards higher values, whereas the opposite occurs during La Niña years (see supplementary material; Fig. S3). According to the t-test, both shifts are significantly different compared to the distribution of PC1 (p < 0.05). The second statistical approach investigates if extremes in $\delta^{18}O_{cel}$ time series are co-occurring with El Niño or La Niña events. For this purpose, we apply the Event Coincidence Analysis (Siegmund et al., 2017; Donges et al., 2016) using PC1 and December Nino 3.4 index (HadISST1; Rayner et al., 2003). Over the period 1871-2005, 41.2% high and low extremes in the Nino 3.4 coincided significantly during winter, with high and low extremes of PC1 (p < 0.01). By extending the analysis period from 1750 to 1850, coincidence rates (28% % of the NDJ Nino 3.4 (Li et al., 2013) high and low extremes coincided not significantly (p>0.1) during winter with high and low extremes of PC1) and also the correlations (Supp. 3) between PC1 and ENSO reconstructions Dätwyler et al. (2019) and Li et al. (2011, 2013) are weakening.

### 3.4 Comparison of $\delta^{18}O_{cel}$ with modelled $\delta^{18}O$ in precipitation and soil water

By employing nudged climate simulations with ECHAM5-wiso (Butzin et al., 2014), we evaluate how the $\delta^{18}O_{cel}$ tree signature is related to the modelled $\delta^{18}O$ in precipitation and soil water. A significant correlation between PC1 and the modelled $\delta^{18}O_P$ is shown in the correlation maps for winter, spring and summer, where Central Europe is characterized by a moderate correlation (Fig. 7A-C). A similar pattern can be identified for the correlation between $\delta^{18}O_{SW}$ and PC1. Compared to our previous analysis, the correlation between these quantities is increasing from winter to summer where it reached the maximum correlation (Fig. 7G). Since the $\delta^{18}O_{cel}$ ratio is largely dependent on $\delta^{18}O_{SW}$, the relation with $\delta^{18}O_{SW}$ is stronger compared to $\delta^{18}O_P$. Overall, the results of Fig. 5 indicate that significant correlations for both quantities can be computed for entire Europe except the eastern parts.

### 3.5 Further climate signals in $\delta^{18}O_{cel}$

Besides the multi-seasonal signal, the second component of the $\delta^{18}O_{cel}$ values significantly relates to the summer climate (Fig. 8, 9). A positive (negative) geopotential height anomaly in northern Europe with the centre over the North Sea co-occurs with a negative (positive) Z500 anomaly in Southeast Europe (Fig. 9). This coincides with low (high) temperature in Central and North Europa whereas Northeast Europe is characterised by high (low) temperature (Fig. 8A). The same pattern is also shown in the composite maps for precipitation where a similar pattern is presented (Fig. 8B). Based on these patterns,

the temporal distribution of extremes in the PC2 time series, indicates that the 19th century has experienced increased dryness in northern Europe and enhanced precipitation in the Adriatic region (Fig. 3D).

## 4 Discussion

### 4.1 Latitudinal and altitudinal dependence of the $\delta^{18}O_{cel}$ network

The $\delta^{18}O_{cel}$ ratio is affected by the isotopic composition of the source water ($\delta^{18}O_P$, $\delta^{18}O_{SW}$), which varies according to the latitude and altitude of the sample site (e.g., McCarroll and Loader, 2004). The latitudinal position has an influence on the $\delta^{18}O$ in the atmosphere because of the strong correlation between the temperature and the $\delta^{18}O$ composition of water vapour in the atmosphere (Dansgaard, 1964). In addition, Dansgaard (1964) has suggested that altitude have an influence on the $\delta^{18}O$ ratio which primarily results from the cooling of air masses as they ascend a mountain, accompanied by the rainout of
the excess moisture (Gat, 2010).

The presented results show that there is a linear relationship between $\delta^{18}O_{cel}$ and site altitude (Figure 2B) and latitude (Figure 2C), which proofs the concept of the often-stated effects of latitude and altitude on the $\delta^{18}O_{cel}$ (McCarroll and Loader, 2004). Therefore, our results are in line with other studies which have already proved that concept, e.g., Szejner et al. (2016). The effects of altitude and latitude on the photosynthesis process are also visible for other tree-ring based proxies. For
example, according to the studies of Körner et al. (1991), Marshall & Zhang (1994) and Dietendorf et al. (2010), there is a highly significant altitude effect on the tree carbon isotope composition ($\delta^{13}C$). Since temperature decreases with increasing altitude and the partial air pressure is approximately 21% lower at 2000 m than at sea level, Körner et al. (2007) have argued that these environmental conditions lead to a faster molecular gas diffusion at any given temperature.

However, the distribution of our sample sites across Europe indicate that the isotope network is spatially limited (Fig. 1). For
instance, the time series from Central and Western Europe are overrepresented compared to the ones from Southeast and Northern Europe. In addition, it is imperative to extent the isotope network by collecting more $\delta^{18}O_{cel}$ records from Eastern Europe to improve the validity of our results for this region. Furthermore, the samples were not taken from trees growing at the same altitude, which is critical for identifying a latitudinal effect. In fact, the sampled trees in Southern Europe grew at higher altitudes ($\geq 1600$ m) compared to the other sample sites, which could also bias the $\delta^{18}O_{cel}$ ratio. It is, thereby, not
viable to compare two adjacent sites located at different altitudes, and challenging to distinguish between latitude and altitude effects.

Another point that needs to be considered is that the $\delta^{18}O$ time series used in this study were created using the pooling approach. Overall, the pooling approach was tested and proved successful for climate analysis (Treydte et al., 2007), but the approach also has some weaknesses and it is strongly discussed in the literature (e.g., Foroozan et al., 2019). For example,
Hangartner et al. (2012) recommend to avoid pooling, because an unsuitable tree cannot be omitted without resampling the whole period. They suggest to measure the trees individually instead, and to use pooling only by given strong correlation between the trees. Nevertheless, the pooling approach was adopted for all the ISONET sites allowing, for the first time, to

establish a tree-ring stable isotope network from more than 20 sites across Europe in collaboration with several laboratories within a reasonable time frame. Without pooling, the spatial data set, analysis and interpretation presented here would not be available for the community.

## 4.2 Links between ENSO and PC1

The presented results in Section 3.3 are an indicator that ENSO activity influences the climate signal of PC1. The reason for this is the described warm/cold SST pattern in the Equatorial Pacific and on the west coast of North and South America which are associated with El Niño and La Niña events (Allan et al., 1996). Furthermore, the clear and significant separation between El Niño and La Niña events in probability density functions and the coincidence of high and low values of Nino 3.4 and PC1 support our argumentation.

Moreover, the described temperature pattern in Fig. 6B resembles the effects of the winter NAO (Fig. 6B). During the negative phase of the NAO, North Europe experience cold and Southeast Europe warm conditions in winter. Studies based on observations (Fraedrich and Müller, 1992; Fraedrich, 1994; Brönnimann et al., 2004; Pozo-Vazquez et al., 2005; Brönnimann et al., 2007) and models (Merkel and Latif, 2002; Mathieu et al., 2004) have shown that ENSO variability can influence the winter NAO. Their results showed that an El Niño event leads to a negative phase of the NAO. Based on our argumentation, we suggest that high values of PC1 are co-evolving with the occurrence of the El Niño conditions and low values of PC1 are co-evolving with the occurrence of La Niña conditions.

## 4.3 The stability of the ENSO signal in the isotope network

In our study, we test the correlation between ENSO and PC1 of the $\delta^{18}O_{cel}$ network with three different reconstructions (Li et al., 2011; Li et al., 2013; Dätwyler et al., 2019) and for two different time periods (1750-1849, 1850-1949), which is shown in Supp. 4. Despite the fact that the sample density of the isotope network is relatively high in these two periods, the correlation between the PC1 and the ENSO reconstructions is weaker and not significant for the period 1750 to 1849. But not only the correlation gets weaker, also the correlation between a set of different ENSO reconstructions is getting weaker in the 18[th] century which was shown for specific periods in Dätwyler et al. (2019). Furthermore, Dätwyler et al. (2019) have found a consistent teleconnection pattern during the 18[th] century, which is different to the known teleconnection pattern of ENSO in the instrumental period. Moreover, the 1850s mark the end of Little Ice Age (LIA) period, when the atmospheric circulation over Europe (Felis et al., 2018) and its teleconnections to ENSO changed significantly (Rimbu et al., 2003). In their study, Felis et al. (2018) found evidence of an abrupt reorganization of the atmospheric circulation over Europe at the end of the LIA, transitioning from predominantly negative phases of the NAO (weakening of westerly winds) to predominantly westerly flow patterns over central Europe. Furthermore, modelling studies (e.g., Henke et al., 2017) have also reported an increased frequency of El Niño during LIA due to southern displacement of the Intertropical Convergence Zone.

A change of the ENSO characteristics would also have an influence on the teleconnection with the European climate. For instance, Rimbu et al. (2003) have investigated coral time series from the northern Red Sea, and identified a nonstationary relationship between the tropical Pacific and the European–Middle Eastern climate during the pre-instrumental period (see also Supp. 5). They have showed that from the mid-1930s to late 1960s, a PNA-like pattern in its negative phase, which is compatible with La Niña conditions, is associated with positive $\delta^{18}O$ anomalies in the Red Sea coral record. After 1970, they have detected a shift in the teleconnections which leads to the fact that positive anomalies in the Red Sea coral $\delta^{18}O$ are related to El Niño conditions. An unstable relationship between ENSO variability and the climate of Europe is also found in studies based on instrumental data (e.g., Fraedrich and Müller, 1992; Fraedrich, 1994; Pozo-Vázquez et al., 2005) or ocean-atmosphere coupled models (e.g., Raible et al., 2004; Deser et al., 2006; Brönnimann, 2007). The temporally unstable relationship between climate variables and ENSO is not only restricted to Europe, but also present in other regions of the planet (e.g., Álvarez et al., 2015).

Weak or inconclusive correlations between PC1 and ENSO reconstructions could also arise from the fact that the quality of the ENSO reconstructions decreases which could be based on a too low number of samples (especially in first years of the reconstruction period) and a non-stationarity of the used teleconnection (Batehup et al., 2015). Furthermore, ENSO reconstructions are mostly trained within the last 150 years and used for time periods characterized by an absence of instrumental data. Therefore, confident statements can only be made from 1850 onwards, since instrumental measurements of different climate variables are available. Thus, the only possibility to test and analyse the teleconnection before 1850 is by using ENSO reconstructions.

Another reason for the decrease in correlation could be the change of the climate signal of $\delta^{18}O_{cel}$. It is important to consider that the climate signal is directly coupled to the limiting factor for tree growth. It is possible that the limiting factor changes over time, which would result in different responses to climate. Esper et al. (2017) have shown that the climate signals in $\delta^{18}O_{cel}$ and $\delta^{13}C_{cel}$ change during warm and cold periods for trees in Switzerland, and they proposed to split the calibration between these two periods or to use corresponding transfer models. Future research is therefore required to investigate the climate signal during warm and cold periods, as well as the influence on our PCA results. However, we can only suggest that the relationship between ENSO and the European climate may not be stable over time.

## 4.4 The winter climate signal in $\delta^{18}O_{cel}$

The exact mechanism through which the $\delta^{18}O_{cel}$ captures a climate signal of the pre-seasons is still debated. For example, Heinrich et al. (2013) mentioned that winters with very low temperatures may damage the cambium more than usually requiring a longer recovery period. Such winters may have a negative effect on the cambial activity and on the photosynthesis process. Furthermore, Vaganov et al. (1999) have shown that precipitation during winter can sustainably affect tree growth in the following year. Their findings are similar to Treydte et al. (2006), who have shown that $\delta^{18}O_{cel}$ can contain a winter signal. Treydte et al. (2006) further argued that depending on the root system, winter snow fall, and the

characteristics of groundwater reservoirs, it is likely that trees use precipitation from the pre-seasons. Nevertheless, it is possible that winter climate conditions can also be memorized in $\delta^{18}O_{cel}$ through different climate feedback processes. For example, Ogi et al. (2003) highlighted that a positive NAO is frequently followed by higher pressures and warmer temperatures in Europe during the next summer. The authors suggested that SST, sea ice extensions and snow fall anomalies capture the winter climate conditions and influence the summer climate.

However, the oxygen isotope signal in cellulose depends primarily on the corresponding oxygen signal of the soil water and precipitation. According to Saurer et al. (2012), $\delta^{18}O_{SW}$ constitutes the average $\delta^{18}O$ input to the arboreal system over several precipitation events, and is modified by partial evaporation from the soil (depending on soil texture and porosity) and by a potential time lag, depending on rooting depth. Here we obtain the strongest correlations with $\delta^{18}O_P$ in winter, spring and summer (Fig. 7A, B, C). Because the correlations with $\delta^{18}O_{SW}$ are strongest in summer and autumn (Fig. 7G, H) and $\delta^{18}O_{SW}$ is the input of the arboreal system, we suggest that the isotopic signal of $\delta^{18}O_{cel}$ corresponds to an average over $\delta^{18}O_P$ events from winter, spring and summer, transferred via the $\delta^{18}O_{SW}$. It may also explain the reason behind the strong ENSO signal that $\delta^{18}O_{cel}$ is able to capture during winter. Moreover, it indicates the high potential of $\delta^{18}O_{cel}$ to capture climate signals even outside of the growing season.

When considering the multiseasonal signal, one has to take into account that this can lead to certain limitations in the analysis of $\delta^{18}O_{cel}$. This is because there is only one $\delta^{18}O_{cel}$ value in the network used for each site and for each year. However, this value does not represent a clear seasonal signal referring to one season, but a mix of signals referring to several seasons. Overall, the seasonal contributions to the climate signal can vary in strength. Due to the combinations of these influencing factors, the correlation to the individual seasons is certainly weakened, as no clear seasonal signal can be represented by the $\delta^{18}O_{cel}$ value. On the other hand, this enables the observation of several seasons and an understanding of which season has an influence on the biochemical processes in the tree.

**4.5 Links between PC2 and large-scale atmospheric modes**

In the composite maps for the PC2 (Fig. 9), we obtain a dipole structure between North Europe and the Mediterranean region. The dipole is characterized by a pressure anomaly centred on the North Sea, and which expands from the Northeast Atlantic to the Baltic Sea. Its counterpart is present in the northern Mediterranean region, especially in the area of Italy, northern parts of Greece and the Adriatic region.

The described pattern shows similarities to the summer European blocking pattern (Barnston & Livezey, 1997; Cassou et al., 2005) which is often associated with the Summer North Atlantic Oscillation (SNAO) (Hurrell & van Loon, 1997). According to Cassou et al. (2005), 17.8% of the positive phase and 17.9% of the negative phase of the summer European blocking pattern influence the total summer weather regimes in Europe.

The summer European blocking pattern is a surrogate indicator for storm track activities. During the positive index phase, the storm track moves farther northwards (Folland et al. 2009; Lehmann & Coumou 2015). This results in a low storm

activity over Northern Europe characterized by dry conditions, less cloudiness, high temperatures and a blocked cyclonic flow (Lehmann & Coumou 2015). On the other hand, the Mediterranean region is affected by lower temperatures and more

precipitation. The opposite phenomenon can be identified for the negative index phase. Northern Europe experiences an enhanced storm activity through the southward movement of storm track over Northwest Europe (Folland et al. 2009; Lehmann & Coumou 2015), which leads to higher precipitation, higher cloudiness and lower temperatures. Whereas, the northern Mediterranean experiences dry and warm conditions. These predominant summer European blocking pattern features are well represented in our composite maps of precipitation and temperature for PC2 (Fig. 8). Based on temporal

evolution of PC2, we suggest that there is a tendency towards a negative index phase starting at the beginning of the 20th century.

Beside the link to summer blocking activity in Europe, the geopotential height pattern is often used in another context. For example, Sillmann and Croci-Maspoli (2009) have shown that a positive geopotential height anomaly over the North Sea describes an atmospheric blocking-like pattern which relates to climate extremes like floods and droughts in the European

mid-latitudes. Moreover, this circulation anomaly pattern has been identified as the main driver for extreme dry periods over the Eastern Mediterranean (Oikonomou et al., 2010; Rimbu et al., 2014; Ionita and Nagavciuc, 2020), and for summer air temperature variability in Greece (Xoplaki et al., 2003a, b).

## 5 Conclusions

We present here a $\delta^{18}O_{cel}$ isotope network from tree rings for the last 400 years which was used to investigate the large-scale

climate teleconnections related to the European climate. According to our analysis, the climate signals of the network indicate that a link between the $\delta^{18}O$ variability and ENSO exists in winter, spring and summer. The investigation of the modelled $\delta^{18}O_{SW}/\delta^{18}O_P$ suggests that the summer signal still dominates $\delta^{18}O_{cel}$ but is partly influenced by lagged winter and spring precipitation signals. We argue that this is based on hydroclimatic feedback processes as well as characteristics of the water reservoirs of the different sample sites. The ENSO signal is detected for the last 130 years. However, no significant

links can be deduced during the period 1750 to 1850 which is indicating that the relationship between ENSO and the European climate could be not stable over time. The teleconnection changes between the tropical Pacific and Europe during the pre-instrumental period were also identified by coral data (Rimbu et al., 2003). Further knowledge about a change of teleconnections is essential because teleconnections have a remote climate impact on top of the current global warming. Our study shows that the EOF2 is characterized by a dipole pattern between North and Southeast Europe which is

comparable to the characteristics of the summer European blocking pattern. Since this mode is highly relevant for the summer climate conditions on the entire European continent, the temporal perspective gives new insights about how the frequency of this mode changed through time. Our findings suggest that there is a tendency towards a situation whereby Southeast Europe is predominantly characterized by a high-pressure system and North Europe by a low-pressure system

starting at the beginning of the 20th century. The described pressure pattern is relevant for the society because it can influence the spatial and frequency characteristics of climate extremes.

In the context of the ongoing discussion about the anthropogenic climate change, water isotope records can provide useful information about spatial and frequency changes of specific large-scale atmospheric circulation patterns. As a logical next step, more high-resolution paleoclimate data as well as comprehensive model simulations are required to provide additional insights into the stationarity of reconstructed European climate signals and their stationarity in teleconnections.

**Acknowledgements**

D.B. and D.C. are funded by the PalEX Project (AWI Strategy Fund) and M.I. is funded by the REKLIM project. All but four tree-ring stable isotope chronologies were established within the project ISONET supported by the European Union (EVK2-CT-2002-00147 'ISONET'). We want to thank all participants of the ISONET project (L. Andreu, Z. Bednarz, F. Berninger, T. Boettger, C. M. D'Alessandro, J. Esper, N. Etien, M. Filot, D. Frank, M. Grabner, M. T. Guillemin, E. Gutierrez, M. Haupt, E. Hilasvuori, H. Jungner, M. Kalela-Brundin, M. Krapiec, M. Leuenberger, H.H. Leuschner, N. J. Loader, V. Masson-Delmotte, A. Pazdur, S. Pawelczyk, M. Pierre, O. Planells, R. Pukiene, C. E. Reynolds-Henne, K. T. Rinne, A. Saracino, M. Saurer, E. Sonninen, M. Stievenard, V. R. Switsur, M. Szczepanek, E. Szychowska-Krapiec, L. Todaro, K. Treydte, J. S. Waterhouse, and M. Weigl). The data from Turkey, Slovenia and Southwest Germany were produced with the EU-funded project MILLENNIUM (GOCE 017008-2'MILLENNIUM'), special thanks to T. Levanic and R. Touchan. The tree-ring stable isotope chronologies from Bulgaria were established with support of the German Research Foundation DFG (HE3089-1, GR 1432/11-1) and in cooperation with the administration of Pirin National Park, Bulgaria. Additionally, we want to thank M. Butzin and M. Werner for providing the $\delta^{18}$O in precipitation and soil water from nudged ECHAM5-wiso simulations and two anonymous reviewers for their helpful comments.

**Data availability**

The ISONET network is not publicly available. The time series of the individual sample sites can be request by the corresponding authors of the mentioned studies in subsection 2.1. The used climate datasets are publicly available, please check the references and links in subsection 2.2 for more details.

**Author contributions**

DB undertook the research, writing and analysis. MI, MW, GH, GS, NR, MF, IH, DC and GL supported DB in writing, data analysis and research approaches. GH, GS and IH were involved in the ISONET project. GL and MI are the PhD supervisors of DB.

**Competing interests**

The authors declare that there is no conflict of interest.

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

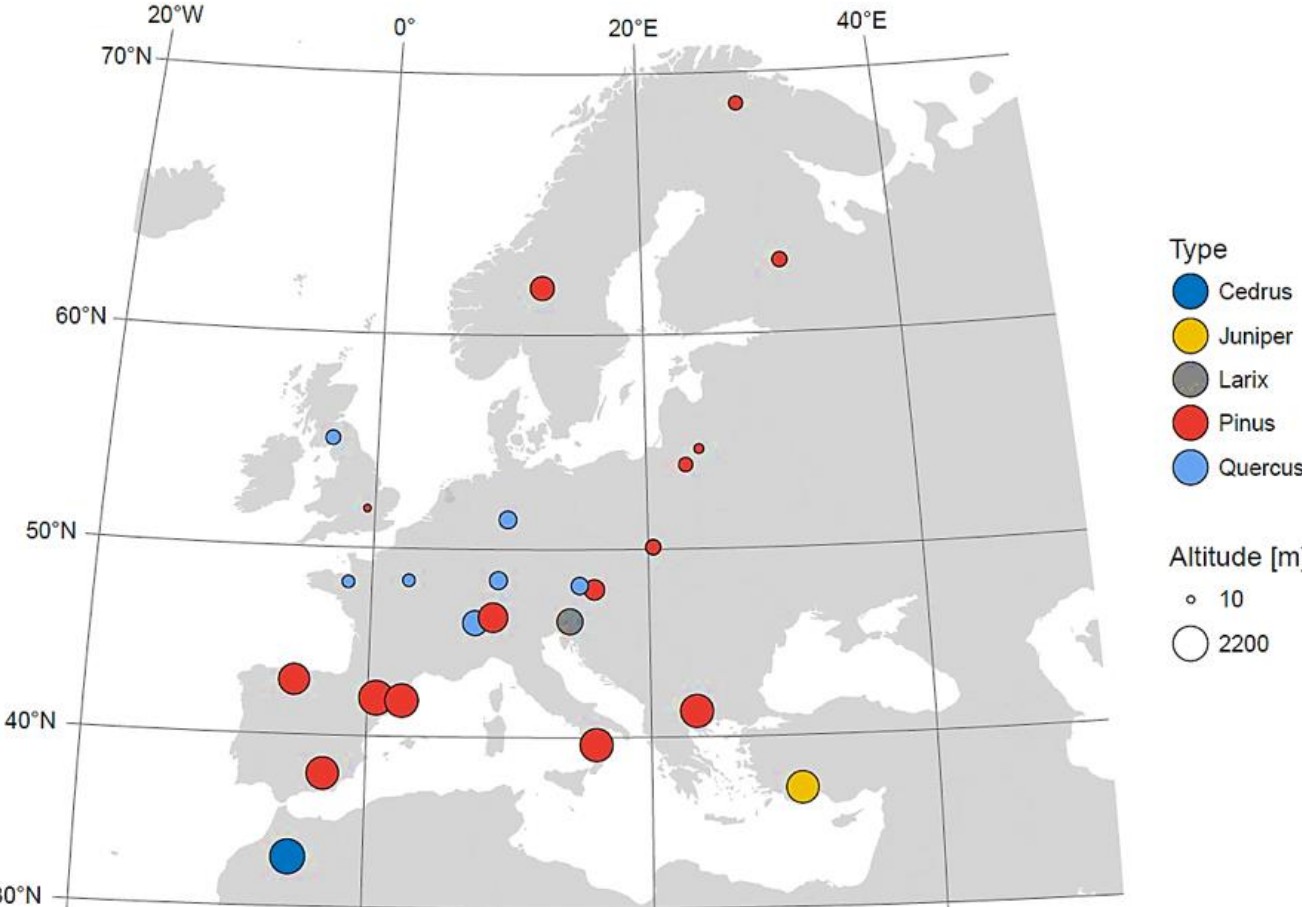

**Figure 1: Spatial distribution of sample sites combined with the corresponding altitude.** The highest density of sample sites exists in Central and Western Europe. The colour indicates the tree type (*Cedrus* (dark blue), *Juniper* (yellow), *Larix* (grey), *Pinus* (red) and *Quercus* (light blue)). The corresponding elevation (10-2200 m) is shown by the size of the circles.

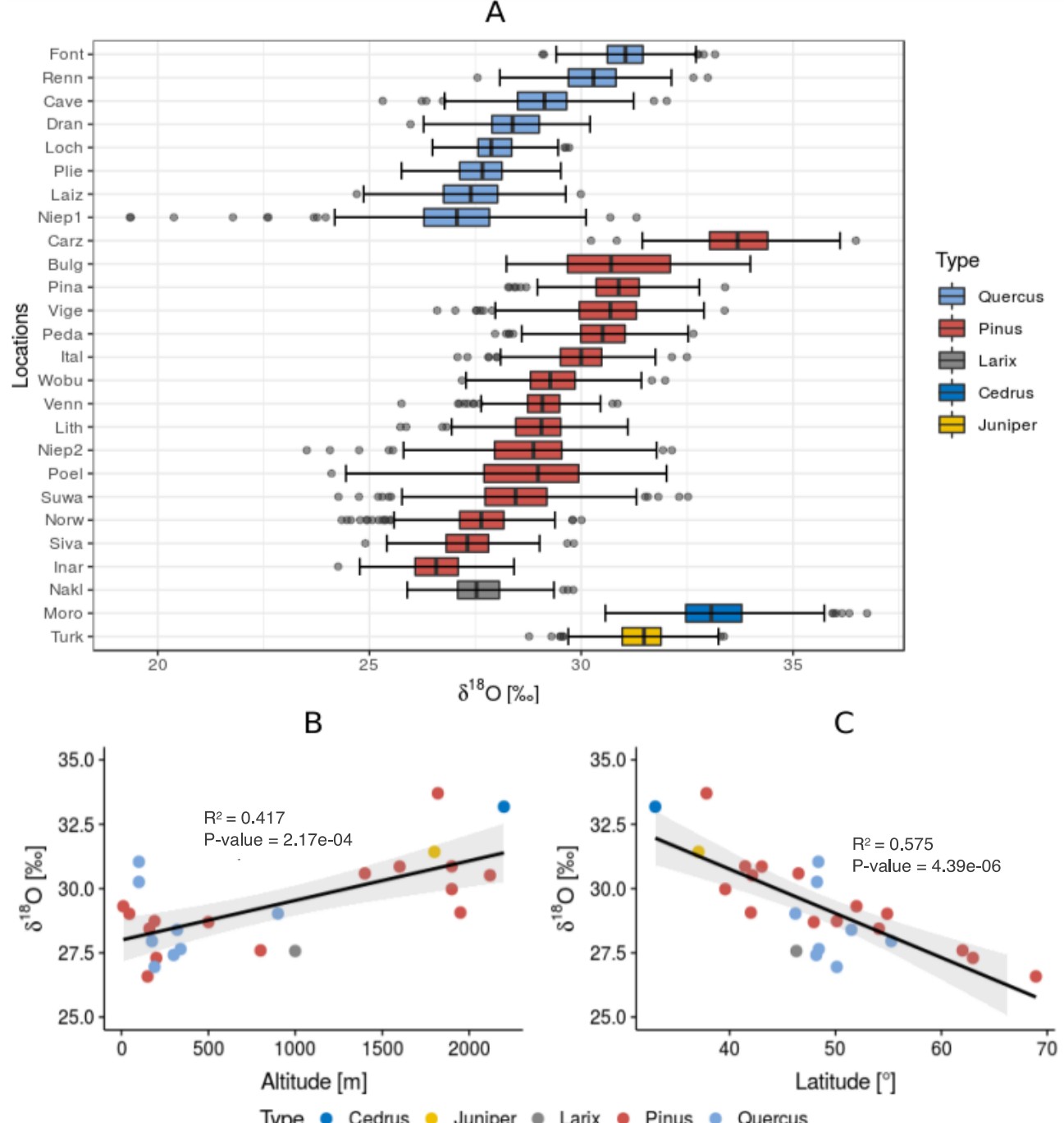

**Figure 2: The characteristics of the European δ¹⁸O time series/network. A,** describes each time series with boxplots which were firstly ordered after the tree type and secondly after the average value. Additionally, the relation between the average value of each time series is plotted against their latitudinal (**B**) and their altitudinal position (**C**).

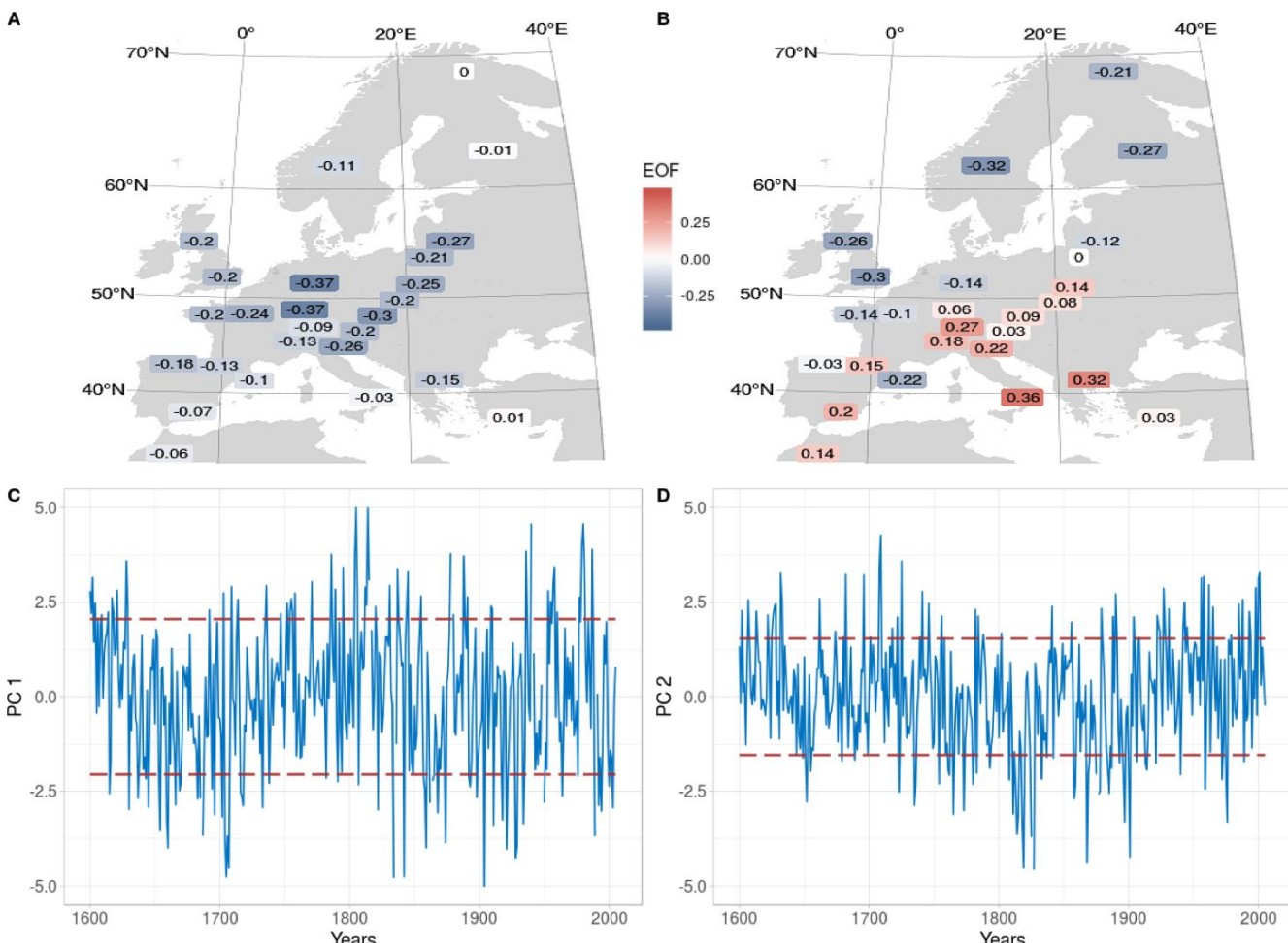

**Figure 3: Spatial and temporal variability of the first two δ¹⁸O components and EOFs. A**, EOF for the first δ¹⁸O component (16.2% explained variance), **B**, EOF for the second δ¹⁸O component (9.5% explained variance). **C** and **D** are the time series for the first and second δ¹⁸O component. The dashed red lines indicate the standard deviation for the years 1600-2005.

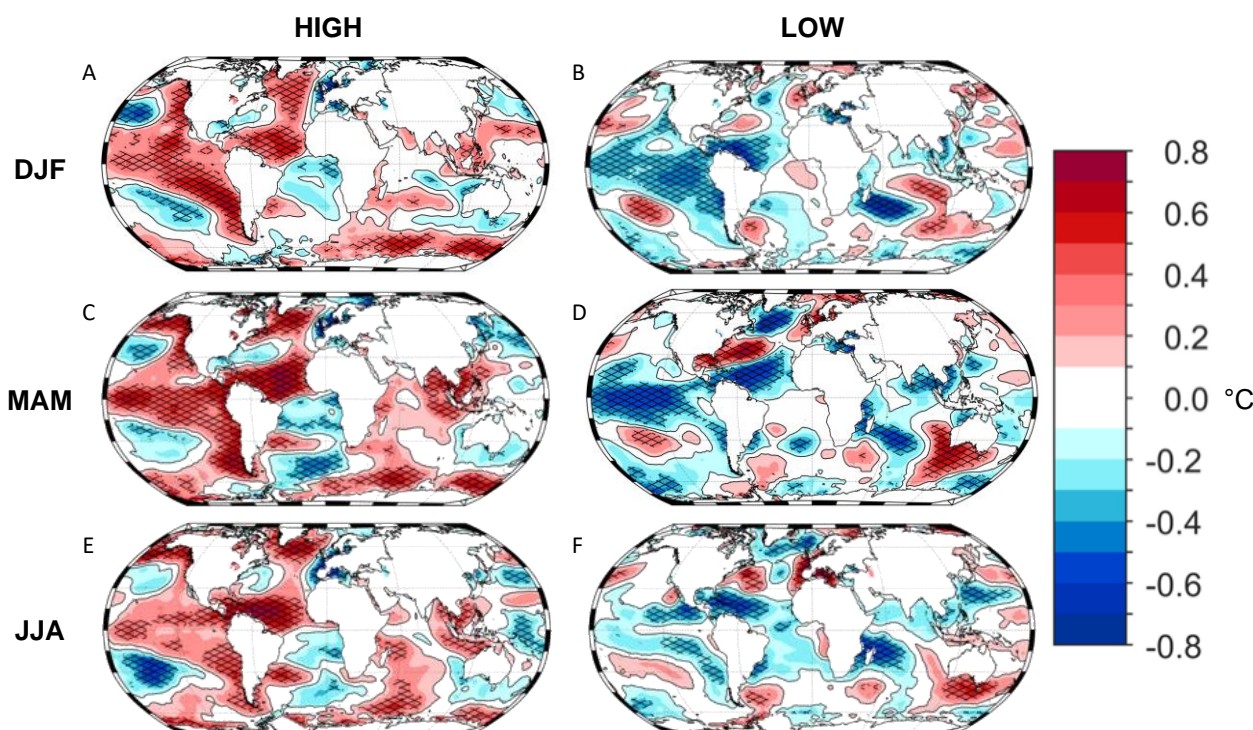

**Figure 4: Composite maps (high and low) for SST related to the first δ¹⁸O component for the seasons DJF, MAM and JJA.** The first row shows the characteristics of the climate in DJF, the second in MAM and the third row in JJA, whereas the first column shows the results for maxima and the second for minima events of PC1. The SSTs in winter, spring and summer are characterized by ENSO activity. Furthermore, the significance is shown with a black grid in front of the colour. The SST dataset from ERSST (Huang et al., 2017) are included in this figure for the period 1854 to 2005.

**HIGH**          **LOW**

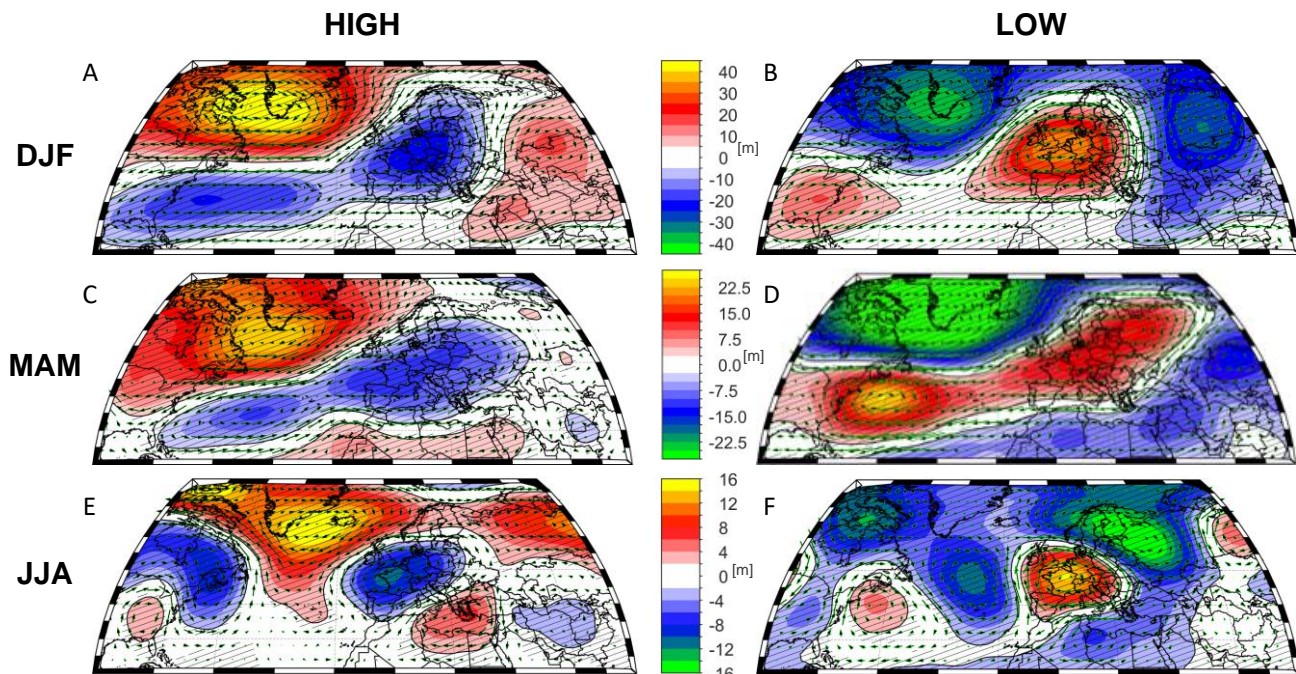

**Figure 5: Composite maps (high and low) for Z500 related to the first δ¹⁸O component for the seasons DJF, MAM and JJA.** The first row shows the characteristics of the climate in DJF, the second in MAM and the third row in JJA, whereas the first column shows the results for maxima and the second for minima events of PC1.The Z500 maps show similar characteristics in winter and spring, whereas a pressure regime is directly located over Central Europe in summer. The Z500 dataset from 20CRv2c (Compo et al., 2011) are used in this figure for the period 1901 to 2005. Furthermore, the significance is shown with a black contour line.

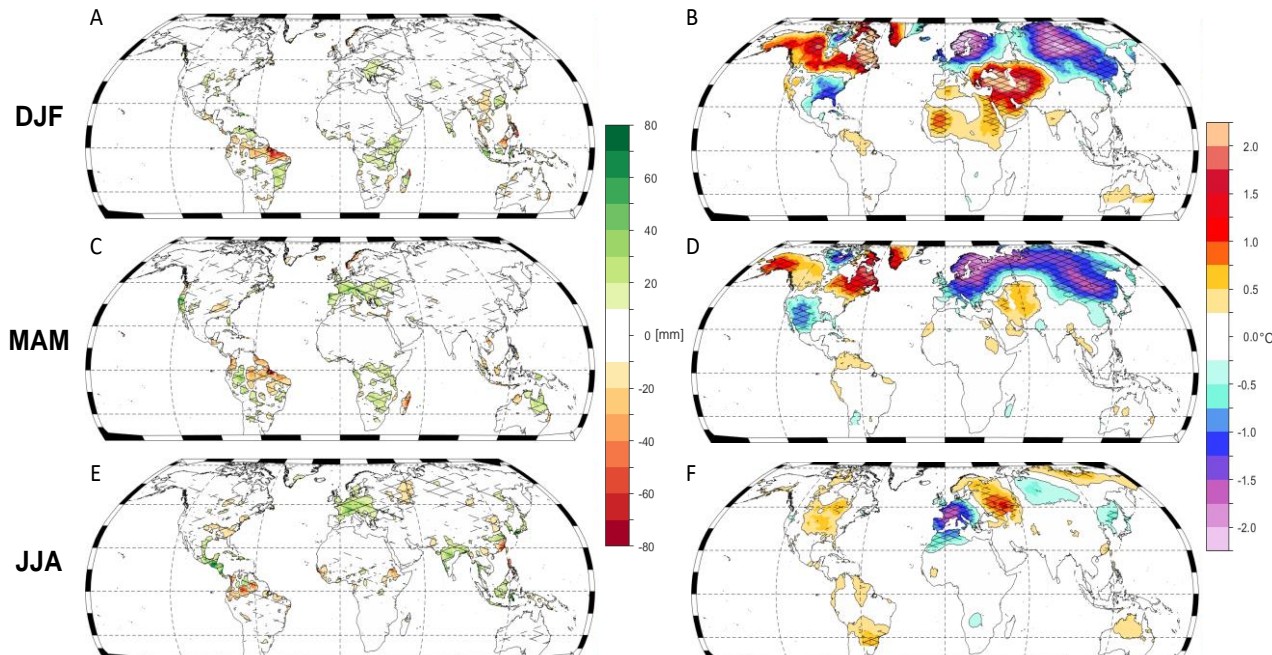

**Figure 6: Composite maps (high-low) for precipitation and air temperature related to the first δ¹⁸O component for the seasons DJF, MAM and JJA.** The first row shows the characteristics of the climate in DJF, the second in MAM and the third row in JJA, whereas the first column shows the results for precipitation and the second for air temperature. The precipitation and air temperature dataset from CRU TS are included in this Figure for the period 1901 to 2005. The significance is shown with a black grid in front of the colour.

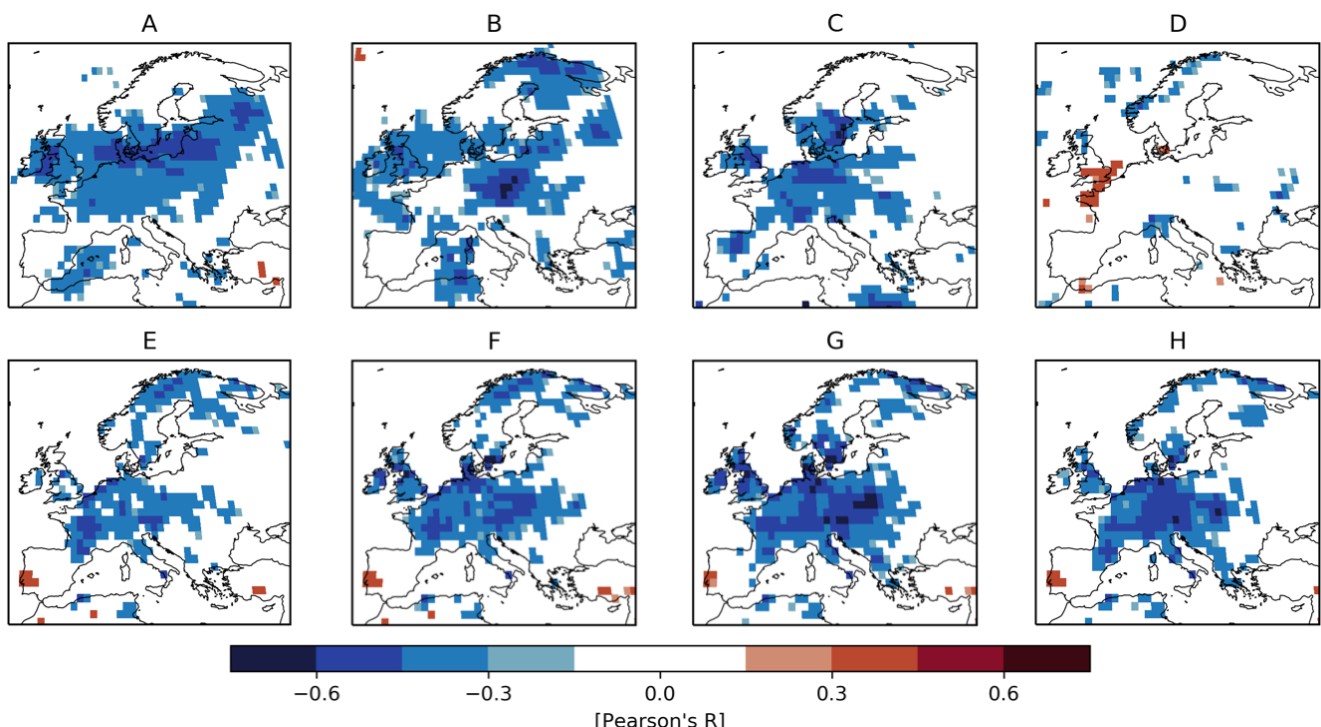

**Figure 7: Links between the first δ$^{18}$O component and the modelled δ$^{18}$O in soil water and precipitation from nudged climate simulations with ECHAM5-wiso (Butzin et al., 2014).** The upper row is showing the correlation between the first δ$^{18}$O component (PC1) and δ$^{18}$O in precipitation for winter (A), spring (B), summer (C) and autumn (D). Panels E, F, G, H are the correlation maps for PC1 and δ$^{18}$O in soil water for winter, spring, summer and autumn. In all maps, the significant grid cells are coloured.

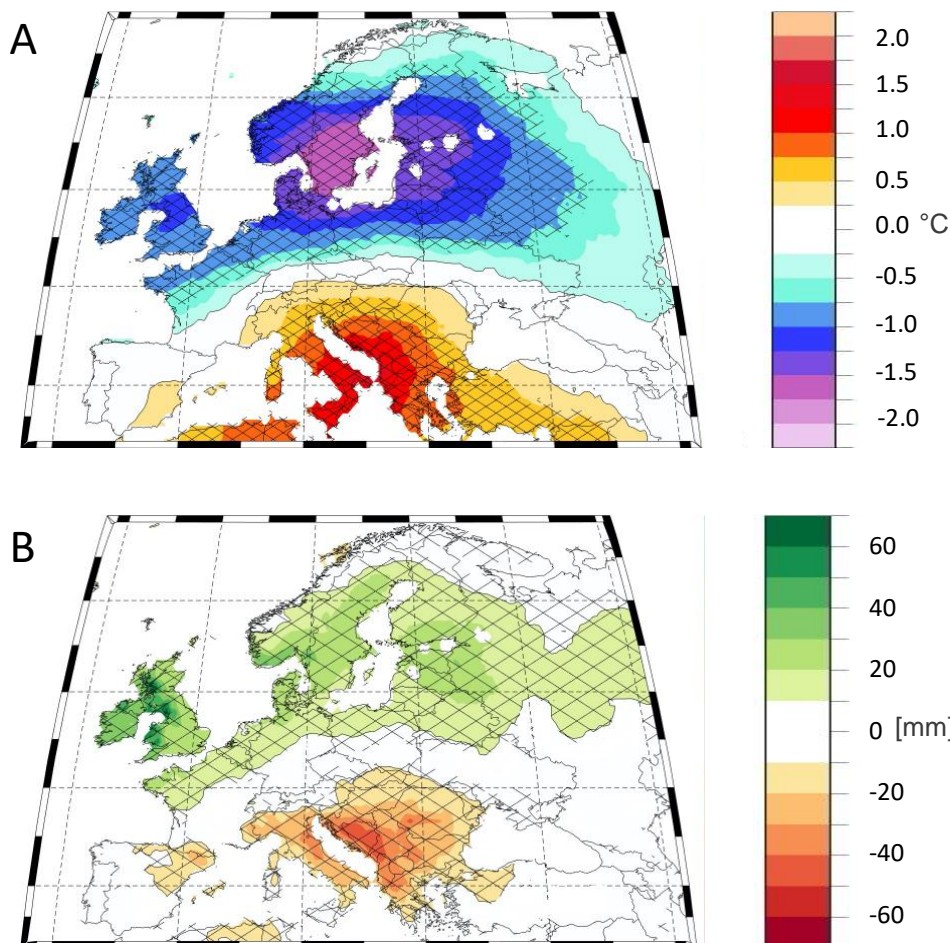

**Figure 8: Composite maps (high-low) for the boreal summer related to the second δ¹⁸O component. A,** surface temperature JJA, **B,** precipitation JJA, **C,** Z500 JJA. The datasets are the same as in Figure 6. Furthermore, the significance is shown with a black grid in front of the colour.

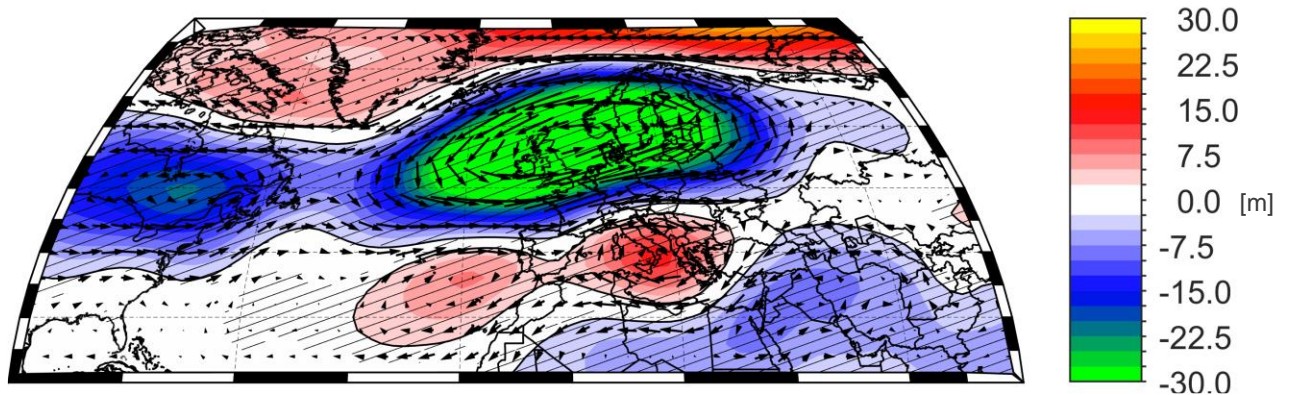

**Figure 9: Composite map (high-low) for Z500 related to the second δ$^{18}$O component (PC2) for the boreal summer.** The Z500 dataset
from 20CRv2c (Compo et al., 2011) are used in this figure for the period 1901 to 2005. Furthermore, the significance is shown with a
black contour line.