# Peer review of "Large-scale climate signals of a European oxygen isotope network from tree-rings"

_Climate of the Past, 2020_

## Referee Comment (RC1) · Anonymous Referee #1 · 5 Aug 2020

By analysing a European network of 26 tree-ring sites with $\delta$18O measurements, the authors aim at extracting regional climate signal imprinted in the records to investigate the dominant modes of variability of the European climate and their relationships with the large-scale atmospheric circulation, in particular ENSO. Their findings suggest that climate variability in Europe is strongly modulated by ENSO teleconnections at least over the past 130 years, but that some differences arises between the northern and southern regions.

Although the results are promising, I do not think the manuscript is ready for publication yet. A restructuration and reorganisation of the paper is strongly needed. While the

introduction is relatively well written and easy to follow, many confusions arise from the Material and Methods section and some clarifications are required to allow the readers to easily understand why and how the proposed analyses were made. The division of the 'Results and Discussion' section into two separate sections should improve the readability of the manuscript. It seems that the authors have not carefully re-read their manuscript to check for typos and ensure that the text is fully understandable before submitting it. The authors also should make an effort to properly, clearly and thoroughly discuss their results and their implications for the understanding of the atmospheric teleconnections. So far only in the Summary and Conclusion section are the results clearly highlighted and interpreted.

Some additional comments and suggestions:

L20: 'may not be stable. . .'

L42-43: Actually, it is the other way around: $\delta$18Ocel depends on $\delta$18OSW but $\delta$18OSW itself does not depend on $\delta$18Ocel. Please rewrite.

L55-56: You could also cite more papers showing the potential of $\delta$18Ocel for reconstructing large-scale patterns of climate variability (since it is one aim of your study), e.g.: Brienen, R. J. W., Helle, G., Pons, T. L., Guyot, J.-L., Gloor, M., Oxygen isotopes in tree rings are a good proxy for Amazon precipitation and El Niño-Southern Oscillation variability, PNAS, (42) 16957-16962; DOI: 10.1073/pnas.1205977109, 2012 Lavergne, A., Daux, V., Villalba, R., Pierre, M., Stievenard, M., Srur, A. M., Vimeux, F., Are the $\delta$18O of F. cupressoides and N. pumilio promising proxies for climate reconstructions in northern Patagonia?, J. Geophys. Res.-Biogeo., 121, 767–776, https://doi.org/10.1002/2015JG003260, 2016.

L76-77: I am not sure what is the meaning of this sentence. Please rewrite. L75-80: I would suggest clearly rewriting this part as it is difficult to read. You should get right to the point: how are you going to achieve your goals? What are the main analyses you are going to perform to reach those goals?

L94-96: Please comment on the implication of only using latewood for oak but both early- and late- woods for the coniferous species. Are you suggesting that earlywood in the coniferous species is only derived from carbohydrates formed during the current year? Please rewrite the sentence accordingly.

L100-101: "Here" is repeated twice in the sentence. Furthermore, the sentence is not grammatically correct. Please be more careful!

L108-109: what is the COBE-SST2 dataset? Please describe it here. Also, which index of ENSO are you using to define El Niño/La Niña years?

L114-116: What is a 'nudge model scenarios/simulation'? It is not clear why you choose this title for the section. I would recommend combining sections 2.2 and 2.3 instead. How using both $\delta18OP$ and $\delta18OSW$ will inform you about 'fractionation/ photosynthesis processes'? You will never get insights into the fractionation processes occurring during photosynthesis using only those two timeseries! Please clarify.

L123-131: What is the difference between EOF and PCA? From my understanding of those analyses, EOF and PCA are really similar. Are you suggesting that EOF provides information about spatial patterns, while PCA gives information about temporal patterns? The whole paragraph is confusing (especially the filtering actually done to fulfil the North et al. (1982) rule), please rewrite.

L132-133: Why are you mentioning this here? It should be already stated in Section 2.1.

L133-140: How can you be sure that by using the gap fill method, you will not influence your results? Also, why would you need to fill in the gaps for 400 years knowing that your climate data only goes up to 1851?

L141-146: What do you mean? The whole paragraph is pretty confusing and after reading it several times, I still do not get what you are really doing here. What kind of information is providing the geopotential height 500mb (Z500) for the analysis?

[Figure]

L151: 'Nino 3.4 index' this should come earlier in section 2.2 when you are presenting the environmental data used in the analyses.

L158-160: The first sentence does not provide any clear information. Please remove.

Figure 1: Could you add more information in the figure A related to the latitude of each site? The names and characteristics of the sites are not presented anywhere in the manuscript. Even though the data have already been published, a Table with sites information should be included. In figures B and C, how is it possible that R2 and p-value are exactly the same for the relationships between $\delta$18Ocel and altitude and between $\delta$18Ocel and latitude? I suspect that there is a mistake here.

L160-161 and L172-173: Since no information about the exact location of the site presented in Figure 1A is provided, it is difficult to follow this statement.

L162: I would remove 'This might be determined genetically,' from the sentence as it is not completely accurate (different species of Quercus also have different genetic information).

L165: You could also cite more updated papers describing differences between angiosperms and gymnosperms, e.g.: Carnicer J., Barbeta A., Sperlich D., Coll M., Penuelas J., Contrasting trait syndromes in angiosperms and conifers are associated with different responses of tree growth to temperature on a large scale, Front. Plant Sci., 4, 409, https://doi.org/10.3389/fpls.2013.00409, 2013

L177: 'which could influence the relation by a latitudinal effect.' please rewrite as 'thus the latitudinal and altitudinal gradients may have confounding effects on $\delta$18Ocel' or something similar

L177-179: I would rewrite this sentence as the effects of the two gradients on $\delta$18Ocel have already been observed and documented in many other studies, for instance: Szejner, P., W. E. Wright, F. Babst, S. Belmecheri, V. Trouet, S. W. Leavitt, J. R. Ehleringer, R. K. Monson, Latitudinal gradients in tree ring stable carbon and oxygen isotopes

reveal differential climate influences of the North American Monsoon System, J. Geophys. Res. Biogeosci., 121, 1978–1991, doi:10.1002/2016JG003460. 2016

L181-186: Here again comes the confusion between EOF and PCA. You should clarify from the beginning (see previous comment) what is the difference between the two especially given that EOF1 and PC1 both seem to explain 16.2% of the variance in the records.

Figure 4: In the legend, you are describing the columns not the rows.

L216 Is the distribution of PC1 for El Niño (or La Niña) years significantly different from that during normal years (i.e. when excluding El Niño/La Niña years)?

L222-224: Please rewrite the sentence. As it reads now, it looks like you are saying that Europe is characterized by higher precipitation and lower air surface temperatures in summer! And it is not clear what the parentheses apply for.

L230-231: 'because we to take into account. . .' Why would SPEI3 index accounting for the climate conditions prevailing over the previous season? So far, nothing has been said about this dataset.

L227-233: this part mostly belongs to the Material and Methods section and could be improved for readability.

L235: 'the used reconstruction': which one?

L240: 'to capture a multi-seasonal signal' what do you mean?

L243-244: where is it shown?

L244-245: so why then $\delta18O_{cel}$ is not more strongly related to $\delta18O_{sw}$? Your argument is contradictory with what is actually described.

L239-248: And so what? What are you really trying to say here? Also, I do not think the results are properly discussed and compared to the literature.

Figure 6: You mean the upper row Why description of Figure 6 comes before Figure 5?

L168-169 and L276: Please rewrite sentences

L236-237: The instability of the relationship between climate variables and ENSO has also been documented by other tree-ring studies in southern South America, e.g: Álvarez, C., Veblen, T.T., Christie, D.A., González-Reyes, Á., Relationships between climate variability and radial growth of Nothofagus pumilio near altitudinal treeline in the Andes of northern Patagonia, Chile. For. Ecol. Manage. 342, 112–121, 2015

---

## Referee Comment (RC2) · Anonymous Referee #2 · 18 Aug 2020

Review of 'Large scale climate signals of a European oxygen isotope network from tree-rings - predominantly caused by ENSO teleconnections¿ by Balting et al.

Summary

The authors investigate delta18O tree ring records of 26 site distributed over Europe for the last 400 years. They claim that they were able to identify a connection of the leading mode of variability of this data to El Nino Southern Oscillation. They speculate that this connection is only found in the last 130 years and thus the connection is not stable in time. The second mode of the data is suggested to reconstruct regional summer atmospheric circulation. Finally, the author team claim that delta18O tree ring records

2" />
can be used to reconstruct atmospheric circulation.

General comments

The topic would certainly be of high interests. However, the authors fail to convincingly show evidence for their main claims listed in the summary. Therefore, I recommend to reject the manuscript.

Major comments

1. At several places in the manuscript the authors claim that their analysis suggests "the relationship between ENSO and the European climate may not stable over time". The connection is only found for the instrumental period after 1850 CE. I think the first order interpretation is that the reconstruction of ENSO might be not perfect, as normally the reconstruction methods are trained in the last 100 to 150 years. So differences between the training periods and the period before are a hint that the reconstruction might be not successful. So, from your analysis you cannot conclude that you have identified non stationarity of teleconnections.

2. The conclusion that the analysis shows that "We infer that the investigation of large-scale atmospheric circulation patterns and related teleconnections far beyond instrumental records can be done with oxygen isotopic signature derived from tree rings. " is not convincingly demonstrated. There is only 2 line in the introduction which gives a hint why this should be possible, i.e., fractionation happens during the transport from source to sink areas. Most of the studies however try to reconstruct temperature and precipitation when using delta18O as delta18O is first order temperature dependent. The authors also nicely discuss that fractionation processes are also relevant within the tree. Then, at the different sites the water can be transported form different source regions during the seasonal cycle, e.g., North Atlantic versus Mediterranean, or long distance transport versus local water recycling. Moreover, seasonality plays an important role, so mostly tree ring records are interpreted to record growing season signals and not winter signals. So given all these uncertainties how can the transport aspect

(which is related to the atmospheric circulation) survive?

3. For the first EOF I have a different interpretation, which takes into account the fact that temperature play the dominant role in delta18O. What we see is a monopole structure. The authors claim to see a link to ENSO. I hypothesize that the link is simply due to the fact that ENSO has a global impact on the global mean temperature. So, due to an El Nino event, the Earth warms and thus also the North Atlantic and the Mediterranean (visible in the composite plots). Warmer source regions affect the fractionation of delta18O without any change of the circulation we see in the sink regions (at the tree sites) a uniform signal.

4. There are problems with the data (see comment below Section 2.2, L132-140, L145) ignored which might be influential to the analysis

Minor comments

L18: What is meant by "reflects a multi-seasonal climatic signal."? ENSO works on timescales of 3 to 7 years.

L20: "out of phase variability": I would interpret this in the time and not in space as the authors. Just say the 2. EOF is a dipole pattern with centers over northwestern and southeastern Europe.

L47-50: Hard to read.

L53: please change to "leaf water clearly affects"

L55: I disagree with this statement, see major comments 2 and 3.

L56: What is meant by "resulting long-term perspective"? Where does it result from?

L84: Created -> generated

L88: Please include a space between number and unit throughout the manuscript.

L91: What is SMOW?

Section 2.1: Which method is used to get the delta18O samples from trees. This is relevant as studies show that the method (pooling or not pooling) makes a huge difference

Hangartner et al. Methods to merge overlapping tree-ring isotope series to generate multi-centennial chronologies CHEMICAL GEOLOGY Volume: ‏ 294 Pages: ‏ 127-134 Published: ‏ FEB 10 2012

Section 2.2: Again if is unclear what the authors are using. Is it the ensemble mean of 20CR or an individual ensemble member. Please note that the 20CR is only constraint with sea level pressure (SLP) data so no sea surface temperatures (SST) are used which are relevant for ENSO. My guess is that the authors use the ensemble mean. This is problematic as in the early part of the reanalysis the constraint (via SLP) is rather weak leading variance deflation and thus can have a strong impact on the analysis (so it is normally recommended to use all individual ensemble members). As said, the other problem is that ENSO might not be realistically included in the first part of the reanalysis.

Section 2.3: too short and not clear why the simulations are used and how the simulations are generated. The reader needs to understand which model is used and how, just references is not enough.

Section 2.4: It reads like EOF and PCA are different analysis, but actually they are not. The method of empirical orthogonal function (EOF) analysis is a decomposition of a data set in terms of orthogonal basis functions determined from this data. Thus it is the same as geographically weighted PCAs.

L132-140: How many tree ring records cover the entire period with no gaps? How sensitive it the analysis to filling the gaps? How many cycles are needed to reach convergence? What if you use only the tree ring records which cover the entire period?

L142: This is certainly not extreme.

L145: I would say that the authors misuse the composite analysis by focusing on the linear response. If they would like to analyze the linear response, a simple correlation analysis is enough. The beauty of the composite analysis is that you can easily show non-linear effects, but only if you make the difference between the mean above the threshold with the long term mean and in a second plot the mean below the negative threshold and the long term mean. This was done by Fraedrich 1992 mentioned in the manuscript. He highlighted the nonlinearity of the ENSO response over Europe and thus is in contradiction to the linear relationship suggested here.

L150: Event Coincidence Analysis needs to be explained.

L177-179: If this is the case one could speculate that EOF2 showing a North South patterns just resemble this latitudinal effect.

L195: I do not see this is there a typo and the authors mean PC1?

L200-210: Avoid using the bracket with e.g. (cold). This makes the text unreadable. Just say what you show in Fig. 4.

L208: I do not see a AO pattern, again the reference to Fraedrich are incorrect as they claim that ENSO has a nonlinear response behavior over Europe.

L225: Why drought we see a positive precipitation anomaly? Section3.4: What do we learn from this? What is shown and why? This section is unclear and to my feeling can be removed.

L250: section 3.5

L251-252: This sentence is a repetition.

L260 -263: You need to show this with more proxies. Note that dry conditions are not droughts!

L263-265: Given your study you cannot conclude this. The authors study certainly is inadequate to reconstruct blocking so this statement is not supported by the authors

analysis.

L269-70: Please change to "... signal still dominates".

Figures: Statistical significance is not tested (or not shown).
* * *

---

## Author Comment (AC1) · 22 Sep 2020

R: We thank the anonymous referee #1 for the time and effort in reviewing our manuscript (cp-2020-39). The comments, suggestions and feedback raised in the review are highly appreciated as they help us to clarify our statements and to improve the quality of our manuscript. Below you will find a point by point response (*reviewer*, response).

⋯⋯⋯⋯⋯⋯⋯⋯⋯⋯⋯⋯⋯⋯⋯⋯⋯⋯⋯⋯⋯⋯⋯⋯⋯⋯

*By analysing a European network of 26 tree-ring sites with δ18O measurements, the authors aim at extracting regional climate signal imprinted in the records to investigate the dominant modes of variability of the European climate and their relationships with the large-scale atmospheric circulation, in particular ENSO. Their findings suggest that climate variability in Europe is strongly modulated by ENSO teleconnections at least over the past 130 years, but that some differences arises between the northern and southern regions.*

*Although the results are promising, I do not think the manuscript is ready for publication yet. A restructuration and reorganisation of the paper is strongly needed. While the introduction is relatively well written and easy to follow, many confusions arise from the Material and Methods section and some clarifications are required to allow the readers to easily understand why and how the proposed analyses were made. The division of the 'Results and Discussion' section into two separate sections should improve the readability of the manuscript. It seems that the authors have not carefully re-read their manuscript to check for typos and ensure that the text is fully understandable before submitting it. The authors also should make an effort to properly, clearly and thoroughly discuss their results and their implications for the understanding of the atmospheric teleconnections. So far only in the Summary and Conclusion section are the results clearly highlighted and interpreted.*

R: We are glad that the results are promising, and we agree with the reviewer that structural changes are necessary. We agree that the "Material and Methods" section has potential for improvement. Therefore, the "Material and Methods" section will be extended by a more detailed description of the isotope measurements (e.g. the used measurement technique, sample site). Furthermore, we will extend our description of the used climate data including the used ENSO indices and the SST dataset, and we add more argumentation why we used the specific datasets (e.g. the usage of the ensemble mean of NOAA-CIRES Twentieth Century Reanalysis (V2c) (Compo et al., 2011)). Moreover, the uncertainties were highlighted more prominently in the method section as well as in the discussion. We agree that we should

extend our discussion and separated it from the results chapter. The interpretation of the results will be worked out in more detail.

*Some additional comments and suggestions:*
*L20: 'may not be stable. . .'*

R: We will add the missing word.

*L42-43: Actually, it is the other way around: δ18Ocel depends on δ18OSW but δ18OSW itself does not depend on δ18Ocel. Please rewrite.*

R: Thank you for the comment! We will change it in the revised version of the manuscript.

*L55-56: You could also cite more papers showing the potential of δ18Ocel for reconstructing large-scale patterns of climate variability (since it is one aim of your study), e.g.: Brienen, R. J. W., Helle, G., Pons, T. L., Guyot, J.-L., Gloor, M., Oxygen isotopes in tree rings are a good proxy for Amazon precipitation and El Niño-Southern Oscillation variability, PNAS, (42) 16957-16962; DOI: 10.1073/pnas.1205977109, 2012 Lavergne, A., Daux, V., Villalba, R., Pierre, M., Stievenard, M., Srur, A. M., Vimeux, F., Are the δ18O of F. cupressoides and N. pumilio promising proxies for climate reconstructions in northern Patagonia?, J. Geophys. Res.-Biogeo., 121, 767–776, https://doi.org/10.1002/2015JG003260, 2016.*

R: We will integrate the suggested references and further references, which are using $\delta^{18}O_{cel}$ for the reconstruction of large-scale climate patterns, in the revised version of the manuscript.

*L76-77: I am not sure what is the meaning of this sentence. Please rewrite. L75-80: I would suggest clearly rewriting this part as it is difficult to read. You should get right to the point: how are you going to achieve your goals? What are the main analyses you are going to perform to reach those goals?*

R: We will rewrite the last paragraph of the introduction according to the suggestions of the reviewer. Especially, we will shortly mention the analysis technique which is used to achieve the mentioned goals.

*L94-96: Please comment on the implication of only using latewood for oak but both early- and late- woods for the coniferous species. Are you suggesting that earlywood in the coniferous species is only derived from carbohydrates formed during the current year? Please rewrite the sentence accordingly.*

R: Basically, trees form their annual ring from current assimilates and reserves (starch and fats). The used carbohydrates come from a pool which, depending on the season and the rate of assimilation during the vegetation period, is in part clearly dominated by reserves from the previous year(s). Since the conifers of the isotope network are evergreen, the proportion of the reserves used for wood accumulation decreases rapidly at the beginning of the earlywood formation and is usually not as high at the beginning as in the case of the deciduous oak trees utilized here. Therefore, we don't suggest that earlywood is only derived from carbohydrates formed during the current year for coniferous. However, oak is a ring-porous tree species known for having its earlywood growth almost completing before the leaves are fully green and net photosynthesis is positive. Hence, for this species it makes sense to skip earlywood as it is rather easy to distinguish from latewood. On the other hand, it is difficult to identify a clear boundary between the early- and latewood of conifers without technical means of quantitative wood anatomy. We will rewrite the sentence and add the mentioned short explanation.

*L100-101: "Here" is repeated twice in the sentence. Furthermore, the sentence is not grammatically correct. Please be more careful!*

R: Thank you for the comment. We will correct it in the revised version of the manuscript.

*L108-109: what is the COBE-SST2 dataset? Please describe it here. Also, which index of ENSO are you using to define El Niño/La Niña years?*

R: The COBE-SST2 dataset was created by Hirahara et al. (2014) and it is based on in situ sea surface temperature (SST) and sea ice concentration (SIC) observations. The monthly SST fields are provided by NOAA/OAR/ESRL PSL, Boulder, Colorado, USA and can be downloaded from https://psl.noaa.gov/data/gridded/data.cobe2.html. For detailed information about the used methods and the uncertainties, we refer to the publication of Hirahara et al. (2014). Furthermore, we used the anomaly (1981-2010 mean removed) of the December Nino 3.4 index (HadISST1; Rayner et al., 2003). The index represents the averaged SST from 5°S-5°N and 170°-120°W and can be downloaded from https://psl.noaa.gov/gcos_wgsp/Timeseries/Nino34/.

We agree with the reviewer that the used SST dataset should be explained more in detail and also the ENSO index shall be presented in this subchapter. These points will be changed in the revised version of the manuscript.

*L114-116: What is a 'nudge model scenarios/simulation'? It is not clear why you choose this title for the section. I would recommend combining sections 2.2 and 2.3 instead. How using both*

*δ18OP and δ18OSW will inform you about 'fractionation/ photosynthesis processes'? You will never get insights into the fractionation processes occurring during photosynthesis using only those two timeseries! Please clarify.*

R: We agree with the reviewer that more information is required to understand why we used the simulations. In general, the name of the chapter is given based on the method which produces $\delta^{18}O_P$ and $\delta^{18}O_{SW}$ (in our case the nudged model simulations from Butzin et al. (2014)). Our goal with the model output is to test the correlation between $\delta^{18}O_{cel}$ and modelled $\delta^{18}O_P/\delta^{18}O_{SW}$ to identify if the water, which is used within the photosynthesis processes, has a multi-seasonal isotopic signature. If yes (as shown in our plots), it is an explanation how $\delta^{18}O_{cel}$ is able to capture the ENSO signal. Our results are of interest to the community because a significant ENSO influence of the European climate have been identified for winter (Fraedrich and Müller, 1992; Fraedrich, 1994; Pozo-Vazquez et al., 2005; Brönnimann et al., 2004; Brönnimann et al., 2007) and spring (Brönnimann et al., 2007; Lloyd-Hughes and Saunders, 2002; Helama et al., 2009). Since the captured climate information of the $\delta^{18}O_{cel}$ network mainly reflects the summer season, our correlation analysis suggests that $\delta^{18}O_{cel}$ is able to capture multi-seasonal climate signals through hydrological processes (soil moisture and soil water content) and therefore, can also contain a winter/spring climate signal.

*L123-131: What is the difference between EOF and PCA? From my understanding of those analyses, EOF and PCA are really similar. Are you suggesting that EOF provides information about spatial patterns, while PCA gives information about temporal patterns? The whole paragraph is confusing (especially the filtering actually done to fulfil the North et al. (1982) rule), please rewrite.*

R: Overall the PCA and EOF technique are related, but differences exist and both abbreviations are often mixed up. Yes, the PCA provides information about the temporal pattern and the EOF gives information about the spatial pattern. After this paragraph we will add a simple explanation about PCA and EOF analysis. In the revised version of the manuscript, we will rewrite the entire section and clarify the differences of the two techniques. Additionally, we will mathematically describe the North et al. (1982) rule.

The concept of the Principal Component Analysis (PCA) was firstly described by Pearson (1902) and Hotteling (1935) and used for the first time by Lorenz (1959) for climatological research (Storch & Zwiers, 1999). The general aim of the PCA is to find a new set of axes which explains the most of the variability within the dataset. This is done by rotating the initial data onto axes which are orthogonal to each other (Schönwiese, 2013). For this purpose, a vector is necessary which indicates the direction of the new coordinate axis which is called

eigenvector. This type of vector doesn't change its direction by a rotation. Therefore, the eigenvectors are used as a transformation matrix for the input high dimensional datasets onto the new axis. To indicate if a rotation maximizes the explained variance, every eigenvector has a corresponding eigenvalue. The corresponding eigenvalue is a kind of stretch factor for the eigenvectors. A huge eigenvalue indicates that the eigenvector has to be strongly stretched to map high variabilities within the dataset which can be explained with the new set of axes. Therefore, the eigenvalue ($\lambda$) is equal to the variance of the time series ($\vec{X}$) from matrix M which got rotated by the corresponding eigenvector ($\vec{e}$) (Equation 1).

$$Var(\langle \vec{X}, \vec{e} \rangle) = \lambda \qquad (1)$$

To find a first set of eigenvalues and eigenvectors, the data is rotated until an axis can be defined which explains the highest variance. Storch & Zwiers (1999) described this with the effort of minimizing $\epsilon_1$ respectively to maximize $Var(\langle \vec{X}, \vec{e}^1 \rangle)$(Equation 2).

$$\epsilon_1 = Var(\vec{X}) - Var(\langle \vec{X}, \vec{e}^1 \rangle) \qquad (2)$$

Finally, the rotated data forms the first component. Like in a traditional coordinate system, it is possible to calculate a new axis which is orthogonal to the first one. Therefore, the second component is formed by an orthogonal rotation around the axis of the first component. The total number of components is given by the absolute number of time series. To compute the time series for the first component, the e.g. first value of all input time series is multiplied with the individual eigenvector and afterwards, summed up over all time series for each year. This process forms the time series of a principle component which has the same temporal coverage as the input time series.

Especially a separate analysis of the eigenvectors of $\vec{X}$ is commonly used. This analysis is known as Empirical Orthogonal Functions (EOF) and the goal of it is to identify spatially coherent climate patterns which explain a significant part of the variance for a specific region (this is shown and used for example in Ionita et al. (2008)). Therefore, the largest part of the variance can be explained by the pattern of the leading EOF.

*L132-133: Why are you mentioning this here? It should be already stated in Section 2.1.*

R: It is mentioned here again to explain why we used the filling algorithm of Josse and Husson (2016).

*L133-140: How can you be sure that by using the gap fill method, you will not influence your results? Also, why would you need to fill in the gaps for 400 years knowing that your climate data only goes up to 1851?*

R: Since the used climate data is only available from 1851, the advantage of using the presented tree ring network is to go beyond this time scale and to introduce a new perspective on the observed relationship between $\delta^{18}O_{cel}$ and ENSO activity back in the past. If we want to go back in the past the filling algorithm is necessary because the temporal coverage of the $\delta^{18}O_{cel}$ records are different as described in the "Material and Method" section.

**EOF2 JJA - OWDA**
**Explained Variance - 16.1%**

[Figure]

Figure 1: The EOF for the second component of the OWDA (Cook et al., 2015) for JJA which explains 16.1% of the variance.

To test if our results are influenced by the gap filling method, we tested the correlation with the reconstruction with summer wetness and dryness reconstruction from the Old World Drought Atlas (OWDA) which was developed by Cook et al. (2015). At first, we investigated the correlation between the first 4 PCs for JJA with the PC1 of the isotope network for the period 1850 to 2005, where we have closely the highest sample density. The highest correlation is detected with the PC2 (EOF plot below) of Cook et al. (2015). The component is explaining 16.1% of the variance (very similar to the explained variance of the first component of the isotope network) and the correlation is characterized by Pearson's R=0.43 and p-value=3.1e^-08. If we test the correlation for the entire period 1600 to 2005, it would be expected that the correlation is strongly changing in the case that the filling algorithm is influencing the representation of climate signals. In our study, the correlation is only slightly changing to Pearson's R=0.39 (p-value=4.2e^-16) which indicates that the influence of the filling algorithm on the results is not so strong, because climate signals are presented in a

similar manner as in comparison for the period with a high sample coverage. Nevertheless, we have to consider the uncertainties based on the used gap filling method, especially for the first decades where the sample density is low. Therefore, the interpretation of the first decades have to be handled with care and is not pronounced in our manuscript.

[Figure]

Figure 2: Comparison of the time series of the first component of our study and the second component of the OWDA (Cook et al., 2015) for JJA.

*L141-146: What do you mean? The whole paragraph is pretty confusing and after reading it several times, I still do not get what you are really doing here.*

R: We will try to explain the technique behind the composite maps. In the next version of the manuscript, we will rewrite the explanation of composite maps for a better understanding. This will also make it easier to investigate the presented results of the composite maps for readers.

*What kind of information is providing the geopotential height 500mb (Z500) for the analysis?*

R: The geopotential height is a standard variable in atmospheric sciences which is often represented in contour maps with isohyets. They connect places at the same altitude at which a certain air pressure (here 500 hPa which is on average 5500 meters high) prevails. This height is called geopotential in meteorology. With the differences in the geopotential height maps, it is possible to identify low- and high-pressure systems based on the different pressure patterns as for example shown in Figure 4. From the Z500 maps, the wind direction can be determined which is helpful for the understanding and the investigation of large-scale flows in the atmosphere. A big advantage of this variable is that the influence of friction at the surface and the influence of dynamic disruptive factors of the upper layers of the atmosphere, e.g. the Jetstream, are relatively minor.

*L151: 'Nino 3.4 index' this should come earlier in section 2.2 when you are presenting the environmental data used in the analyses.*

R: We agree with the reviewer and we will add more information about it in the "Material and Method" section (see above).

*L158-160: The first sentence does not provide any clear information. Please remove.*

R: We will follow the suggestion of the reviewer and remove the sentence from the manuscript.

*Figure 1: Could you add more information in the figure A related to the latitude of each site? The names and characteristics of the sites are not presented anywhere in the manuscript. Even though the data have already been published, a Table with sites information should be included. In figures B and C, how is it possible that R2 and p-value are exactly the same for the relationships between δ18Ocel and altitude and between δ18Ocel and latitude? I suspect that there is a mistake here.*

R: We agree with the reviewer that more information about the sample sites would be helpful. Therefore, we will add a table with the coordinates, tree type and altitude for each site in the revised version of the manuscript. Furthermore, the reviewer is right that there is a mistake within the textbox of Figure 1B (should be $R^2=0.417$ and p-value=$2.17e^{-04}$). We will correct it in the new version of the manuscript.

*L160-161 and L172-173: Since no information about the exact location of the site presented in Figure 1A is provided, it is difficult to follow this statement.*

R: We agree with it. It will be easier when the table with the site characteristics are included.

*L162: I would remove 'This might be determined genetically,' from the sentence as it is not completely accurate (different species of Quercus also have different genetic information).*

R: We thank the reviewer for that suggestion, and we will remove that part.

*L165: You could also cite more updated papers describing differences between an giosperms and gymnosperms, e.g.: Carnicer J., Barbeta A., Sperlich D., Coll M., Penuelas J., Contrasting trait syndromes in angiosperms and conifers are associated with different responses of tree growth to temperature on a large scale, Front. Plant Sci., 4, 409, https://doi.org/10.3389/fpls.2013.00409, 2013*

R: We will integrate the mentioned reference and further references in the revised version of the manuscript.

*L177: 'which could influence the relation by a latitudinal effect.' please rewrite as 'thus the latitudinal and altitudinal gradients may have confounding effects on δ18Ocel' or something similar*

R: Thank you for the comment. We will rewrite this sentence to make it clearer.

*L177-179: I would rewrite this sentence as the effects of the two gradients on δ18Ocel have already been observed and documented in many other studies, for instance: Szejner, P., W. E. Wright, F. Babst, S. Belmecheri, V. Trouet, S. W. Leavitt, J. R. Ehleringer, R. K. Monson, Latitudinal gradients in tree ring stable carbon and oxygen isotopes reveal differential climate influences of the North American Monsoon System, J. Geo- phys. Res. Biogeosci., 121, 1978–1991, doi:10.1002/2016JG003460. 2016*

R: We will rewrite the sentence and highlight the findings of previous studies in this field.

*L181-186: Here again comes the confusion between EOF and PCA. You should clarify from the beginning (see previous comment) what is the difference between the two especially given that EOF1 and PC1 both seem to explain 16.2% of the variance in the records.*

R: We see the point that the differences between EOF and PCA is not well explained and can be confusing. With the explanation above, it will be easier to understand the differences. As mentioned above, we will extend the explanation of the EOF and PCA techniques.

*Figure 4: In the legend, you are describing the columns not the rows.*

R: Thank you for the comment. We will change it.

*L216 Is the distribution of PC1 for El Niño (or La Niña) years significantly different from that during normal years (i.e. when excluding El Niño/La Niña years)?*

R: Yes. This is shown in the first figure of the supplement and written from L214 to L216. To make it easier to understand, we will rewrite these sentences in the revised version of the manuscript.

*L222-224: Please rewrite the sentence. As it reads now, it looks like you are saying that Europe is characterized by higher precipitation and lower air surface temperatures in summer! And it is not clear what the parentheses apply for.*

R: We will rewrite and hopefully improve this part and we will give a short general introduction how to read and analyze composite maps. The changes will make it easier to understand Figure 4 and the argumentation.

*L230-231: 'because we to take into account. . .' Why would SPEI3 index accounting for the climate conditions prevailing over the previous season? So far, nothing has been said about this dataset.*

*L227-233: this part mostly belongs to the Material and Methods section and could be improved for readability.*

R: Good point! We will write more details about the SPEI3 dataset within the "Material and Method" section. Furthermore, we will highlight our motivation why we used that drought index.

*L235: 'the used reconstruction': which one?*

R: We will add another subchapter to the "Material & Method" section to describe the used reconstruction for the comparison within our study.

*L240: 'to capture a multi-seasonal signal' what do you mean?*

R: We agree with the reviewer that this part is not easy to understand. We will rewrite this paragraph and add further information about the goal and method of nudged model simulations which are listed in our response to the reviewer's comment L114-116.

*L243-244: where is it shown?*

R: This is shown in Figure 6.

*L244-245: so why then δ18Ocel is not more strongly related to δ18Osw? Your argument is contradictory with what is actually described.*

R: Thank you for the comment! The referred sentence is difficult to understand and we will re-rewrite and improve it.

*L239-248: And so what? What are you really trying to say here? Also, I do not think the results are properly discussed and compared to the literature.*

R: As mentioned above, we will extend the description of the performed analysis and rewrite and specify the goals which we want to achieve. Based on that, we will add further information that the main findings are better represented. We agree with the reviewer that

the results should be compared to literature and discussed more thoroughly. In the revised version of the manuscript, we will address and change these points.

*Figure 6: You mean the upper row Why description of Figure 6 comes before Figure 5?*

R: Thank you for the suggestion. Yes, we do mean the upper row; we will rewrite this caption in the revised version of the manuscript. Also, we will change the order of the figures so that the order is determined by the explanation with the text.

*L168-169 and L276: Please rewrite sentences*

R: We will rewrite the mentioned sentences.

*L236-237: The instability of the relationship between climate variables and ENSO has also been documented by other tree-ring studies in southern South America, e.g: Álvarez, C., Veblen, T.T., Christie, D.A., González-Reyes, Á., Relationships between climate variability and radial growth of Nothofagus pumilio near altitudinal treeline in the Andes of northern Patagonia, Chile. For. Ecol. Manage. 342, 112–121, 2015*

R: In the revised version of the manuscript, we will look for further literature in this field and we will integrate the mentioned reference and further references in the revised version of the manuscript.

**References used in our answer to the anonymous referee #1:**

Brönnimann, S., Luterbacher, J., Staehelin, J., Svendby, T. M., Hansen, G. and Svenøe, T.: Extreme climate of the global troposphere and stratosphere in 1940–42 related to El Niño, Nature, 431(7011), 971, doi:10.1038/nature02982, 2004.

Brönnimann, S., Xoplaki, E., Casty, C., Pauling, A. and Luterbacher, J.: ENSO influence on Europe during the last centuries, Climate Dynamics, 28(2–3), 181–197, doi:10.1007/s00382-006-0175-z, 2007.

Butzin, M., Werner, M., Masson-Delmotte, V., Risi, C., Frankenberg, C., Gribanov, K., Jouzel, J. and Zakharov, V. I.: Variations of oxygen-18 in West Siberian precipitation during the last 50 years, Atmospheric Chemistry and Physics, 14(11), 5853–5869, doi:10.5194/acp-14-5853-2014, 2014.

Compo, G. P., Whitaker, J. S., Sardeshmukh, P. D., Matsui, N., Allan, R. J., Yin, X., Gleason, B. E., Vose, R. S., Rutledge, G., Bessemoulin, P., Brönnimann, S., Brunet, M., Crouthamel, R. I., Grant, A. N., Groisman, P. Y., Jones, P. D., Kruk, M. C., Kruger, A. C., Marshall, G. J., Maugeri, M., Mok, H. Y., Nordli, ø., Ross, T. F., Trigo, R. M., Wang, X. L., Woodruff, S. D. and Worley, S. J.: The Twentieth Century Reanalysis Project, Quarterly Journal of the Royal Meteorological Society, 137(654), 1–28, doi:10.1002/qj.776, 2011.

Cook, E. R., Seager, R., Kushnir, Y., Briffa, K. R., Büntgen, U., Frank, D., Krusic, P. J., Tegel, W., Schrier, G. van der, Andreu-Hayles, L., Baillie, M., Baittinger, C., Bleicher, N., Bonde, N., Brown, D., Carrer, M., Cooper, R., Čufar, K., Dittmar, C., Esper, J., Griggs, C., Gunnarson, B., Günther, B., Gutierrez, E., Haneca, K., Helama, S., Herzig, F., Heussner, K.-U., Hofmann, J., Janda, P., Kontic, R., Köse, N., Kyncl, T., Levanič, T., Linderholm, H., Manning, S., Melvin, T. M., Miles, D., Neuwirth, B., Nicolussi, K., Nola, P., Panayotov, M., Popa, I., Rothe, A., Seftigen, K., Seim, A., Svarva, H., Svoboda, M., Thun, T., Timonen, M., Touchan, R., Trotsiuk, V., Trouet, V., Walder, F., Ważny, T., Wilson, R. and Zang, C.: Old World megadroughts and pluvials during the Common Era, Science Advances, 1(10), e1500561, doi:10.1126/sciadv.1500561, 2015.

Fraedrich, K.: An ENSO impact on Europe?, Tellus A, 46(4), 541–552, doi:10.1034/j.1600-0870.1994.00015.x, 1994.

Fraedrich, K. and Müller, K.: Climate anomalies in Europe associated with ENSO extremes, International Journal of Climatology, 12(1), 25–31, doi:10.1002/joc.3370120104, 1992.

Helama, S., Meriläinen, J. and Tuomenvirta, H.: Multicentennial megadrought in northern Europe coincided with a global El Niño–Southern Oscillation drought pattern during the Medieval Climate Anomaly, Geology, 37(2), 175–178, doi:10.1130/G25329A.1, 2009.

Hirahara, S., Ishii, M. and Fukuda, Y.: Centennial-Scale Sea Surface Temperature Analysis and Its Uncertainty, Journal of Climate, 27(1), 57–75, doi:10.1175/JCLI-D-12-00837.1, 2014.

Hotelling, H.: The most predictable criterion, Journal of Educational Psychology, 26(2), 139–142, doi:10.1037/h0058165, 1935.

Ionita, M., Lohmann, G. and Rimbu, N.: Prediction of Spring Elbe Discharge Based on Stable Teleconnections with Winter Global Temperature and Precipitation, J. Climate, 21(23), 6215–6226, doi:10.1175/2008JCLI2248.1, 2008.

Josse, J. and Husson, F.: missMDA : A Package for Handling Missing Values in Multivariate Data Analysis, Journal of Statistical Software, 70(1), doi:10.18637/jss.v070.i01, 2016.
Lloyd-Hughes, B. and Saunders, M. A.: Seasonal prediction of European spring precipitation from El Niño–Southern Oscillation and Local sea-surface temperatures, International Journal of Climatology, 22(1), 1–14, doi:10.1002/joc.723, 2002.

Lorenz, E. N.: Prospects for statistical weather forecasting: final report: Statistical Forecasting Project, Massachusetts Institute of Technology, Dept. of Meteorology, Cambridge, Mass. [online] Available from: https://catalog.hathitrust.org/Record/007185933 (Accessed 16 November 2018), 1959.

North, G. R., Bell, T. L., Cahalan, R. F. and Moeng, F. J.: Sampling Errors in the Estimation of Empirical Orthogonal Functions, Mon. Wea. Rev., 110(7), 699–706, doi:10.1175/1520-0493(1982)110<0699:SEITEO>2.0.CO;2, 1982.

Pearson, K.: On lines and planes of closest fit to systems of points in space, The London, Edinburgh, and Dublin Philosophical Magazine and Journal of Science, 2(11), 559–572, doi:10.1080/14786440109462720, 1902.

Pozo-Vázquez, D., Gámiz-Fortis, S. R., Tovar-Pescador, J., Esteban-Parra, M. J. and Castro-Díez, Y.: El Niño–southern oscillation events and associated European winter precipitation anomalies, International Journal of Climatology, 25(1), 17–31, doi:10.1002/joc.1097, 2005.
Rayner, N. A.: Global analyses of sea surface temperature, sea ice, and night marine air temperature since the late nineteenth century, Journal of Geophysical Research, 108(D14), doi:10.1029/2002JD002670, 2003.

Schönwiese, C.-D.: Praktische Statistik für Meteorologen und Geowissenschaftler, Borntraeger, Berlin., 2013.

Storch, H. v and Zwiers, F. W.: Statistical analysis in climate research, Cambridge University Press, Cambridge ; New York., 1999.

---

## Author Comment (AC2) · 22 Sep 2020

R: We thank the anonymous referee #2 for the time and effort in reviewing our manuscript (cp-2020-39). The comments, suggestions, critics and feedback raised in the review are highly appreciated as they help us to clarify our statements and to improve the quality of our manuscript. Below you will find a point by point response (*reviewer*, response) regarding the suggestions and concerns raised throughout the review process.
* * *
*Summary*
*The authors investigate delta18O tree ring records of 26 site distributed over Europe for the last 400 years. They claim that they were able to identify a connection of the leading mode of variability of this data to El Nino Southern Oscillation. They speculate that this connection is only found in the last 130 years and thus the connection is not stable in time. The second mode of the data is suggested to reconstruct regional summer atmospheric circulation. Finally, the author team claim that delta18O tree ring records can be used to reconstruct atmospheric circulation.*

*General comments*
*The topic would certainly be of high interests. However, the authors fail to convincingly show evidence for their main claims listed in the summary. Therefore, I recommend to reject the manuscript.*

*Major comments*
*1. At several places in the manuscript the authors claim that their analysis suggests "the relationship between ENSO and the European climate may not stable over time". The connection is only found for the instrumental period after 1850 CE. I think the first order interpretation is that the reconstruction of ENSO might be not perfect, as normally the reconstruction methods are trained in the last 100 to 150 years. So differences between the training periods and the period before are a hint that the reconstruction might be not successful. So, from your analysis you cannot conclude that you have identified non stationarity of teleconnections.*

R: We thank the reviewer for this comment. It stresses that we have to put further work into the manuscript clarifying our results and interpretations as we are not intending to say that we have identified a non-stationarity teleconnection. We only say that the correlations and coincidence rates are weakening, which indicates that the relationship between PC1 of the δ18Ocel network and ENSO might not be stable over time. Therefore, we can only suggest that the relationship between ENSO and the European climate may be not stable over time which is supported also by other studies based on proxy data (e.g. Rimbu et al., 2003). The idea of a unstable relationship is also supported by the results from studies based on instrumental data (e.g. Fraedrich, 1994; Fraedrich and Müller, 1992; Pozo-Vázquez et al., 2005) or studies based on ocean-atmosphere coupled models (e.g. Raible et al.,

2004; Deser et al., 2006; Brönnimann et al., 2007) The fact that the relationship between climate variables and ENSO is not stable over time has been also recognized in tree-ring studies in other regions, e.g. South Amerika (Álvarez et al., 2015). Nevertheless, we are aware of the fact that further research is necessary to make a confident statement.

It is true that reconstructed ENSO indices before instrumental period are not perfect which could be the cause of decrease in the correlations between our PC1 and these indices before 1850s. However, it is also true, that 1850s represent the end of Little Ice Age (LIA) period, when ENSO properties and its teleconnections changed significantly. Modeling studies (e.g. Henke et al., 2017) report an increased frequency of El Nino during LIA due to southern displacement of ITCZ. Although the reviewer hypothesis could be true, also our interpretation that decreasing in the correlation between our PC1 and reconstructed ENSO indices is due to changes in ENSO teleconnections over Europe could be true.

We agree with the reviewer that a comparison with ENSO reconstruction which are trained within the last 150 years can be problematic, since every reconstruction has its own uncertainties and limits. Confident statements are only possible for the time from 1850 onwards, since instrumental measurements of different climate variables are available. However, the usage of climate reconstructions is the only possibility to test and analyze the teleconnection before 1850 because no observational data is available. Overall, we tested the relationship with three different reconstructions for two different time ranges, where the sample density of isotope network is relatively high and the correlation is getting weaker over time (Li et al., 2011; Li et al., 2013; Dätwyler et al., 2019) which is shown in Table 1. Not only the correlation between the first component of the $\delta^{18}O_{cel}$ network and the ENSO reconstructions is getting weaker, also the correlation between different ENSO reconstructions is getting weaker in the 18th century which was shown for specific periods in Dätwyler et al. (2019). They suggest that is based on changes of the ENSO teleconnections, because they found a consistent teleconnection pattern that is different to the known teleconnection pattern of ENSO in the instrumental period (Dätwyler et al., 2019). Based on this argumentation and comment of the reviewer, we will write more carefully about the stationarity of teleconnections and the stability in the revised version of the manuscript.

| | LI ET AL. 2011 | LI ET AL. 2013 | DÄTWYLER ET AL. 2019 |
|---|---|---|---|
| **1750-1849** | r=0.121 p-value=0.231 | r=-0.008 p-value=0.936 | r=-0.078 p-value=0.442 |
| **1850-1949** | r=0.223 p-value=0.026 | r=0.303 p-value=0.002 | r=0.296 p-value=0.003 |

Table 1: Correlation between the first component of the $\delta^{18}O_{cel}$ network with three different ENSO reconstructions for two different time periods.

In order to show how stable the connection between the sea surface temperature (SST) of the tropics and the first mode of $\delta^{18}O_{cel}$ is, we have computed stability maps of the correlation between these two quantities. The stability map is a tool which is primarily used for streamflow predictions to identify stable teleconnection (for more details see Ionita et al. (2008), Lohmann et al. (2005) or Rimbu et al. (2005)). SST anomalies from the Extended Reconstructed Sea Surface Temperature (ERSST) v5 dataset (Huang et al., 2017) have been correlated with the first mode of $\delta^{18}O_{cel}$ in a moving window of 21 years. The correlation is considered to be stable for those grid points where anomalies are significantly correlated at the 90% level ($r = 0.25$) for more than 80% of the 21-yr windows, covering the period 1850–2005. This is shown in Figure 1. The results from the stability maps analysis will be also integrated in the revised version of the manuscript.

[Figure]

Figure 1: Stability maps for the correlation between SST anomalies (Huang et al., 2017) and the first mode of $\delta^{18}O_{cel}$. The color bar indicates how many years are characterized by a significant correlation (at the 90% level) in a 21-yr window, covering the period 1850–2005. Positive correlations are shown from yellowish to reddish and negative correlations from greenish to blueish.

As it can be inferred from Figure 1 (below), one of the largest locations with stable correlations is located in the tropical Pacific. Stable correlations are shown from September (last year) to June which also supports our result that $\delta^{18}O_{cel}$ is able to capture a multi-seasonal signal and that the first mode of $\delta^{18}O_{cel}$ is sensitive for ENSO variability. With the help of these maps, it is also possible to track the

development of the correlation over time. During July and August, the pattern dissolves in the tropical Pacific whereas a stable negative correlation is shown around Europe. Based on these results, we suggest that the relation between the first mode of $\delta^{18}O_{cel}$ and the SST in the tropical Pacific/Atlantic in winter and spring is stable in the period from 1850 to 2005. We will add the shown figure in the appendix of the revised version of the manuscript and we will discuss why $\delta^{18}O_{cel}$ is able to capture this stable teleconnection.

2. The conclusion that the analysis shows that "We infer that the investigation of large- scale atmospheric circulation patterns and related teleconnections far beyond instrumental records can be done with oxygen isotopic signature derived from tree rings." is not convincingly demonstrated. There is only 2 line in the introduction which gives a hint why this should be possible, i.e., fractionation happens during the transport from source to sink areas. Most of the studies however try to reconstruct temperature and precipitation when using delta18O as delta18O is first order temperature dependent. The authors also nicely discuss that fractionation processes are also relevant within the tree. Then, at the different sites the water can be transported form different source regions during the seasonal cycle, e.g., North Atlantic versus Mediterranean, or long distance transport versus local water recycling. Moreover, seasonality plays an important role, so mostly tree ring records are interpreted to record growing season signals and not winter signals. So given all these uncertainties how can the transport aspect (which is related to the atmospheric circulation) survive?

R: We agree with the reviewer that transport can lead to uncertainties. The earth system is interconnected and includes many factors that make it hard to identify a single cause. Nevertheless, we think that the aforementioned statement of the reviewer is somehow mixes up several points. First of all, the source of oxygen isotopes in cellulose is mostly the water from the atmosphere. Through precipitation, the soil gets enriched with water and depending on the length and depth of the root system, a tree uses surface water or water from groundwater reservoirs for photosynthesis processes. Therefore, $\delta^{18}O_{cel}$ is coupled to the hydrological cycle on Earth. This cycle is strongly dependent on large-scale atmospheric circulation, since large-scale flows determine the path of clouds, precipitation patterns and the distribution of water vapor in the atmosphere. Also, the mentioned water transport from different source regions is related to large-scale atmospheric circulation. Synoptic patterns and associated indices (e.g. cyclone and anticyclone activity, air pressure) have also been reconstructed from $\delta^{18}O_{cel}$, with stronger correlations revealed during extreme years or periods (e.g. Saurer et al., 2012). $\delta^{18}O_{cel}$ is related to the $\delta^{18}O$ of the precipitation source via soil water. $\delta^{18}O$ of soil water constitutes the $\delta^{18}O$ input to the arboreal system and represents an average $\delta^{18}O$ over several precipitation events modified by partial evaporation from the soil (depending on soil texture and porosity) and by a possible time lag, depending on rooting depth (Saurer et al., 2012). $\delta^{18}O_{cel}$ is further dependent on two tree-internal processes: evaporative [18]O-enrichment of leaf or needle water via transpiration, as well as biochemical fractionations and isotopic exchange of $\delta^{18}O$ with trunk water during cellulose biosynthesis (e.g. Barbour, 2007; Kahmen et al., 2011; Roden et al., 2000; Saurer et al., 2012; Treydte et al., 2014). Fractionations occurring at leaf level, are partly reset by isotopic exchange between sugar oxygen and stem water during cellulose synthesis in the trunk allowing the

soil water isotopic signal to be largely preserved in the tree-ring cellulose (e.g. Gessler et al., 2014 and citations therein).

The seasonality is essential since trees store the climate signal within the $\delta^{18}O_{cel}$ ratio during the growing season. One of the key messages of the manuscript is that winter climate signals can be stored which is shown by significant correlation plots (Figure 6). We argue that this is possible through hydrological feedback processes (e.g. via the soil moisture content). The fact that trees are able to store a winter signal is not new and was published for example by Treydte et al. (2006) and Treydte et al. (2014).

With the knowledge about the physical climate processes (the hydrological cycle) and the understanding of the theory behind the fractionation of $\delta^{18}O_{cel}$ ratio, we feel that the points mentioned by the reviewer are not uncertainties. They are part of the climate system and essential for the understanding of climate proxies.

3. For the first EOF I have a different interpretation, which takes into account the fact that temperature play the dominant role in delta18O. What we see is a monopole structure. The authors claim to see a link to ENSO. I hypothesize that the link is simply due to the fact that ENSO has a global impact on the global mean temperature. So, due to an El Nino event, the Earth warms and thus also the North Atlantic and the Mediterranean (visible in the composite plots). Warmer source regions affect the fractionation of delta18O without any change of the circulation we see in the sink regions (at the tree sites) a uniform signal.

R: The climate over the European region has a rather peculiar variability, and in most of the cases the first EOF is monopolar, especially if we consider mostly the central and western part of Europe in the analysis (like in our isotope network). This is valid for precipitation, temperature and even for drought indices. As we discussed in the first point raised by the reviewer, we disagree that the relationship with ENSO is purely by chance given the tests of statistical significance. The fact that PC1 correlates significantly with the SST from the tropical Pacific in winter and this signal is transmitted to the tropical Atlantic in spring and central Atlantic in summer is part of the natural cycle of ENSO. The ENSO anomalies (either El Niño or La Niña) develop in winter and it needs 3-6 months to see a signal in the European climate. This lagged relationship is typical for many ENSO related teleconnections. This long transition from the tropical Pacific to central North Atlantic affects in turn the large-scale atmospheric circulation and as a consequence the climate over Europe, especially in spring and summer. We do not agree with the idea that just because we have El Niño or La Niña the earth will be either warm or colder. ENSO dynamics are more complicated than this and the signal from ENSO to the $\delta^{18}O$ is transmitted mainly via the large-scale atmospheric circulation.

On the composite maps the ENSO teleconnection patterns are clearly emphasized. This means that there is not only a thermodynamical influence on $\delta^{18}O$, that is variation of global temperature with ENSO, playing a role, but also ENSO related teleconnections impact on European climate is important.

4. There are problems with the data (see comment below Section 2.2, L132-140, L145) ignored which might be influential to the analysis.

R: In the revised version of the manuscript we will make a better description of Section 2.2 so it is clearer for the reader which data are used. We will add the source of the data also in the figure caption, for each figure (where needed). Please see our detailed answers and argumentation to the aforementioned comment below.

Minor comments

L18: What is meant by "reflects a multi-seasonal climatic signal."? ENSO works on timescales of 3 to 7 years.

R: Quote from the submitted manuscript (L17-18): "The first mode of $\delta^{18}O$ variability is associated with anomaly patterns of the El Niño-Southern Oscillation (ENSO) and reflects a multi-seasonal climatic signal". So, the last part of the sentence is connected to the first mode of $\delta^{18}O$ which means that the first mode of $\delta^{18}O$ variability reflects a multi-seasonal climatic signal. The multi-seasonal climatic signal stored in $\delta^{18}O$ is essential to capture El Niño/La Niña events and the related ENSO variability.

L20: "out of phase variability": I would interpret this in the time and not in space as the authors. Just say the 2. EOF is a dipole pattern with centers over northwestern and southeastern Europe.

R: Thank you for the suggestion. We will rewrite it in the revised version of the manuscript.

L47-50: Hard to read.

R: We will rewrite the mentioned section.

L53: please change to "leaf water clearly affects"

R: We will change it in the new version of the manuscript.

L55: I disagree with this statement, see major comments 2 and 3.

R: In this point, we disagree with the reviewer. Please have a look above for our argumentation.

L56: What is meant by "resulting long-term perspective"? Where does it result from?

R: Thanks to the reviewer for highlighting this. We mean the long-term perspective that results from the usage of $\delta^{18}O_{cel}$ as a climate proxy. We will rewrite the sentence to make it clearer.

L84: Created -> generated

R: We will change it.

L88: Please include a space between number and unit throughout the manuscript.

R: Good point. We will improve it in the new version of the manuscript.

L91: What is SMOW?

R: It should be VSMOW (Vienna Standard Mean Ocean Water). We will rewrite it.

Section 2.1: Which method is used to get the delta18O samples from trees. This is relevant as studies show that the method (pooling or not pooling) makes a huge difference Hangartner et al. Methods to merge overlapping tree-ring isotope series to generate multi-centennial chronologies CHEMICAL GEOLOGY Volume: âAR 294 Pages: âAR 127-134 Published: âAR FEB 10 2012

R: We agree with the reviewer that further information about the sampling method is very important. According to Treydte et. al (2007a, b), all tree rings from the same year were pooled prior to cellulose for the majority of sites. We will add this in Section 2.1 in the revised version of the manuscript.

Section 2.2: Again if is unclear what the authors are using. Is it the ensemble mean of 20CR or an individual ensemble member. Please note that the 20CR is only constraint with sea level pressure (SLP) data so no sea surface temperatures (SST) are used which are relevant for ENSO. My guess is that the authors use the ensemble mean. This is problematic as in the early part of the reanalysis the constraint (via SLP) is rather weak leading variance deflation and thus can have a strong impact on the analysis (so it is normally recommended to use all individual ensemble members). As said, the other problem is that ENSO might not be realistically included in the first part of the reanalysis.

R: The argumentation that 20CR (V2c) is not using SST is not correct since the SST dataset from 18 members of Simple Ocean Data Assimilation with Sparse Input version 2 (SODAsi.2, Giese et al., 2016) are used as SST boundary condition (the high latitudes (>60°) were corrected to COBE-SST2 (Hirahara et al. 2014)). Furthermore, the sea ice cover (SIC) reconstruction of Hirahara et al. (2014) are also used as boundary condition (for more details see https://www.psl.noaa.gov/data/gridded/data.20thC_ReanV2c.html). Therefore, the SST and SIC are influencing the atmosphere (Compo et al, 2011) and therefore, ENSO activity is represented in 20CR (V2c).

In our study, we used the ensemble mean of 20CR (V2c). The ensemble spread is definitely bigger at the beginning compared to the end. In our opinion, this is not the reason why we will not use the ensemble mean especially if we focus on around 150 years of the reanalysis. Furthermore, the advantage of our study is that we are focusing on the climate in Europe. For this continent the effect of large ensemble spread at the beginning of 20CR (V2c) is relatively minor compared to other continents since a lot of observations are available from the beginning of the observation period.

Section 2.3: too short and not clear why the simulations are used and how the simulations are generated. The reader needs to understand which model is used and how, just references is not enough.

R: We thank the reviewer for the comment. We will extend the section and explain the nudged model simulations in more detail in the revised version of the manuscript.

Section 2.4: It reads like EOF and PCA are different analysis, but actually they are not. The method of empirical orthogonal function (EOF) analysis is a decomposition of a data set in terms of orthogonal basis functions determined from this data. Thus it is the same as geographically weighted PCAs.

R: We identify the dominant pattern of $\delta^{18}O_{cel}$ variability using Empirical Orthogonal Function (EOF) analysis (von Storch and Zwiers, 1999). This helps to eliminate the noise from the data and identify the climatic signal (etc.).

The concept of the Principal Component Analysis (PCA) was firstly described by Pearson (1902) and Hotteling (1935) and used for the first time by Lorenz (1959) for climatological research (Storch & Zwiers, 1999). The general aim of the PCA is to find a new set of axes which explains the most part of the variability within the dataset. This is done by rotating the initial data onto axes which are orthogonal to each other (Schönwiese, 2013). For this purpose, a vector is necessary which indicates the direction of the new coordinate axis which is called eigenvector. This type of vector doesn't change its direction by a rotation. Therefore, the eigenvectors are used as a transformation matrix for the input high dimensional datasets onto the new axis. To indicate if a rotation maximizes the explained variance, every eigenvector has a corresponding eigenvalue. The corresponding eigenvalue is a kind of stretch factor for the eigenvectors. A huge eigenvalue indicates that the eigenvector has to be strongly stretched to map high variabilities within the dataset which can be explained with the new set of axes. Therefore, the eigenvalue ($\lambda$) is equal to the variance of the time series ($\vec{X}$) from matrix M which got rotated by the corresponding eigenvector ($\vec{e}$) (Equation 1).

$$Var(\langle \vec{X}, \vec{e} \rangle) = \lambda \qquad\qquad (1)$$

To find a first set of eigenvalues and eigenvectors, the data is rotated until an axis can be defined which explains the highest variance. Storch & Zwiers (1999) described this with the effort of minimizing $\epsilon_1$ respectively to maximize $Var(\langle \vec{X}, \vec{e}^1 \rangle)$ (Equation 2).

$$\epsilon_1 = Var(\vec{X}) - Var(\langle \vec{X}, \vec{e}^1 \rangle) \qquad\qquad (2)$$

Finally, the rotated data forms the first component. Like in a traditional coordinate system, it is possible to calculate a new axis which is orthogonal to the first one. Therefore, the second component is formed by an orthogonal rotation around the axis of the first component. The total number of components is given by the absolute number of time series. To compute the time series for the first component, the e.g. first value of all input time series is multiplied with the individual eigenvector and afterwards, summed up over all time series for each year. This process forms the time series of a principle component which has the same temporal coverage as the input time series.

Especially a separate analysis of the eigenvectors of $\vec{X}$ is commonly used. This analysis is known as Empirical Orthogonal Functions (EOF) and the goal of it is to identify spatially coherent climate patterns which explain a significant part of the variance for a specific region (this is shown and used

for example in Ionita et al. (2008)). Therefore, the largest part of the variance can be explained by the pattern of the leading EOF.

L132-140: How many tree ring records cover the entire period with no gaps? How sensitive it the analysis to filling the gaps? How many cycles are needed to reach convergence? What if you use only the tree ring records which cover the entire period?

R: The temporal coverage is listed in Line 100-102. "Here, we use the extended ISONET+ product with the longest chronologies covering a period from 1600 to 2005. The highest data density is available for the period 1850-1998 with 26 time series available for further analysis. 12 time series cover the entire period of 400 years." The 12 time series, which are mentioned at the end, cover the entire period. Our goal is to use all time series of the ISONET+ product and therefore, we have to fill the gaps. If we calculate the EOFs and PCs for the period 1850 to 1998 the spatial and temporal correlation to the EOFs and PCs is very high. If we remove samples in the period 1850 to 1998, also the PCs and EOFs are changing because the input variability of the network is changing. The results are similar patterns, but the temporal and spatial correlation is varying. For a comparison, the reader is referred to the publication of the used isotope network of Treydte et al (2007b).

[Figure]

Figure 2: The EOF for the second component of the OWDA (Cook et al., 2015) for JJA which explains 16.1% of the variance.

To test if our results are influenced by the gap filling method, we tested the correlation with the reconstruction with summer wetness and dryness reconstruction from the Old World Drought Atlas (OWDA) which was developed by Cook et al. (2015). At first, we investigated the correlation between the first 4 PCs for JJA with the PC1 of the isotope network for the period 1850 to 2005, where we have

closely the highest sample density. The highest correlation is detected with the PC2 (EOF plot below) of Cook et al. (2015). The component is explaining 16.1% of the variance (very similar to the explained variance of the first component of the isotope network) and the correlation is characterized by Pearson's R=0.43 and p-value=3.1e^-08. If we test the correlation for the entire period 1600 to 2005, it would be expected that the correlation is strongly changing in the case that the filling algorithm is influencing the representation of climate signals. In our study, the correlation is only slightly changing to Pearson's R=0.39 (p-value=4.2e^-16) which indicates that the influence of the filling algorithm on the results is not so strong, because climate signals are presented in a similar manner as in comparison for the period with a high sample coverage. Nevertheless, we have to consider the uncertainties based on the used gap filling method, especially for the first decades where the sample density is low. Therefore, the interpretation of the first decades need to be handled with care, and statements should be regarded as less robust.

[Figure]

Figure 3: Comparison of the time series of the first component of our study and the second component of the OWDA (Cook et al., 2015) for JJA.

L142: This is certainly not extreme.

R: The detection of extremes by using the standard deviation is often used in climate science. There are many other definitions of extremes. We decided to use the standard deviation as an approximation given the available degrees of freedom. These extremes may not be extremely rare but certainly unusual and different from the mean.

L145: I would say that the authors misuse the composite analysis by focusing on the linear response. If they would like to analyze the linear response, a simple correlation analysis is enough. The beauty of the composite analysis is that you can easily show non-linear effects, but only if you make the difference between the mean above the threshold with the long term mean and in a second plot the mean below the negative threshold and the long term mean. This was done by Fraedrich 1992 mentioned in the manuscript. He highlighted the nonlinearity of the ENSO response over Europe and thus is in contradiction to the linear relationship suggested here.

R: We think that this point is a misunderstanding of our idea because we used the composite maps to learn more about the linear response. It is not our aim to test relation for non-linearities. The reference

of Fraedrich (1992) is used to highlight the uncertainties and show the reader that the shown relationship in the results cannot be seen as linear. In the revised version of the manuscript we will replace the figures with new figures in which we show both the high (PC1 > 1 standard deviation) and low high (PC1 < -1 standard deviation) composite maps. There are cases in which the high and low composite maps show different results, thus indicating non-linearity between the variables analyzed, but in our case the high and low composite maps are similar (in terms of structure), that is why we have shown just the difference map. But in order to be clearer in our description and analysis, we will add both the high and low composite maps.

L150: Event Coincidence Analysis needs to be explained.

R: Thank you for the comment. We will extend the explanation and also the entire "Material and Method" section.

L177-179: If this is the case one could speculate that EOF2 showing a North South patterns just resemble this latitudinal effect.

R: In this comment, two essentials were mixed up. First of all, the latitudinal effect was found by Daansgard (1964) and is defined as a steady shift of the isotope values from the equator to the poles. The reason for this is that temperature changes within different seasons and varies over the latitudes. According to Daansgard (1964) and Gat (2010), the mean annual temperature and the $\delta^{18}O$ in the atmosphere are strongly correlated. For a documentation of several effects on the $\delta^{18}O$ ratio, e.g. the continental effect, we refer to the publication of Gat (2010).

The latitudinal effect is important for the $\delta^{18}O_{cel}$ ratio at each site (Figure 1 of the submitted manuscript), because it is one factor which influences the source values ($\delta^{18}O_{precipitation}$, $\delta^{18}O_{soilwater}$). In contrast, the latitudinal effect cannot explain the (climate) variability which is investigated and represented by the PCA and EOF technique in our study. The climate variability is determined by several other quantities which cannot be explained by a simple latitudinal position, e.g. large-scale atmospheric flows. Based on that argumentation, we disagree with the speculation of the reviewer.

L195: I do not see this is there a typo and the authors mean PC1?

R: Thank you for the comment. We will rewrite it to: The time series of the second component (PC2) is characterized by an underlying positive trend from the mid of the 19th century onwards.

L200-210: Avoid using the bracket with e.g. (cold). This makes the text unreadable. Just say what you show in Fig. 4.

R: This is the standard way how composite maps are described. It is necessary to add the brackets since both tails of the distribution function are combined and represented in these maps. Therefore, we have to write this additional information to clearly represent both extremes. To make it more understandable, we will add an explanation of how composite maps are working.

L208: I do not see a AO pattern, again the reference to Fraedrich are incorrect as they claim that ENSO has a nonlinear response behavior over Europe.

R: It is true that there is some asymmetry between El Niño and La Niña teleconnection patterns over Europe. However, the analysis of long-term data reveals that El Niño (La Niña) conditions in the tropical Pacific are associated with a negative (positive) phase of the North Atlantic Oscillation in the North Atlantic region during winter. The composite pattern represented in the Figure reflects this ENSO-NAO relationship, although the pattern is not identical with the AO/NAO.

L225: Why drought we see a positive precipitation anomaly?

R: We think that your comment is a misunderstanding of our composite map. As mentioned above both extremes of the distribution are represented by the composite map. This is also the reason why we use the brackets for the explanation. We will add an explanation on how to read the composite maps.

Section3.4: What do we learn from this? What is shown and why? This section is unclear and to my feeling can be removed.

R: Our goal with the model output is to test the correlation between $\delta^{18}O_{cel}$ and $\delta^{18}O_P/\delta^{18}O_{SW}$ to identify if the water, which is used during the photosynthesis processes, has a multi-seasonal isotopic signature. If yes (as shown in our plots), it is an explanation how $\delta^{18}O_{cel}$ is able to capture the ENSO signal. The results are interesting because a significant ENSO influence of the European climate has been identified for the winter (Fraedrich and Müller, 1992; Fraedrich, 1994; Pozo-Vazquez et al., 2005; Brönnimann et al., 2004; Brönnimann et al., 2007) and spring season (Brönnimann et al., 2007; Lloyd-Hughes and Saunders, 2002; Helama et al., 2009). Since the climate information is predominantly stored in $\delta^{18}O_{cel}$ in the summer season in Europe, we suggest that our results are an indicator that $\delta^{18}O_{cel}$ is able to capture multi-seasonal climate signals through hydrological processes (soil moisture and soil water content) and therefore can capture a winter/spring climate signal. We agree with the reviewer that it is not clearly written in the manuscript. Therefore, we will add an extended explanation of the used dataset and results. Also, the findings will be compared with other studies and discussed in the revised version of the manuscript.

L250: section 3.5

R: We will correct it in the new version of the manuscript.

L251-252: This sentence is a repetition.

R: It is a short repetition of the results of the PCA. We will remove it in the next version of the manuscript.

L260 -263: You need to show this with more proxies. Note that dry conditions are not droughts!

R: Thank you for the comment. We will be very sensitive with usage of the term "drought" in the next version of the manuscript. Furthermore, the comparison with more proxies is a really good idea. Based on the complexity of this topic, this would be another big study and a topic for the future work in this field. In our study, we want to show primarily the results of the analysis of the $\delta^{18}O_{cel}$ isotope network.

L263-265: Given your study you cannot conclude this. The authors study certainly is inadequate to reconstruct blocking so this statement is not supported by the authors analysis.

R: We agree with the reviewer and we will remove this part from the manuscript.

L269-70: Please change to ". . . signal still dominates".

R: We will rewrite it.

Figures: Statistical significance is not tested (or not shown).

R: We think that the comment is relevant for Figure 4 and 5. To show the significance for the SST variable, we have attached the composite maps with a separation between high and low events. The SST and Z500 composite map with significance will be part of the new appendix of the revised version of the manuscript. In the revised version of the manuscript we will replace each figure with figures with the statistical significance.

[Figure]

Figure 4: High and low SST composite maps for the first component of the $\delta^{18}O_{cel}$ network. The significance is shown where a black grid is shown in front of the color. The ERSST (Huang et al., 2017) is used in this figure.

**References used in our answer to the anonymous referee #2:**

Álvarez, C., Veblen, T. T., Christie, D. and González-Reyes, Á.: Relationships between climate variability and radial growth of Nothofagus pumilio near altitudinal treeline in the Andes of northern Patagonia, Chile, , doi:10.1016/J.FORECO.2015.01.018, 2015.

Barbour, M. M.: Stable oxygen isotope composition of plant tissue: a review, Funct Plant Biol, 34, 83-94, 2007.

Brönnimann, S., Luterbacher, J., Staehelin, J., Svendby, T. M., Hansen, G. and Svenøe, T.: Extreme climate of the global troposphere and stratosphere in 1940–42 related to El Niño, Nature, 431(7011), 971, doi:10.1038/nature02982, 2004.

Brönnimann, S., Xoplaki, E., Casty, C., Pauling, A. and Luterbacher, J.: ENSO influence on Europe during the last centuries, Climate Dynamics, 28(2–3), 181–197, doi:10.1007/s00382-006-0175-z, 2007.

Compo, G. P., Whitaker, J. S., Sardeshmukh, P. D., Matsui, N., Allan, R. J., Yin, X., Gleason, B. E., Vose, R. S., Rutledge, G., Bessemoulin, P., Brönnimann, S., Brunet, M., Crouthamel, R. I., Grant, A. N., Groisman, P. Y., Jones, P. D., Kruk, M. C., Kruger, A. C., Marshall, G. J., Maugeri, M., Mok, H. Y., Nordli, ø., Ross, T. F., Trigo, R. M., Wang, X. L., Woodruff, S. D. and Worley, S. J.: The Twentieth Century Reanalysis Project, Quarterly Journal of the Royal Meteorological Society, 137(654), 1–28, doi:10.1002/qj.776, 2011.

Cook, E. R., Seager, R., Kushnir, Y., Briffa, K. R., Büntgen, U., Frank, D., Krusic, P. J., Tegel, W., Schrier, G. van der, Andreu-Hayles, L., Baillie, M., Baittinger, C., Bleicher, N., Bonde, N., Brown, D., Carrer, M., Cooper, R., Čufar, K., Dittmar, C., Esper, J., Griggs, C., Gunnarson, B., Günther, B., Gutierrez, E., Haneca, K., Helama, S., Herzig, F., Heussner, K.-U., Hofmann, J., Janda, P., Kontic, R., Köse, N., Kyncl, T., Levanič, T., Linderholm, H., Manning, S., Melvin, T. M., Miles, D., Neuwirth, B., Nicolussi, K., Nola, P., Panayotov, M., Popa, I., Rothe, A., Seftigen, K., Seim, A., Svarva, H., Svoboda, M., Thun, T., Timonen, M., Touchan, R., Trotsiuk, V., Trouet, V., Walder, F., Ważny, T., Wilson, R. and Zang, C.: Old World megadroughts and pluvials during the Common Era, Science Advances, 1(10), e1500561, doi:10.1126/sciadv.1500561, 2015.

Dansgaard, W.: Stable isotopes in precipitation, Tellus, 16(4), 436–468, doi:10.1111/j.2153-3490.1964.tb00181.x, 1964.

Dätwyler, C., Abram, N. J., Grosjean, M., Wahl, E. R. and Neukom, R.: El Niño–Southern Oscillation variability, teleconnection changes and responses to large volcanic eruptions since AD 1000, International Journal of Climatology, 39(5), 2711–2724, doi:10.1002/joc.5983, 2019.

Deser, C., Capotondi, A., Saravanan, R. and Phillips, A. S.: Tropical Pacific and Atlantic Climate Variability in CCSM3, , doi:10.1175/JCLI3759.1, 2006a.

Deser, C., Capotondi, A., Saravanan, R. and Phillips, A. S.: Tropical Pacific and Atlantic Climate Variability in CCSM3, JOURNAL OF CLIMATE, 19, 32, 2006b.

Dongmann, G., Nürnberg, H. W., Förstel, H. and Wagener, K.: On the Enrichment of $H_2^{18}O$ in the Leaves of Transpiring Plants, Radiat. and Environ. Biophys., 11, 41–52, 1974.

Fraedrich, K.: An ENSO impact on Europe?, Tellus A, 46(4), 541–552, doi:10.1034/j.1600-0870.1994.00015.x, 1994.

Fraedrich, K. and Müller, K.: Climate anomalies in Europe associated with ENSO extremes, International Journal of Climatology, 12(1), 25–31, doi:10.1002/joc.3370120104, 1992.

Gat, J.: Isotope Hydrology: A Study of the Water Cycle, World Scientific, London., 2010.

Gessler, A., Ferrio, J. P., Hommel, R., Treydte, K., Werner, R. A., and Monson, R. K.: Stable isotopes in tree rings: towards a mechanistic understanding of isotope fractionation and mixing processes from the leaves to the wood, Tree Physiology, 34, 796-818, 2014.

Giese, B. S., Seidel, H. F., Compo, G. P. and Sardeshmukh, P. D.: An ensemble of ocean reanalyses for 1815–2013 with sparse observational input, Journal of Geophysical Research: Oceans, 121(9), 6891–6910, doi:10.1002/2016JC012079, 2016.

Harris, P., Brunsdon, C. and Charlton, M.: Geographically weighted principal components analysis, International Journal of Geographical Information Science, 25(10), 1717–1736, doi:10.1080/13658816.2011.554838, 2011.

Helama, S., Meriläinen, J. and Tuomenvirta, H.: Multicentennial megadrought in northern Europe coincided with a global El Niño–Southern Oscillation drought pattern during the Medieval Climate Anomaly, Geology, 37(2), 175–178, doi:10.1130/G25329A.1, 2009.

Helle, G. and Schleser, G. H.: Beyond $CO_2$-fixation by Rubisco - an interpretation of 13C/12C variations in tree rings from novel intra-seasonal studies on broad-leaf trees, Plant, Cell and Environment, 27(3), 367–380, doi:10.1111/j.0016-8025.2003.01159.x, 2004.

Henke LMK 2017, Was Little Ice Age more or less El Nino like than Medieval Climate anomaly? Climate of the past, doi: 10.5194/cp-13-267-2017

Hirahara, S., Ishii, M. and Fukuda, Y.: Centennial-Scale Sea Surface Temperature Analysis and Its Uncertainty, Journal of Climate, 27(1), 57–75, doi:10.1175/JCLI-D-12-00837.1, 2014.

Hotelling, H.: The most predictable criterion, Journal of Educational Psychology, 26(2), 139–142, doi:10.1037/h0058165, 1935.

Huang, B., Thorne, P. W., Banzon, V. F., Boyer, T., Chepurin, G., Lawrimore, J. H., Menne, M. J., Smith, T. M., Vose, R. S. and Zhang, H.-M.: NOAA Extended Reconstructed Sea Surface Temperature (ERSST), Version 5, doi:10.7289/V5T72FNM, 2017.

Ionita, M., Lohmann, G. and Rimbu, N.: Prediction of Spring Elbe Discharge Based on Stable Teleconnections with Winter Global Temperature and Precipitation, J. Climate, 21(23), 6215–6226, doi:10.1175/2008JCLI2248.1, 2008.

Kahmen, A., Sachse, D., Arndt, S. K., Tu, K. P., Farrington, H., Vitousek, P. M., and Dawson, T. E.: Cellulose delta O-18 is an index of leaf-to-air vapor pressure difference (VPD) in tropical plants, Proceedings of the National Academy of Sciences of the United States of America, 108, 1981-1986, 2011.

Li, J., Xie, S.-P., Cook, E. R., Huang, G., D'Arrigo, R., Liu, F., Ma, J. and Zheng, X.-T.: Interdecadal modulation of El Niño amplitude during the past millennium, Nature Climate Change, 1(2), 114–118, doi:10.1038/nclimate1086, 2011.

Li, J., Xie, S.-P., Cook, E. R., Morales, M. S., Christie, D. A., Johnson, N. C., Chen, F., D'Arrigo, R., Fowler, A. M., Gou, X. and Fang, K.: El Niño modulations over the past seven centuries, Nature Climate Change, 3(9), 822–826, doi:10.1038/nclimate1936, 2013.

Lloyd-Hughes, B. and Saunders, M. A.: Seasonal prediction of European spring precipitation from El Niño–Southern Oscillation and Local sea-surface temperatures, International Journal of Climatology, 22(1), 1–14, doi:10.1002/joc.723, 2002.

Lohmann, G., Rimbu, N. and Dima, M.: Where can the Arctic oscillation be reconstructed? Towards a reconstruction of climate modes based on stable teleconnections, Climate of the Past Discussions, 1(1), 17–56, 2005.

Lorenz, E. N.: Prospects for statistical weather forecasting: final report: Statistical Forecasting Project, Massachusetts Institute of Technology, Dept. of Meteorology, Cambridge, Mass. [online] Available from: https://catalog.hathitrust.org/Record/007185933 (Accessed 16 November 2018), 1959.

[revised manuscript text omitted]

---

## Author Response (AR1)

**Reply to reviewer comments**

Dear Editor,

Please find below our point by point response to the comments of the reviewers (*reviewer*, response). We thank both anonymous referees for the time and effort in reviewing our manuscript (cp-2020-39). The comments, suggestions and feedback raised in the review are highly appreciated as they help us to clarify our statements and to improve the quality of our manuscript. In our response, we have highlighted the relevant changes made in the manuscript with line numbers. Furthermore, we have attached a marked-up manuscript version.

On behalf of all co-authors, I would like to thank the Editor.

Best regards,

Daniel Balting

**Anonymous referee #1**

*By analysing a European network of 26 tree-ring sites with δ18O measurements, the authors aim at extracting regional climate signal imprinted in the records to investigate the dominant modes of variability of the European climate and their relationships with the large-scale atmospheric circulation, in particular ENSO. Their findings suggest that climate variability in Europe is strongly modulated by ENSO teleconnections at least over the past 130 years, but that some differences arises between the northern and southern regions.*

*Although the results are promising, I do not think the manuscript is ready for publication yet. A restructuration and reorganisation of the paper is strongly needed. While the introduction is relatively well written and easy to follow, many confusions arise from the Material and Methods section and some clarifications are required to allow the readers to easily understand why and how the proposed analyses were made. The division of the 'Results and Discussion' section into two separate sections should improve the readability of the manuscript. It seems that the authors have not carefully re-read their manuscript to check for typos and ensure that the text is fully understandable before submitting it. The authors also should make an effort to properly, clearly and thoroughly discuss their results and their implications for the understanding of the atmospheric teleconnections. So far only in the Summary and Conclusion section are the results clearly highlighted and interpreted.*

R: We are glad that the results are promising, and we agree with the reviewer that structural changes are necessary. Therefore, the "Material and Methods" section have been sustainably revised by a more detailed description of the isotope measurements and of the used climate data including the used ENSO indices and the SST dataset as well as an extended explanation of the used methods [section 2].

Furthermore, we have separated our discussion from the results section. The comparison to other studies and the implications of our results have been worked out more in detail [section 3 & 4].

*Some additional comments and suggestions:*
*L20: 'may not be stable. . .'*

R: We have added the missing word [L20].

*L42-43: Actually, it is the other way around: δ18Ocel depends on δ18OSW but δ18OSW itself does not depend on δ18Ocel. Please rewrite.*

R: Thank you for the comment! We have fixed it in the revised version of the manuscript [L40-41].

*L55-56: You could also cite more papers showing the potential of δ18Ocel for reconstructing large-scale patterns of climate variability (since it is one aim of your study), e.g.: Brienen, R. J. W., Helle, G., Pons, T. L., Guyot, J.-L., Gloor, M., Oxygen isotopes in tree rings are a good proxy for Amazon precipitation and El Niño-Southern Oscillation variability, PNAS, (42) 16957-16962; DOI: 10.1073/pnas.1205977109, 2012 Lavergne, A., Daux, V., Villalba, R., Pierre, M., Stievenard, M., Srur, A. M., Vimeux, F., Are the δ18O of F. cupressoides and N. pumilio promising proxies for climate reconstructions in northern Patagonia?, J. Geophys. Res.-Biogeo., 121, 767–776, https://doi.org/10.1002/2015JG003260, 2016.*

R: The suggested references have been implemented into the revised manuscript [L53-54].

*L76-77: I am not sure what is the meaning of this sentence. Please rewrite. L75-80: I would suggest clearly rewriting this part as it is difficult to read. You should get right to the point: how are you going to achieve your goals? What are the main analyses you are going to perform to reach those goals?*

R: We agree with the reviewer that the last paragraph of the introduction was difficult to read. According to the comment of the reviewer, we have rewritten the entire paragraph and have highlighted the goals and methods to reach our goals [L71-77].

*L94-96: Please comment on the implication of only using latewood for oak but both early- and late- woods for the coniferous species. Are you suggesting that earlywood in the coniferous species is only derived from carbohydrates formed during the current year? Please rewrite the sentence accordingly.*

R: Basically, trees form their annual ring from current assimilates and reserves (starch and fats). The used carbohydrates come from a pool which, depending on the season and the rate of assimilation during the vegetation period, is in part clearly dominated by reserves from the previous year(s). Since the conifers of the isotope network are evergreen, the proportion of the reserves used for wood accumulation decreases rapidly at the beginning of the earlywood formation and is usually not as high at the beginning as in the case of the deciduous oak trees utilized here. Therefore, we don't suggest that earlywood is only derived from carbohydrates formed during the current year for coniferous. However, oak is a ring-porous tree species

known for having its earlywood growth almost completing before the leaves are fully green and net photosynthesis is positive. Hence, for this species it makes sense to skip earlywood as it is rather easy to distinguish from latewood. On the other hand, it is difficult to identify a clear boundary between the early- and latewood of conifers without technical means of quantitative wood anatomy. We have implemented our comment in a short version in the revised version of the manuscript [L95-97].

*L100-101: "Here" is repeated twice in the sentence. Furthermore, the sentence is not grammatically correct. Please be more careful!*

Erroneous repeating of the word "here" has been fixed as well as the sentence has been grammatically corrected [L101-102].

*L108-109: what is the COBE-SST2 dataset? Please describe it here. Also, which index of ENSO are you using to define El Niño/La Niña years?*

R: We agree with the reviewer that the used SST dataset should be explained more in detail [L114-119].and also the ENSO index is now presented in this subsection [L122-129]. In the revised version of manuscript, we have revised the entire section and have added arguments why we used the climate data [subsection 2.2].

*L114-116: What is a 'nudge model scenarios/simulation'? It is not clear why you choose this title for the section. I would recommend combining sections 2.2 and 2.3 instead. How using both δ18OP and δ18OSW will inform you about 'fractionation/ photosynthesis processes'? You will never get insights into the fractionation processes occurring during photosynthesis using only those two timeseries! Please clarify.*

R: We thank the reviewer for pointing out that the subsection a 'nudged model scenarios/simulation' was not well explained and also confusing. In general, the name of the chapter is given based on the method which produces $\delta^{18}O_P$ and $\delta^{18}O_{SW}$ (in our case the nudged model simulations from Butzin et al. (2014)). Our goal with the model output is to test the correlation between $\delta^{18}O_{cel}$ and modelled $\delta^{18}O_P/\delta^{18}O_{SW}$ to identify if the water, which is used within the photosynthesis processes, has a multi-seasonal isotopic signature. If yes (as shown in our plots), it is an explanation how $\delta^{18}O_{cel}$ is able to capture the ENSO signal. Based on the questions and suggestions of the reviewer, the subsection has been renamed and sustainably changed with a better explanation of the technique behind and an argumentation why we used the model output [subsection 2.3].

*L123-131: What is the difference between EOF and PCA? From my understanding of those analyses, EOF and PCA are really similar. Are you suggesting that EOF provides information about spatial patterns, while PCA gives information about temporal patterns? The whole paragraph is confusing (especially the filtering actually done to fulfil the North et al. (1982) rule), please rewrite.*

R: Overall the PCA and EOF technique are related, but differences exist and both abbreviations are often mixed up. Based on the comments of both reviewers, we have rewritten our explanation of the PCA and EOF technique. Furthermore, we have added a more detailed explanation of the North et al. (1982) rule [L153-170]. After this paragraph we will add a simple explanation about PCA and EOF analysis.

The concept of the Principal Component Analysis (PCA) was firstly described by Pearson (1902) and Hotteling (1935) and used for the first time by Lorenz (1959) for climatological research (Storch & Zwiers, 1999). The general aim of the PCA is to find a new set of axes which explains the most of the variability within the dataset. This is done by rotating the initial data onto axes which are orthogonal to each other (Schönwiese, 2013). For this purpose, a vector is necessary which indicates the direction of the new coordinate axis which is called eigenvector. This type of vector doesn't change its direction by a rotation. Therefore, the eigenvectors are used as a transformation matrix for the input high dimensional datasets onto the new axis. To indicate if a rotation maximizes the explained variance, every eigenvector has a corresponding eigenvalue. The corresponding eigenvalue is a kind of stretch factor for the eigenvectors. A huge eigenvalue indicates that the eigenvector has to be strongly stretched to map high variabilities within the dataset which can be explained with the new set of axes. Therefore, the eigenvalue ($\lambda$) is equal to the variance of the time series ($\vec{X}$) from matrix M which got rotated by the corresponding eigenvector ($\vec{e}$) (Equation 1).

$$Var(\langle \vec{X}, \vec{e} \rangle) = \lambda \qquad (1)$$

To find a first set of eigenvalues and eigenvectors, the data is rotated until an axis can be defined which explains the highest variance. Storch & Zwiers (1999) described this with the effort of minimizing $\epsilon_1$ respectively to maximize $Var(\langle \vec{X}, \vec{e}^1 \rangle)$ (Equation 2).

$$\epsilon_1 = Var(\vec{X}) - Var(\langle \vec{X}, \vec{e}^1 \rangle) \qquad (2)$$

Finally, the rotated data forms the first component. Like in a traditional coordinate system, it is possible to calculate a new axis which is orthogonal to the first one. Therefore, the second component is formed by an orthogonal rotation around the axis of the first component. The total number of components is given by the absolute number of time series. To compute the time series for the first component, the e.g., first value of all input time series is multiplied

with the individual eigenvector and afterwards, summed up over all time series for each year. This process forms the time series of a principal component which has the same temporal coverage as the input time series.

Especially a separate analysis of the eigenvectors of $\vec{X}$ is commonly used. This analysis is known as Empirical Orthogonal Functions (EOF) and the goal of it is to identify spatially coherent climate patterns which explain a significant part of the variance for a specific region (this is shown and used for example in Ionita et al. (2008)). Therefore, the largest part of the variance can be explained by the pattern of the leading EOF.

*L132-133: Why are you mentioning this here? It should be already stated in Section 2.1.*

R: It is mentioned here again to explain why we used the filling algorithm of Josse and Husson (2016).

*L133-140: How can you be sure that by using the gap fill method, you will not influence your results? Also, why would you need to fill in the gaps for 400 years knowing that your climate data only goes up to 1851?*

R: Since the used climate data is only available from 1851, the advantage of using the presented tree ring network is to go beyond this time scale and to introduce a new perspective on the observed relationship between $\delta^{18}O_{cel}$ and ENSO activity back in the past. If we want to go back in the past the filling algorithm is necessary because the temporal coverage of the $\delta^{18}O_{cel}$ records is different as described in the "Material and Method" section.

We agree with reviewer that the gap filling method needs to be better evaluated. In the revised version, we have added a new section in the methods [179-183] and in the results [L244-L254] which contextualizes the uncertainties of the used gap filling method.

*L141-146: What do you mean? The whole paragraph is pretty confusing and after reading it several times, I still do not get what you are really doing here.*

R: Thanks to reviewer for highlighting this point. We agree that our explanation of the composite maps has potential for improvements. In the revised version of the manuscript, we have rewritten and supplement our explanation of the composite maps [L184-190]. Furthermore, we have changed the corresponding figures to show the difference for high and low values of PC1 for SST and Z500 [Fig. 4 & 5].

*What kind of information is providing the geopotential height 500mb (Z500) for the analysis?*

R: The geopotential height is a standard variable in atmospheric sciences which is often represented in contour maps with isohyets. They connect places at the same altitude at which a certain air pressure (here 500 hPa which is on average 5500 meters high) prevails. This height is called geopotential in meteorology. With the differences in the geopotential height maps, it is possible to identify low- and high-pressure systems based on the different pressure patterns as for example shown in Figure 4. From the Z500 maps, the wind direction can be determined which is helpful for the understanding and the investigation of large-scale flows in the atmosphere. A big advantage of this variable is that the influence of friction at the surface and the influence of dynamic disruptive factors of the upper layers of the atmosphere, e.g. the Jetstream, are relatively minor.

*L151: 'Nino 3.4 index' this should come earlier in section 2.2 when you are presenting the environmental data used in the analyses.*

R: We agree with the reviewer and we have shifted the explanation of Nino 3.4 index to the explanation of the used climate data [L122-124].

*L158-160: The first sentence does not provide any clear information. Please remove.*

R: The highlighted sentence has been removed in the revised version of the manuscript.

*Figure 1: Could you add more information in the figure A related to the latitude of each site? The names and characteristics of the sites are not presented anywhere in the manuscript. Even though the data have already been published, a Table with sites information should be included. In figures B and C, how is it possible that R2 and p-value are exactly the same for the relationships between δ18Ocel and altitude and between δ18Ocel and latitude? I suspect that there is a mistake here.*

R: We agree with the reviewer that more information about the sample sites would be helpful. Therefore, we have added a table with the coordinates, tree type and altitude for each site in the revised version of the manuscript [Supp. 1]. Furthermore, the reviewer is right that there is a mistake within the textbox of Figure 2B (should be $R^2=0.417$ and p-value=2.17e^-04) of submitted manuscript. That has been fixed in the revised version of the manuscript [Fig. 2] .

*L160-161 and L172-173: Since no information about the exact location of the site presented in Figure 1A is provided, it is difficult to follow this statement.*

R: We agree with it. It will be easier to understand in the revised version of the manuscript, because of the added table which contains the characteristics of each sample site [Supp. 1].

*L162: I would remove 'This might be determined genetically,' from the sentence as it is not completely accurate (different species of Quercus also have different genetic information).*

R: We thank the reviewer for that suggestion, and we have removed that part.

*L165: You could also cite more updated papers describing differences between an giosperms and gymnosperms, e.g.: Carnicer J., Barbeta A., Sperlich D., Coll M., Penuelas J., Contrasting trait syndromes in angiosperms and conifers are associated with different responses of tree growth to temperature on a large scale, Front. Plant Sci., 4, 409, https://doi.org/10.3389/fpls.2013.00409, 2013*

R: The mentioned reference has been integrated in the revised version of manuscript [L. 209-210].

*L177: 'which could influence the relation by a latitudinal effect.' please rewrite as 'thus the latitudinal and altitudinal gradients may have confounding effects on δ18Ocel' or something similar*

R: Thanks to reviewer for that comment. We have rewritten this sentence according to the suggestion of the reviewer [L220-221].

*L177-179: I would rewrite this sentence as the effects of the two gradients on δ18Ocel have already been observed and documented in many other studies, for instance: Szejner, P., W. E. Wright, F. Babst, S. Belmecheri, V. Trouet, S. W. Leavitt, J. R. Ehleringer, R. K. Monson, Latitudinal gradients in tree ring stable carbon and oxygen isotopes reveal differential climate influences of the North American Monsoon System, J. Geo- phys. Res. Biogeosci., 121, 1978–1991, doi:10.1002/2016JG003460. 2016*

R: We have rewritten the sentence [L221-222] and have added a new subsection to discuss the effects of latitude and altitude $\delta^{18}O_{cel}$ [subsection 4.1].

*L181-186: Here again comes the confusion between EOF and PCA. You should clarify from the beginning (see previous comment) what is the difference between the two especially given that EOF1 and PC1 both seem to explain 16.2% of the variance in the records.*

R: We see the point that the differences between EOF and PCA is not well explained and can be confusing in the submitted manuscript. With the new explanation [L153-170], it will be easier to understand the differences.

*Figure 4: In the legend, you are describing the columns not the rows.*

R: Thank you for the comment. Since we have added new figures, the figure captions have been updated [Fig. 4].

*L216 Is the distribution of PC1 for El Niño (or La Niña) years significantly different from that during normal years (i.e. when excluding El Niño/La Niña years)?*

R: Yes. This is shown in Supp.3. To make it easier to understand, we have rewritten this part in the revised version of the manuscript [L286-290].

*L222-224: Please rewrite the sentence. As it reads now, it looks like you are saying that Europe is characterized by higher precipitation and lower air surface temperatures in summer! And it is not clear what the parentheses apply for.*

R: We agree with the reviewer that the explanation of the composite maps can be confusing. The updated explanation of the composite maps [L184-190] and the new figures [Fig. 4,5,6,8,9] in the revised version of the manuscript will make it easier to follow and to understand our argumentation.

*L230-231: 'because we to take into account. . .' Why would SPEI3 index accounting for the climate conditions prevailing over the previous season? So far, nothing has been said about this dataset.*

*L227-233: this part mostly belongs to the Material and Methods section and could be improved for readability.*

R: We have added more details about the SPEI3 dataset within the Section 2.2 [L110-113] and have removed the explanation from the results subsection.

*L235: 'the used reconstruction': which one?*

R: Thanks to the reviewer for highlighting the point that it is not clear which reconstruction was used. We have rewritten the sentence and have added the reference of the used reconstructions [L294-297].

*L240: 'to capture a multi-seasonal signal' what do you mean?*

R: We agree with the reviewer that this part is not easy to understand. We have revised that part [L299-300] and have added further information about the goal and method of nudged model simulations [subsection 2.3].

*L243-244: where is it shown?*

R: This is shown in Figure 7 of the revised manuscript.

*L244-245: so why then δ18Ocel is not more strongly related to δ18Osw? Your argument is contradictory with what is actually described.*

R: We thank the reviewer for the helpful comment! The referred sentence is difficult to understand and we have rewritten it [L304-305].

*L239-248: And so what? What are you really trying to say here? Also, I do not think the results are properly discussed and compared to the literature.*

R: We agree with the reviewer that the results should be compared to literature and discussed more thoroughly. In the revised version of the manuscript, we have added a subsection to discuss the winter climate signal in $\delta^{18}O_{cel}$.Furthermore, we have highlighted our implications of our results [subsection 4.4].

*Figure 6: You mean the upper row Why description of Figure 6 comes before Figure 5?*

R: Thank you for the suggestion. Yes, we do mean the upper row; we have changed the figure caption accordingly [caption of Fig. 7]. Furthermore, we have changed the order of the figures.

*L168-169 and L276: Please rewrite sentences*

R: We have rewritten the mentioned sentences[L210-212 ,L444-445].

*L236-237: The instability of the relationship between climate variables and ENSO has also been documented by other tree-ring studies in southern South America, e.g: Álvarez, C., Veblen, T.T., Christie, D.A., González-Reyes, Á., Relationships between climate variability and radial growth of Nothofagus pumilio near altitudinal treeline in the Andes of northern Patagonia, Chile. For. Ecol. Manage. 342, 112–121, 2015*

R: The suggested reference has been implemented in the revised version of the manuscript [L371]. Furthermore, we have added a subsection to discuss the stability of the ENSO signal [subsection 4.3].

**Anonymous referee #2**

*Summary*

*The authors investigate delta18O tree ring records of 26 site distributed over Europe for the last 400 years. They claim that they were able to identify a connection of the leading mode of variability of this data to El Nino Southern Oscillation. They speculate that this connection is only found in the last 130 years and thus the connection is not stable in time. The second mode of the data is suggested to reconstruct regional summer atmospheric circulation. Finally, the author team claim that delta18O tree ring records can be used to reconstruct atmospheric circulation.*

*General comments*

*The topic would certainly be of high interests. However, the authors fail to convincingly show evidence for their main claims listed in the summary. Therefore, I recommend to reject the manuscript.*

*Major comments*

*1. At several places in the manuscript the authors claim that their analysis suggests "the relationship between ENSO and the European climate may not stable over time". The connection is only found for the instrumental period after 1850 CE. I think the first order interpretation is that the reconstruction of ENSO might be not perfect, as normally the reconstruction methods are trained in the last 100 to 150 years. So differences between the training periods and the period before are a hint that the reconstruction might be not successful. So, from your analysis you cannot conclude that you have identified non stationarity of teleconnections.*

R: R: We thank the reviewer for this comment. It stresses that we have to put further work into the manuscript clarifying our results and interpretations as we are not intending to say that we have identified a non-stationarity teleconnection. We only say that the correlations and coincidence rates are weakening, which indicates that the relationship between PC1 of the δ18Ocel network and ENSO might not be stable over time. Therefore, we can only suggest that the relationship between ENSO and the European climate may be not stable over time which is supported also by other studies based on proxy data (e.g., Rimbu et al., 2003). The idea of a unstable relationship is also supported by the results from studies based on instrumental data (e.g. Fraedrich, 1994; Fraedrich and Müller, 1992; Pozo-Vázquez et al., 2005) or studies based on ocean-atmosphere coupled models (e.g. Raible et al., 2004; Deser et al., 2006; Brönnimann et al., 2007) The fact that the relationship between climate variables

and ENSO is not stable over time has been also recognized in tree-ring studies in other regions, e.g. South Amerika (Álvarez et al., 2015). Nevertheless, we are aware of the fact that further research is necessary to make a confident statement.

It is true that reconstructed ENSO indices before instrumental period are not perfect which could be the cause of decrease in the correlations between our PC1 and these indices before 1850s. However, it is also true, that 1850s represent the end of Little Ice Age (LIA) period, when ENSO properties and its teleconnections changed significantly. Modeling studies (e.g., Henke et al., 2017) report an increased frequency of El Nino during LIA due to southern displacement of ITCZ. Although the reviewer hypothesis could be true, also our interpretation that decreasing in the correlation between our PC1 and reconstructed ENSO indices is due to changes in ENSO teleconnections over Europe could be true.

We agree with the reviewer that a comparison with ENSO reconstruction which are trained within the last 150 years can be problematic, since every reconstruction has its own uncertainties and limits. Confident statements are only possible for the time from 1850 onwards, since instrumental measurements of different climate variables are available. However, the usage of climate reconstructions is the only possibility to test and analyze the teleconnection before 1850 because no observational data is available. Overall, we tested the relationship with three different reconstructions for two different time ranges, where the sample density of isotope network is relatively high and the correlation is getting weaker over time (Li et al., 2011; Li et al., 2013; Dätwyler et al., 2019) which is shown in Table 1. Not only the correlation between the first component of the $\delta^{18}O_{cel}$ network and the ENSO reconstructions is getting weaker, also the correlation between different ENSO reconstructions is getting weaker in the 18th century which was shown for specific periods in Dätwyler et al. (2019). They suggest that is based on changes of the ENSO teleconnections, because they found a consistent teleconnection pattern that is different to the known teleconnection pattern of ENSO in the instrumental period (Dätwyler et al., 2019).

|  | LI ET AL. 2011 | LI ET AL. 2013 | DÄTWYLER ET AL. 2019 |
|---|---|---|---|
| **1750-1849** | r=0.121 p-value=0.231 | r=-0.008 p-value=0.936 | r=-0.078 p-value=0.442 |
| **1850-1949** | r=0.223 p-value=0.026 | r=0.303 p-value=0.002 | r=0.296 p-value=0.003 |

Table 1: Correlation between the first component of the $\delta^{18}O_{cel}$ network with three different ENSO reconstructions for two different time periods.

In order to show how stable the connection between the sea surface temperature (SST) of the tropics and the first mode of $\delta^{18}O_{cel}$ is, we have computed stability maps of the correlation between these two quantities. The stability map is a tool which is primarily used for streamflow predictions to identify stable teleconnection (for more details see Ionita et al. (2008), Lohmann et al. (2005) or Rimbu et al. (2005)). SST anomalies from the Extended Reconstructed Sea Surface Temperature (ERSST) v5 dataset (Huang et al., 2017) have been correlated with the first mode of $\delta^{18}O_{cel}$ in a moving window of 21 years. The correlation is considered to be stable for those grid points where anomalies are significantly correlated at the 90% level ($r = 0.25$) for more than 80% of the 21-yr windows, covering the period 1850–2005. This is shown in Figure 1. The results from the stability maps analysis will be also integrated in the revised version of the manuscript.

[Figure]

Figure 1: Stability maps for the correlation between SST anomalies (Huang et al., 2017) and the first mode of $\delta^{18}O_{cel}$. The color bar indicates how many years are characterized by a significant correlation (at the 90% level) in a 21-yr window, covering the period 1850–2005. Positive correlations are shown from yellowish to reddish and negative correlations from greenish to blueish.

As it can be inferred from Figure 1 (below), one of the largest locations with stable correlations is located in the tropical Pacific. Stable correlations are shown from September (last year) to June which also supports our result that $\delta^{18}O_{cel}$ is able to capture a multi-seasonal signal and that the first mode of $\delta^{18}O_{cel}$ is sensitive for ENSO variability. With the help of these maps, it is also possible to track the development of the correlation over time. During July and August, the pattern dissolves in the tropical Pacific whereas a stable negative correlation is shown around Europe. Based on these results, we suggest that the relation between the first mode of $\delta^{18}O_{cel}$ and the SST in the tropical Pacific/Atlantic in winter and spring is stable in the period from 1850 to 2005.

Based on the suggestions and critical comments of reviewer #2, we have added a subsection to discuss the stability of the link to ENSO variability [subsection 4.3]. Furthermore, we have highlighted the comparison to other ENSO reconstructions and uncertainties especially given by ENSO reconstructions in the new subsection of the revised manuscript.

Furthermore, we have checked carefully the usage of the term "non-stationarity" in the revised manuscript. We also tried to highlight that we can only suggest that the relationship between ENSO and the European climate may be not stable over time. Finally, we would like to thank reviewer #2 for the critical evaluation.

2. The conclusion that the analysis shows that "We infer that the investigation of large-scale atmospheric circulation patterns and related teleconnections far beyond instrumental records can be done with oxygen isotopic signature derived from tree rings." is not convincingly demonstrated. There is only 2 line in the introduction which gives a hint why this should be possible, i.e., fractionation happens during the transport from source to sink areas. Most of the studies however try to reconstruct temperature and precipitation when using delta18O as delta18O is first order temperature dependent. The authors also nicely discuss that fractionation processes are also relevant within the tree. Then, at the different sites the water can be transported form different source regions during the seasonal cycle, e.g., North Atlantic versus Mediterranean, or long-distance transport versus local water recycling. Moreover, seasonality plays an important role, so mostly tree ring records are interpreted to record growing season signals and not winter signals. So, given all these uncertainties how can the transport aspect (which is related to the atmospheric circulation) survive?

R: We agree with the reviewer that transport can lead to uncertainties. The earth system is interconnected and includes many factors that make it hard to identify a single cause. Nevertheless, we think that the aforementioned statement of the reviewer is somehow mixes up several points. First of all, the source of oxygen isotopes in cellulose is mostly the water from the atmosphere. Through precipitation, the soil gets enriched with water and

depending on the length and depth of the root system, a tree uses surface water or water from groundwater reservoirs for photosynthesis processes. Therefore, $\delta^{18}O_{cel}$ is coupled to the hydrological cycle on Earth. This cycle is strongly dependent on large-scale atmospheric circulation, since large-scale flows determine the path of clouds, precipitation patterns and the distribution of water vapor in the atmosphere. Also, the mentioned water transport from different source regions is related to large-scale atmospheric circulation. Synoptic patterns and associated indices (e.g., cyclone and anticyclone activity, air pressure) have also been reconstructed from $\delta^{18}O_{cel}$, with stronger correlations revealed during extreme years or periods (e.g., Saurer et al., 2012). $\delta^{18}O_{cel}$ is related to the $\delta^{18}O$ of the precipitation source via soil water. $\delta^{18}O$ of soil water constitutes the $\delta^{18}O$ input to the arboreal system and represents an average $\delta^{18}O$ over several precipitation events modified by partial evaporation from the soil (depending on soil texture and porosity) and by a possible time lag, depending on rooting depth (Saurer et al., 2012). $\delta^{18}O_{cel}$ is further dependent on two tree-internal processes: evaporative $^{18}O$-enrichment of leaf or needle water via transpiration, as well as biochemical fractionations and isotopic exchange of $\delta^{18}O$ with trunk water during cellulose biosynthesis (e.g., Barbour, 2007; Kahmen et al., 2011; Roden et al., 2000; Saurer et al., 2012; Treydte et al., 2014). Fractionations occurring at leaf level, are partly reset by isotopic exchange between sugar oxygen and stem water during cellulose synthesis in the trunk allowing the soil water isotopic signal to be largely preserved in the tree-ring cellulose (e.g., Gessler et al., 2014 and citations therein).

The seasonality is essential since trees store the climate signal within the $\delta^{18}O_{cel}$ ratio during the growing season. One of the key messages of the manuscript is that winter climate signals can be stored which is shown by significant correlation plots [Figure 7]. We argue that this is possible through hydrological feedback processes (e.g., via the soil moisture content). The fact that trees are able to store a winter signal is not new and was published for example by Treydte et al. (2006) and Treydte et al. (2014).

With the knowledge about the physical climate processes (the hydrological cycle) and the understanding of the theory behind the fractionation of $\delta^{18}O_{cel}$ ratio, we feel that the points mentioned by the reviewer are not uncertainties. They are part of the climate system and essential for the understanding of climate proxies.

We thank the reviewer for the critical comment, which gave us some hints for the discussion of the $\delta^{18}O_{cel}$ climate signal [subsection 4.4 & 4.5].

3. For the first EOF I have a different interpretation, which takes into account the fact that temperature play the dominant role in delta18O. What we see is a monopole structure. The authors claim to see a link to ENSO. I hypothesize that the link is simply due to the fact that

ENSO has a global impact on the global mean temperature. So, due to an El Nino event, the Earth warms and thus also the North Atlantic and the Mediterranean (visible in the composite plots). Warmer source regions affect the fractionation of delta18O without any change of the circulation we see in the sink regions (at the tree sites) a uniform signal.

R: The climate over the European region has a rather peculiar variability, and in most of the cases the first EOF is monopolar, especially if we consider mostly the central and western part of Europe in the analysis (like in our isotope network). This is valid for precipitation, temperature and even for drought indices. As we discussed in the first point raised by the reviewer, we disagree that the relationship with ENSO is purely by chance given the tests of statistical significance. The fact that PC1 correlates significantly with the SST from the tropical Pacific in winter and this signal is transmitted to the tropical Atlantic in spring and central Atlantic in summer is part of the natural cycle of ENSO. The ENSO anomalies (either El Niño or La Niña) develop in winter and it needs 3-6 months to see a signal in the European climate. This lagged relationship is typical for many ENSO related teleconnections. This long transition from the tropical Pacific to central North Atlantic affects in turn the large-scale atmospheric circulation and as a consequence the climate over Europe, especially in spring and summer. We do not agree with the idea that just because we have El Niño or La Niña the earth will be either warm or colder. ENSO dynamics are more complicated than this and the signal from ENSO to the $\delta^{18}O$ is transmitted mainly via the large-scale atmospheric circulation.

In the composite maps, the ENSO teleconnection patterns are clearly emphasized. This means that there is not only a thermodynamical influence on $\delta^{18}O$, that is variation of global temperature with ENSO, playing a role, but also ENSO related teleconnections impact on European climate is important.

4. There are problems with the data (see comment below Section 2.2, L132-140, L145) ignored which might be influential to the analysis.

R: In the revised version of the manuscript, we have sustainably changed the data section [subsection 2.1 - 2.3]. We have implemented a better description of the used isotope network and a detailed description of the climate data. Please see our detailed answers and argumentation to the aforementioned comment below.

Minor comments
L18: What is meant by "reflects a multi-seasonal climatic signal."? ENSO works on timescales of 3 to 7 years.

R: Quote from the submitted manuscript (L17-18): "The first mode of $\delta^{18}$O variability is associated with anomaly patterns of the El Niño-Southern Oscillation (ENSO) and reflects a multi-seasonal climatic signal". So, the last part of the sentence is connected to the first mode of $\delta^{18}$O which means that the first mode of $\delta^{18}$O variability reflects a multi-seasonal climatic signal. The multi-seasonal climatic signal stored in $\delta^{18}$O is essential to capture El Niño/La Niña events and the related ENSO variability.

L20: "out of phase variability": I would interpret this in the time and not in space as the authors. Just say the 2. EOF is a dipole pattern with centers over northwestern and southeastern Europe.

R: Thank you for the suggestion. We have rewritten it in the revised version of the manuscript [L20].

L47-50: Hard to read.

R: We have revised the mentioned section [L45-48].

L53: please change to "leaf water clearly affects"

R: We have changed the sentence according to the suggestion of the reviewer [L50].

L55: I disagree with this statement, see major comments 2 and 3.

R: In this point, we disagree with the reviewer. Please have a look above for our argumentation.

L56: What is meant by "resulting long-term perspective"? Where does it result from?

R: Thanks to the reviewer for highlighting this. We mean the long-term perspective that results from the usage of $\delta^{18}O_{cel}$ as a climate proxy. We have rewritten the sentence to make it clearer [L54-56].

L84: Created -> generated

R: We have changed it [L81].

L88: Please include a space between number and unit throughout the manuscript.

R: We have improved it in the revised version of the manuscript [L86].

L91: What is SMOW?

R: It should be VSMOW (Vienna Standard Mean Ocean Water). We have changed it [L89].

Section 2.1: Which method is used to get the delta18O samples from trees. This is relevant as studies show that the method (pooling or not pooling) makes a huge difference Hangartner et al. Methods to merge overlapping tree-ring isotope series to generate multi-centennial chronologies CHEMICAL GEOLOGY Volume: âAR 294 Pages: âAR 127-134 Published: âAR FEB 10 2012

R: We agree with the reviewer that information about the sampling method is very important. According to Treydte et. al (2007a, b), all tree rings from the same year were pooled prior to cellulose extraction for the majority of sites. We have added this in Section 2.1 in the revised version of the manuscript [L92-93].

Section 2.2: Again if is unclear what the authors are using. Is it the ensemble mean of 20CR or an individual ensemble member. Please note that the 20CR is only constraint with sea level pressure (SLP) data so no sea surface temperatures (SST) are used which are relevant for ENSO. My guess is that the authors use the ensemble mean. This is problematic as in the early part of the reanalysis the constraint (via SLP) is rather weak leading variance deflation and thus can have a strong impact on the analysis (so it is normally recommended to use all individual ensemble members). As said, the other problem is that ENSO might not be realistically included in the first part of the reanalysis.

R: The argumentation that 20CR (V2c) is not using SST is not correct since the SST dataset from 18 members of Simple Ocean Data Assimilation with Sparse Input version 2 (SODAsi.2, Giese et al., 2016) are used as SST boundary condition (the high latitudes (>60°) were corrected to COBE-SST2 (Hirahara et al. 2014)). Furthermore, the sea ice cover (SIC) reconstruction of Hirahara et al. (2014) is also used as boundary condition (for more details see https://www.psl.noaa.gov/data/gridded/data.20thC_ReanV2c.html). Therefore, the SST and SIC are influencing the atmosphere (Compo et al, 2011) and therefore, ENSO activity is represented in 20CR (V2c).

Nevertheless, we have tried to explain the used climate data more in detail in the revised version of the manuscript. For this purpose, most of the climate data subsection have been rewritten. Additionally, we have added climate data from other sources to get more confidence for our results and to convince reviewer #2 of the correctness of our results [subsection 2.2].

Section 2.3: too short and not clear why the simulations are used and how the simulations are generated. The reader needs to understand which model is used and how, just references is not enough.

R: We thank the reviewer for pointing out that the subsection a 'nudged model scenarios/simulation' was not well explained and also confusing. The subsection has been renamed and sustainably changed with a better explanation of the technique behind and an argumentation why we used the model output [subsection 3.3].

Section 2.4: It reads like EOF and PCA are different analysis, but actually they are not. The method of empirical orthogonal function (EOF) analysis is a decomposition of a data set in terms of orthogonal basis functions determined from this data. Thus it is the same as geographically weighted PCAs.

Overall, the PCA and EOF technique are related, but differences exist and both abbreviations are often mixed up. For detailed explanation of the differences, we refer to our response to reviewer #1 (*L123-131*) .Based on the comments of both reviewers, we have rewritten our explanation of the PCA and EOF technique. Furthermore, we have added a more detailed explanation of the North et al. (1982) rule [L153-170].

L132-140: How many tree ring records cover the entire period with no gaps? How sensitive it the analysis to filling the gaps? How many cycles are needed to reach convergence? What if you use only the tree ring records which cover the entire period?

R: Twelve time series cover the entire period. Since, our goal is to use all time series of the ISONET+ product and therefore, we have to fill the gaps. If we calculate the EOFs and PCs for the period 1850 to 1998 the spatial and temporal correlation to the EOFs and PCs for the entire period is very high. If we remove samples in the period 1850 to 1998, also the PCs and EOFs are changing because the input variability of the network is changing. The results are similar patterns, but the temporal and spatial correlation is varying. For a comparison of the climate signal of the used $\delta^{18}O_{cel}$ network, the reader is referred to the publication of the used isotope network of Treydte et al (2007b). Based on the comments of both reviewers, we have added a new section in the methods [179-183] and in the results [L244-L254] which contextualizes the used gap filling method and the uncertainties. Furthermore, we have added a table in the supplement to describe the characteristics of each time series and sample site [Supp. 1].

L142: This is certainly not extreme.

R: The detection of extremes by using the standard deviation is often used in climate science. There are many other definitions of extremes. We decided to use the standard deviation as an approximation given the available degrees of freedom. These extremes may not be extremely rare but certainly unusual and different from the mean.

L145: I would say that the authors misuse the composite analysis by focusing on the linear response. If they would like to analyze the linear response, a simple correlation analysis is enough. The beauty of the composite analysis is that you can easily show non-linear effects, but only if you make the difference between the mean above the threshold with the long term mean and in a second plot the mean below the negative threshold and the long term mean. This was done by Fraedrich 1992 mentioned in the manuscript. He highlighted the nonlinearity of the ENSO response over Europe and thus is in contradiction to the linear relationship suggested here.

R: We think that this point is a misunderstanding of our idea because we used the composite maps to learn more about the linear response. It is not our aim to test relation for non-linearities. The reference of Fraedrich (1992) is used to highlight the uncertainties and show the reader that the shown relationship in the results cannot be seen as linear. In the revised version of the manuscript, we have replaced the figures with new figures in which we show both the high (PC1 > 1 standard deviation) and low (PC1 < -1 standard deviation) composite maps for SST and Z500 [Fig. 4 & 5]. There are cases in which the high and low composite maps show different results, thus indicating non-linearity between the variables analyzed, but in our case the high and low composite maps are similar (in terms of structure), that is why we had shown just the difference map.

L150: Event Coincidence Analysis needs to be explained.

R: Thank you for the comment. We have extended the explanation of the method [L192-199].

L177-179: If this is the case one could speculate that EOF2 showing a North South patterns just resemble this latitudinal effect.

R: In this comment, two essentials were mixed up. First of all, the latitudinal effect was found by Daansgard (1964) and is defined as a steady shift of the isotope values from the equator to the poles. The reason for this is that temperature changes within different seasons and varies over the latitudes. According to Daansgard (1964) and Gat (2010), the mean annual temperature and the $\delta^{18}O$ in the atmosphere are strongly correlated. For a documentation of several effects on the $\delta^{18}O$ ratio, e.g., the continental effect, we refer to the publication of Gat (2010).

The latitudinal effect is important for the $\delta^{18}O_{cel}$ ratio at each site (Figure 1 of the submitted manuscript), because it is one factor which influences the source values ($\delta^{18}O_{precipitation}$, $\delta^{18}O_{soilwater}$). In contrast, the latitudinal effect cannot explain the (climate) variability which is investigated and represented by the PCA and EOF technique in our study. The climate variability is determined by several other quantities which cannot be explained by a simple latitudinal position, e.g., large-scale atmospheric flows. Based on that argumentation, we disagree with the speculation of the reviewer.

L195: I do not see this is there a typo and the authors mean PC1?

R: Thank you for the comment. We have revised the sentence [L243-244].

L200-210: Avoid using the bracket with e.g. (cold). This makes the text unreadable. Just say what you show in Fig. 4.

R: This is the standard way how composite maps are described. It is necessary to add the brackets since both tails of the distribution function are combined and represented in these maps. Therefore, we have to write this additional information to clearly represent both extremes. To make it more understandable, we have added a more detailed explanation of how composite maps are working [L184-190].

L208: I do not see a AO pattern, again the reference to Fraedrich are incorrect as they claim that ENSO has a nonlinear response behavior over Europe.

R: We thank the reviewer for this comment. It is true that there is some asymmetry between El Niño and La Niña teleconnection patterns over Europe. However, the analysis of long-term data reveals that El Niño (La Niña) conditions in the tropical Pacific are associated with a negative (positive) phase of the North Atlantic Oscillation in the North Atlantic region during winter. The composite pattern represented in the figure reflects this ENSO-NAO relationship, although the pattern is not identical with the AO/NAO. In the revised version, we have written more carefully about the similarities to the NAO pattern [L345-351] and have removed the comparison to the AO pattern.

L225: Why drought we see a positive precipitation anomaly?

R: We think that your comment is a misunderstanding of our composite map. As mentioned above both extremes of the distribution are represented by the composite map. We have added a more detailed description of the composite maps for a better understanding [L184-190].

Section3.4: What do we learn from this? What is shown and why? This section is unclear and to my feeling can be removed.

R: Our goal with the model output is to test the correlation between $\delta^{18}O_{cel}$ and $\delta^{18}O_P/\delta^{18}O_{SW}$ to identify if the water, which is used during the photosynthesis processes, has a multi-seasonal isotopic signature. We agree with the reviewer that it is not clearly explained in the manuscript. Therefore, we have added an extended explanation of the used dataset [subsection 2.3] and the results in the revised version of the manuscript [subsection 3.4]. Furthermore, we have added a subsection to discuss the winter climate signal in $\delta^{18}O_{cel}$ [subsection 4.4].

L250: section 3.5

R: We have corrected it in the revised version of the manuscript [L307].

L251-252: This sentence is a repetition.

R: We have removed the sentence in the revised version of the manuscript.

L260 -263: You need to show this with more proxies. Note that dry conditions are not droughts!

R: Thank you for the comment. We have been very sensitive with the usage of the term "drought" in the revised version of the manuscript. Furthermore, the comparison with more proxies is a really good idea. Based on the complexity of this topic, this would be another big study and a topic for the future work in this field. In our study, we want to show primarily the results of the analysis of the $\delta^{18}O_{cel}$ isotope network.

L263-265: Given your study you cannot conclude this. The authors study certainly is inadequate to reconstruct blocking so this statement is not supported by the authors analysis.

R: We agree with the reviewer that this part was not well discussed and explained. In the revised version of the manuscript, we have added a new subsection to discuss the relation to summer atmospheric blocking [subsection 4.5].

L269-70: Please change to ". . . signal still dominates".

R: We have rewritten it [L437].

Figures: Statistical significance is not tested (or not shown).

R: In the revised version of the manuscript, we have replaced the former figures with figures which are also showing the significance [Fig. 4,5,6,8 & 9].

**References used in our responses to both anonymous referees:**

Álvarez, C., Veblen, T. T., Christie, D. and González-Reyes, Á.: Relationships between climate variability and radial growth of Nothofagus pumilio near altitudinal treeline in the Andes of northern Patagonia, Chile, , doi:10.1016/J.FORECO.2015.01.018, 2015.

Barbour, M. M.: Stable oxygen isotope composition of plant tissue: a review, Funct Plant Biol, 34, 83-94, 2007.

Butzin, M., Werner, M., Masson-Delmotte, V., Risi, C., Frankenberg, C., Gribanov, K., Jouzel, J. and Zakharov, V. I.: Variations of oxygen-18 in West Siberian precipitation during the last 50 years, Atmospheric Chemistry and Physics, 14(11), 5853–5869, doi:10.5194/acp-14-5853-2014, 2014.

Brönnimann, S., Xoplaki, E., Casty, C., Pauling, A. and Luterbacher, J.: ENSO influence on Europe during the last centuries, Climate Dynamics, 28(2–3), 181–197, doi:10.1007/s00382-006-0175-z, 2007.

Compo, G. P., Whitaker, J. S., Sardeshmukh, P. D., Matsui, N., Allan, R. J., Yin, X., Gleason, B. E., Vose, R. S., Rutledge, G., Bessemoulin, P., Brönnimann, S., Brunet, M., Crouthamel, R. I., Grant, A. N., Groisman, P. Y., Jones, P. D., Kruk, M. C., Kruger, A. C., Marshall, G. J., Maugeri, M., Mok, H. Y., Nordli, ø., Ross, T. F., Trigo, R. M., Wang, X. L., Woodruff, S. D. and Worley, S. J.: The Twentieth Century Reanalysis Project, Quarterly Journal of the Royal Meteorological Society, 137(654), 1–28, doi:10.1002/qj.776, 2011.

Dansgaard, W.: Stable isotopes in precipitation, Tellus, 16(4), 436–468, doi:10.1111/j.2153-3490.1964.tb00181.x, 1964.

Dätwyler, C., Abram, N. J., Grosjean, M., Wahl, E. R. and Neukom, R.: El Niño–Southern Oscillation variability, teleconnection changes and responses to large volcanic eruptions since AD 1000, International Journal of Climatology, 39(5), 2711–2724, doi:10.1002/joc.5983, 2019.

Deser, C., Capotondi, A., Saravanan, R. and Phillips, A. S.: Tropical Pacific and Atlantic Climate Variability in CCSM3, , doi:10.1175/JCLI3759.1, 2006.

Fraedrich, K.: An ENSO impact on Europe?, Tellus A, 46(4), 541–552, doi:10.1034/j.1600-0870.1994.00015.x, 1994.

Fraedrich, K. and Müller, K.: Climate anomalies in Europe associated with ENSO extremes, International Journal of Climatology, 12(1), 25–31, doi:10.1002/joc.3370120104, 1992.

Gat, J.: Isotope Hydrology: A Study of the Water Cycle, World Scientific, London., 2010.

Gessler, A., Ferrio, J. P., Hommel, R., Treydte, K., Werner, R. A., and Monson, R. K.: Stable isotopes in tree rings: towards a mechanistic understanding of isotope fractionation and mixing processes from the leaves to the wood, Tree Physiology, 34, 796-818, 2014.

Henke LMK 2017, Was Little Ice Age more or less El Nino like than Medieval Climate anomaly? Climate of the past, doi: 10.5194/cp-13-267-2017

Hirahara, S., Ishii, M. and Fukuda, Y.: Centennial-Scale Sea Surface Temperature Analysis and Its Uncertainty, Journal of Climate, 27(1), 57–75, doi:10.1175/JCLI-D-12-00837.1, 2014.

Hotelling, H.: The most predictable criterion, Journal of Educational Psychology, 26(2), 139–142, doi:10.1037/h0058165, 1935.

Huang, B., Thorne, P. W., Banzon, V. F., Boyer, T., Chepurin, G., Lawrimore, J. H., Menne, M. J., Smith, T. M., Vose, R. S. and Zhang, H.-M.: NOAA Extended Reconstructed Sea Surface Temperature (ERSST), Version 5, doi:10.7289/V5T72FNM, 2017.

Ionita, M., Lohmann, G. and Rimbu, N.: Prediction of Spring Elbe Discharge Based on Stable Teleconnections with Winter Global Temperature and Precipitation, J. Climate, 21(23), 6215–6226, doi:10.1175/2008JCLI2248.1, 2008.

Josse, J. and Husson, F.: missMDA : A Package for Handling Missing Values in Multivariate Data Analysis, Journal of Statistical Software, 70(1), doi:10.18637/jss.v070.i01, 2016.

Kahmen, A., Sachse, D., Arndt, S. K., Tu, K. P., Farrington, H., Vitousek, P. M., and Dawson, T. E.: Cellulose delta O-18 is an index of leaf-to-air vapor pressure difference (VPD) in tropical plants, Proceedings of the National Academy of Sciences of the United States of America, 108, 1981-1986, 2011.

Li, J., Xie, S.-P., Cook, E. R., Huang, G., D'Arrigo, R., Liu, F., Ma, J. and Zheng, X.-T.: Interdecadal modulation of El Niño amplitude during the past millennium, Nature Climate Change, 1(2), 114–118, doi:10.1038/nclimate1086, 2011.

Li, J., Xie, S.-P., Cook, E. R., Morales, M. S., Christie, D. A., Johnson, N. C., Chen, F., D'Arrigo, R., Fowler, A. M., Gou, X. and Fang, K.: El Niño modulations over the past seven centuries, Nature Climate Change, 3(9), 822–826, doi:10.1038/nclimate1936, 2013.

Lohmann, G., Rimbu, N. and Dima, M.: Where can the Arctic oscillation be reconstructed? Towards a reconstruction of climate modes based on stable teleconnections, Climate of the Past Discussions, 1(1), 17–56, 2005.

Lorenz, E. N.: Prospects for statistical weather forecasting: final report: Statistical Forecasting Project, Massachusetts Institute of Technology, Dept. of Meteorology, Cambridge, Mass. [online] Available from: https://catalog.hathitrust.org/Record/007185933 (Accessed 16 November 2018), 1959.

[revised manuscript text omitted]

---

## Author Response (AR2)

Dear Editor,

Please find below our point-by-point response to your comments and the comments of the reviewers (*reviewer*, response). We thank both anonymous referees for the time and effort in reviewing our manuscript (cp-2020-39) especially in these challenging times. The comments, suggestions and feedback raised in the second round of the review process are highly appreciated as they help us to clarify our statements and to improve the quality of our manuscript. In our response, we have highlighted the relevant changes made in the manuscript with line numbers. Furthermore, we have uploaded a marked-up manuscript version.

On behalf of all co-authors, I would like to thank the Editor.

Best regards,

Daniel Balting

**Editor**

*Comments to the Author:*

*Dear authors,*

*Thank you for revising your manuscript and for addressing most of the concerns that were originally brought up after the first round of reviews. As you can see from this second round of comments, there are outstanding issues that need further explanation in your final article. I consider those to be relatively minor changes.*

*In particular, please address the following:*

*1- make sure to clarify language when describing the EOF and double-check for any typos in the formula*

*2- further discuss and nuance the potential meteorological/climatological scenarios that can lead to the d18O signatures you obtained (see comments 2 and 4 from reviewer 1)*

*3- further discuss the issues related to pooling of trees, and also why 4 trees (2 increment cores) is a justifiable/sufficient number of samples for your analysis*

*4- read through the specific comments made by the reviewers and address them to the best of your ability*

*Thanks,*

*Julie Loisel*

R: Dear Dr. Loisel, thank you for your helpful comments and recommendations. We have tried to address your above-mentioned points in our revised manuscript and in our response to the reviewers.

1: We have replaced the word "separate analysis" with "corresponding analysis" and corrected the typo in the formula [L164 & L174].

2. Based on the suggestion of reviewer 1, we have added an additional paragraph in the discussion section to emphasize more clearly the climate signal of $\delta^{18}O$ from tree rings[L430-436]. A detailed discussion is given in our response to reviewer 1 (comments 2 and 4).

3. We have added a section about pooling in the discussion chapter [L347-355] and also discuss why four trees are a justifiable number in our response to reviewer 3.

4. Please find our response below to the comments of the reviewers.

We are looking forward to your next evaluation of our revised manuscript.

**Anonymous referee #3**

*The paper "Large-scale climate signals of a European oxygen isotope network from tree-rings" by Balting et al. is a very good example showed how proxy (isotope 18O) data obtained from tree rings can be used to quantify the past climate patterns over western and central Europe for the last 400 years.*

*I read the revised MS after first round of revision process and noted that the authors carefully followed the most (90-95%) of both reviewers recommendations improving all sections of their MS step-by-step.*

*The overall impression of the revised paper is very good. The logical structure of the manuscript, an in-deep introduction, a detailed description of the methods, visible connections between results and their discussion are noteworthy.*

*Taking into account the authors explanation why they used two terms (PCA and EOF) in their answers to the reviewers, theoretically, EOF is a part of PCA.*

R: We are glad that the reviewer found our revised manuscript suitable for publication. We are grateful for the helpful comment regarding EOF and PCA. According to the suggestions of both reviewers, we have replaced the word "separate analysis" with "corresponding analysis" [L164].

*I understand that the stable isotope measuring in the wood is a time-consuming and an expensive procedure but possibly in further works the authors will explain why four trees (two increment cores) (See section 90) are enough to guarantee the spatial-temporal statistical robustness of obtained time-series for the considered isotope network. Possibly high variation of 18O measurements even for one habitat is one of the reasons to lost a connection between spatial isotope pattern and ENSO signal before 1850.*

*Nevertheless I suggest the paper can be published as it is.*

R: We agree wholeheartedly with the reviewer that measuring stable isotopes in the wood is time-consuming and expensive. In order to be able to ensure the quality and standards in this international project, with many laboratories involved, the methods applied (tree selection, sampling, pooling, isotope analyses) were harmonized and adjusted among the laboratories involved in the establishment of our data set, making it rather homogenous in this regard. How many trees are needed for a pooled chronology cannot be answered, as this depends on the spatial conditions as well as climatological conditions and inter-tree variability. The number of four trees was chosen to introduce a standard for the whole project. This is the

only way to establish a tree-ring stable isotope network from more than 20 sites across Europe within a reasonable time frame. This problem can be overcome by comparison with the increasing number of isotope chronologies from other, additional sites and/or tree species being available to the community. Also, future comparison with reconstructions from other proxy archives, relying on various climate-parameter relationships, will help to test and challenge the data set and interpretation presented here. This would be a great topic for a second manuscript!

**Anonymous referee #2**

...............................................................................

*The authors improved the manuscript but still some issues remain:*

*1. EOF Analysis: Please check e.g. https://atmos.washington.edu/~dennis/552_Notes_4.pdf or other publications, text books etc: The empirical orthogonal function (EOF) analysis decomposes a data set in terms of orthogonal basis functions which are determined from the data. The term EOF analysis is also interchangeable with the geographically weighted PCA (principal component analysis). From reading the answer to reviewer 1 I get the impression that the lead author does not fully understand what the EOF (PCA) analysis is. May there are some typos in the formula presented.*

R: We thank the reviewer for the comment/recommendation regarding the usage of the terms EOF and PCA. Based on the aforementioned comment, it can be stated that we understand EOF and PCA in the same way as the reviewer. The calculation of EOFs is definitely part of PCA, so we have replaced the word "separate analysis" with "corresponding analysis". Furthermore, we have removed the typo from the formula [L164 & L174].

*2. A clear discussion of the former major comment 2 is missing in the manuscript: The authors need to say that a signal recorded by the trees can originate from different source regions during different seasons and that this can have strong implications in the interpretation of the proxies used. To make a simple example. You have a yearly value of delta18O and we just look at one year. The Value is obtained be a mixture of seasonal signals. In Europe the winter circulation and summer circulation deviate dramatically. So you can think of a multitude of combinations between a certain winter circulation (e.g. strong NAO leading to moisture transport from the Atlantic) and a predominant southwesterly flow leading to transport from the Mediterranean. So, your proxy is mixing both signals and I doubt that it is possible with one proxy to say something about this specific circulation of that at least the authors need to discuss the problems related with delta18O from trees.*

R: We agree with the reviewer that it is helpful to have a more detailed description of the problems of the mixed signal of different seasons. This can influence the strength and the variability of the signal. In this respect, we have added a new paragraph in the discussion section where we deal with this issue [L430-436].
Regarding the circulation and transport issue please check our answer to comment 4.

*3. The authors still not mention the problem related to pooling versus not pooling trees to measure delta18O. Please discuss this issue when introducing the data and also in the discussion section, see Hangartner et al. 2012 Methods to merge overlapping tree-ring isotope series to generate multi-centennial chronologies.*

R: To address this issue, we have added a paragraph in the discussion [L347-355].

*4. I disagree with the answers made to mayor comment 3. In winter you find as leading mode diploe structures over Europe. Furthermore, the authors state "The ENSO anomalies (either El Niño or La Niña) develop in winter and it needs 3-6 months to see a signal in the European climate. This lagged relationship is typical for many ENSO related teleconnections. This long transition from the tropical Pacific to central North Atlantic affects in turn the large-scale atmospheric circulation and as a consequence the climate over Europe, especially in spring and summer.". Again an effect on the large scale atmospheric circulation is not necessary as with an unchanged atmospheric circulation over the Atlantic the warming induced by ENSO you change the delta18O at the source region and then transport it to Europe. So I suggest that the authors shall be more careful in the interpretation of the data. They only show statistical relations ships (not causal ones) and sometimes they are rather weak.*

R: We agree with the reviewer that it is necessary to be careful in the interpretation of the data. However, we do not agree with the explanation that the variability of the $\delta^{18}O$ signal in trees is based purely on changes in temperature of the source region. We think that this statement ignores several points. First, it is important to say that $\delta^{18}O$ in trees does not represent a homogeneous temperature signal. In our previous calibrations, we have achieved much higher local correlation with a drought index (e.g., the Standardized Precipitation Evapotranspiration Index) and VPD (vapor pressure deficit) than with temperature and precipitation. This indicates that $\delta^{18}O$ signal is much more a mix of temperature, precipitation, $\delta^{18}O$ in precipitation, moisture availability and other variables. In addition, the signal and the importance of the variables depends on the habitat of the tree. Therefore, the statement that an increase in temperature leads linearly to a change in $\delta^{18}O$ in the tree might be misleading, as $\delta^{18}O$ in trees depends on many other factors.

Furthermore, the transport does not remain constant. For example, El Nino can lead to a certain change in the transport (e.g., Fraedrich and Müller, 1992; Fraedrich, 1994).

Basically, the water vapor transport leads to a change in the $\delta^{18}O$ signature in the atmosphere. This is based, among other things, on the fact that $^{18}O$ tends to condense and precipitate first (for more information see Dansgaard 1964). Thus, if transport paths and the duration of transport from the source to the sink changes, this will lead to a change in the $\delta^{18}O$ signal detected by trees. We agree with the reviewer that the $\delta^{18}O$ source signal can also change, but it must also be considered that transport processes have a strong signature on the $\delta^{18}O$ ratio. We have added a paragraph in the discussion about the changes of the source signal [L430-436].

If the consideration were so simple that the $\delta^{18}O$ source value determines the variability of the isotope ratio and transport is negligible, we would get high correlations with the climate variables in the source area. But these do not exist.

*5. The 20CR reanalysis is not described. It remains unclear what they use (ensemble mean or individual member, I guess it is the ensemble mean) I strongly recommend to use the individual ensemble members in this analysis so that the authors can assess the uncertainty of their results related to the uncertainty of the reanalysis product.*

R: We thank the reviewer for the helpful comment. We have added a paragraph about the used data (ensemble mean) from the 20CR reanalysis (Compo et al., 2011) [L112-117]. Furthermore, we agree with the reviewer that further analysis of the uncertainties of this reanalysis product can be interesting and helpful for our analysis. As we would have to perform detailed analyses of the uncertainties with all the used climate data sets and time series, this is beyond the scope of the study. We can only refer to other studies that have investigated the uncertainties of individual climate products. Nevertheless, it is a great idea for a subsequent manuscript examining the impact of uncertainties in climate data sets on the relationship with $\delta^{18}O$.

*6. How does Figure 7 look like for PC2? In section 4.5, you interpret it as a pure summer signal, so the question is if they confirm it with a similar analysis.*

R: To answer the question of the reviewer, we have computed the correlation between PC2 and the modelled $\delta^{18}O$ in soil water and precipitation. Based on the figure below, it can be seen that a clear correlation pattern is only visible in the summer season.

[Figure]

Figure 1: Links between the second $\delta^{18}O$ component and the modelled $\delta^{18}O$ in soil water and precipitation from nudged climate simulations with ECHAM5-wiso (Butzin et al., 2014). The upper row is showing the correlation between PC2 and $\delta^{18}O$ in precipitation for winter (A), spring (B), summer (C) and autumn (D). Panels E, F, G, H are the correlation maps for PC2 and $\delta^{18}O$ in soil water for winter, spring, summer and autumn. In all maps, the significant grid cells are coloured.

*7. Extremes: It is just a wording issue but I think the 33% of the data are not an adequate definition to be extreme. (see also my previous comment).*

R: We understand the reviewer's problem with the term extremes. As we have clearly defined the term for our study, it should not cause any problems of understanding for the reader [L192-194]. Furthermore, we have distinguished between high and low extremes in our study, each of which accounts for only approximately 16 % of the data.

*8. Specific comments:*
*Around L60: please discuss in 1-2 sentences the nonlinearty of the ENSO response presented in the existing literature.*

R: We have added two sentences to describe that the ENSO response is not stable and nonlinear with the corresponding references [L70-72].

*L104: If you only use the 12 times series in your analysis – will you get similar results for the period 1850 to 1998 when using all data sets? If yes this would be good if not, e.g., the correlation is as weak as for the period 1600-1850 then the "non-stationarity" of the ENSO signal over time is just due the fact that you include more data after 1850.*

R: The same result cannot be achieved with 12 time series which makes sense. The reason for this is that the described and analysed pattern from the $\delta^{18}O$ isotopes of the network requires the availability of the time series, which show a large eigenvector in the EOF plot. Without these time series, the variance that can be explained by the pattern is not available and therefore the pattern cannot be computed. However, since almost all the time series needed to calculate the pattern are available from 1750 onwards, the EOF pattern can be replicated with the available data from 1750 onwards. We are well aware of this uncertainty and this is the reason why we compare the periods 1750 to 1850 and 1870 to 1905 with Event Coincidence Analysis. The detailed discussion of the spatial as well as temporal limiting factors can already be found in the discussion chapter of the revised manuscript.

*L110,L114: The wording "we want to" is weird.*

R: We agree with the reviewer and have removed the "want to" from the sentences [L118 &. L121].

*L121 and elsewhere: The authors mix using different tense - here, they suddenly sue simple perfect. Please correct the entire manuscript.*

R: We have improved the used tense in the mentioned sentence and checked the entire for other tense mistakes [L129].

*L204. Please remove "Discussion" as this is now a separate section.*

R: According to the suggestion of the reviewer, we have removed the word "Discussion" [L212].

*Section 3.4 I think this is more a evaluation of the data so should be mentioned earlier. (Just a suggestion)*

R: We agree with the reviewer that the sub-section could also be moved further forward. However, our aim with the sub-section is to underline the previously explained results and to highlight them from a different perspective. Therefore, we have decided on this position for the sub-section.

*L314: You did not show droughts, so just write dry conditions.*

R: We have improved it in the revised version of the manuscript [L321].

*Section 4.1 Discuss Hangartner et al 2012 here.*

R: To discuss this issue, we have added a paragraph in the suggested subchapter [L347-355].

*L340 is influencing –> influences*
*L357: is getting -> gets*

R: We have changed the mentioned points in the revised version of the manuscript [L357 & L374].

*L361 Please say how the teleconnection have changed in the publications Rimbu and Felis*

R: We have added a better description of the cited studies [L377-381 & L385-390].

*L384: "However as discussion above" reads bad.*

R: We have removed "as discussion above" from the sentence [L407-408].

*Fig 4,5,6: exchange column and row.*

R: We think the reviewer means the wrong description in the figure caption. We have improved it in the revised version of the manuscript [Fig. 4,5,6].

**References used in our responses:**

Butzin, M., Werner, M., Masson-Delmotte, V., Risi, C., Frankenberg, C., Gribanov, K., Jouzel, J., and Zakharov, V. I.: Variations of oxygen-18 in West Siberian precipitation during the last 50 years, 14, 5853–5869, https://doi.org/10.5194/acp-14-5853-2014, 2014.

Compo, G. P., Whitaker, J. S., Sardeshmukh, P. D., Matsui, N., Allan, R. J., Yin, X., Gleason, B. E., Vose, R. S., Rutledge, G., Bessemoulin, P., Brönnimann, S., Brunet, M., Crouthamel, R. I., Grant, A. N., Groisman, P. Y., Jones, P. D., Kruk, M. C., Kruger, A. C., Marshall, G. J., Maugeri, M., Mok, H. Y., Nordli, ø., Ross, T. F., Trigo, R. M., Wang, X. L., Woodruff, S. D., and Worley, S. J.: The Twentieth Century Reanalysis Project, 137, 1–28, https://doi.org/10.1002/qj.776, 2011.

Dansgaard, W.: Stable isotopes in precipitation, 16, 436–468, https://doi.org/10.1111/j.2153-3490.1964.tb00181.x, 1964.

Fraedrich, K.: An ENSO impact on Europe?, 46, 541–552, https://doi.org/10.1034/j.1600-0870.1994.00015.x, 1994.

Fraedrich, K. and Müller, K.: Climate anomalies in Europe associated with ENSO extremes, 12, 25–31, https://doi.org/10.1002/joc.3370120104, 1992.